# Dual-mode harvest solar energy for photothermal $Cu_{2-x}Se$ biomineralization and seawater desalination by biotic-abiotic hybrid

Sheng-Lan Gong[1,2], YangChao Tian[2], Guo-Ping Sheng [1] ✉ & Li-Jiao Tian [2] ✉

Biotic-abiotic hybrid photocatalytic system is an innovative strategy to capture solar energy. Diversifying solar energy conversion products and balancing photoelectron generation and transduction are critical to unravel the potential of hybrid photocatalysis. Here, we harvest solar energy in a dual mode for $Cu_{2-x}Se$ nanoparticles biomineralization and seawater desalination by integrating the merits of *Shewanella oneidensis* MR-1 and biogenic nanoparticles. Photoelectrons generated by extracellular $Se^0$ nanoparticles power $Cu_{2-x}Se$ synthesis through two pathways that either cross the outer membrane to activate periplasmic Cu(II) reduction or are directly delivered into the extracellular space for Cu(I) evolution. Meanwhile, photoelectrons drive periplasmic Cu(II) reduction by reversing MtrABC complexes in *S. oneidensis*. Moreover, the unique photothermal feature of the as-prepared $Cu_{2-x}Se$ nanoparticles, the natural hydrophilicity, and the linking properties of bacterium offer a convenient way to tailor photothermal membranes for solar water production. This study provides a paradigm for balancing the source and sink of photoelectrons and diversifying solar energy conversion products in biotic-abiotic hybrid platforms.

Whole-cell biotic-abiotic hybrid systems, integrating the functionalities of inorganic nanomaterials and the versatility of biocatalytic networks, provide an innovative avenue in solar energy conversion, bioelectrochemical systems and therapeutics[1–4]. To date, several semiconductors and diverse functionalized organisms have been coupled to transduce solar energy into valuable products, including *Moorella thermoacetica* with CdS nanoparticles for acetic acid generation[4], *Azotobacter vinelandii* with CdS quantum dots for ammonia synthesis[5], and *Methanosarcina barkeri* with NiCu@CdS for methane production[6]. Apart from light-harvester semiconductors, conductive FeS nanoparticles were biosynthesized to wire up electronic abiotic/biotic interfaces in bioelectrochemical systems[1,7]. A

hybrid photothermal therapy was designed by coupling *Shewanella oneidensis* MR-1 with biomineralized photothermal palladium nanoparticles[8]. Indeed, biotic-abiotic hybrids are promising strategies to bridge functional synergies between natural and artificial approaches, endowing organisms with emerging functionality.

For whole-cell hybrid photocatalysis, nanoparticles act as solar energy receivers, and living organisms provide catalytic sites. It has been demonstrated that photoelectrons generated from surface-anchored CdS semiconductors or periplasmic biomineralized CdS nanoclusters could be directly transported into cytoplasmic enzymes for photocatalysis via redox membrane-binding proteins[4,9]. The coupling performance is partly governed by the balance between

[1]CAS Key Laboratory of Urban Pollutant Conversion, Department of Environmental Science and Engineering, University of Science and Technology of China, Hefei 230026, China. [2]National Synchrotron Radiation Laboratory, University of Science and Technology of China, Hefei 230026, China. ✉e-mail: gpsheng@ustc.edu.cn; ljtian@ustc.edu.cn

photoelectron generation and utilization. The photoelectron utilization channel was broadened through the simultaneous fixation of $CO_2$ and $N_2$ to tap the full potential of hybrid[10]. Despite significant progress in the diversity of photoelectron utilization, the utilization site of photoelectrons is limited to the intracellular space, which requires efficient transmembrane interfacial electron transfer. Nonetheless, a limited number of redox membrane-binding proteins results in only a few portions of photoelectrons being captured and converted into chemical products. To circumvent the sluggish kinetics of electron transmembrane diffusion, semiconductors were introduced into the target enzyme sites, such as cytoplasm and periplasm[2,11,12], of non-photosynthetic bacteria for the photocatalysis of acetic acid[2] or hydrogen ($H_2$)[11,12]. Such systems exhibit excellent interfacial photoelectron transfer, while the cellular membrane interferes with intracellular semiconductor-mediated light absorption. Thus, breaking this dilemma and establishing a balance between photoelectron generation and utilization in whole-cell hybrid photocatalysis still require substantial effort.

For multimodal interfacial photoelectron-utilizing organisms, dissimilatory metal-reducing bacteria are promising due to their inherent ability to utilize extracellular inorganic substances through direct or indirect electron transfer chains[13–15]. Among them, *S. oneidensis* MR-1 is one of the most versatile non-photosynthetic species for constructing biotic-abiotic hybrid systems, owing to its unique extracellular electron transfer (EET) chain, which ensures efficient interfacial electron transfer and favors for extracellular biomineralization[16,17]. Our recent work found that *S. oneidensis* MR-1 reversed EET to transport photoelectrons from bio-assembled CdS NPs into periplasmic hydrogenase for $H_2$ production[18]. In particular, *S. oneidensis* MR-1 owns multiple enzymes in the periplasm[12], facilitating short-distance and natural interfacial electron transfer chains between the extracellular semiconductor and the enzyme catalytic site. In addition, the genetic tractability of *S. oneidensis* MR-1 could provide insight into the fundamental mechanisms of electron transfer in hybrid. Meanwhile, *S. oneidensis* possesses a powerful biosynthetic capacity to assemble diverse multifunctional nanoparticles[15,18–20]. Among them, the bio-assembled $Se^0$ semiconductor has good biocompatibility and an appropriate band position to capture solar energy[15]. Biogenic copper chalcogenides have near-infrared region (NIR) absorption and photothermal features, which can be applied in numerous fields across anticancer[21], antibacterial[22], and solar steam generation[23]. Thus, *S. oneidensis* was adopted as the model to construct a whole-cell hybrid photocatalytic system with potential multimodal interfacial photoelectron utilization channels.

Here, we harvest solar energy for solid-state nanoparticle synthesis and seawater desalination by biotic-abiotic hybrid systems. The constructed *S. oneidensis*-$Se^0$ NPs hybrid has two photoelectronic networks to boost photothermal $Cu_{2-x}Se$ NPs biosynthesis. Photothermal $Cu_{2-x}Se$ NPs are selected as the target product, as they offer an alternative pathway, solar-to-thermal-to-water, for solar energy utilization. Ultimately, *S. oneidensis*-$Cu_{2-x}Se$ NPs are incorporated into polyvinylidene fluoride to fabricate a photothermal membrane for desalinating seawater. This work reveals the mechanism of the fate of photoelectron flux in a model hybrid and explores the diversity of applications of biotic-abiotic hybrids for solar energy conversion.

## Results
### Construction of the *S. oneidensis*-$Se^0$ hybrid system
The *S. oneidensis*-$Se^0$ NPs hybrid system was fabricated via incubating *S. oneidensis* MR-1 with 0.5 mM selenite following the procedure in Fig. 1A, Supplementary Fig. 1A. After 10 h of incubation, the solution turned red-orange (Supplementary Fig. 2A), implying the formation of $Se^0$ NPs. The relevant characterization is referred to Supplementary Method 1. The morphology of the extracellular biogenic nanoparticles was captured by scanning electron microscopy (SEM). As shown in

Fig. 1B and Supplementary Fig. 2B, the resulting nanoparticles had uniform spherical shapes with a diameter of approximately 85 nm. The corresponding energy dispersive spectroscopy (EDS) mapping demonstrated that the biogenic particles were composed of Se element (Fig. 1C). As the element compositions of microbial cells, the N and C elements were distributed across the microbial cells (Supplementary Fig. 2C, D). Some of the biogenic $Se^0$ NPs formed large aggregates associated with the outer membrane of the cells. X-ray powder diffraction (XRD) displayed two major diffraction peaks that coincided well with those of the trigonal Se standard phase (COD-9008579) (Supplementary Fig. 2E)[24]. These results confirmed the successful construction of the *S. oneidensis*-$Se^0$ NPs hybrid under environmentally benign conditions.

Furthermore, we quantified the production rate of $Se^0$ NPs by testing time-resolved selenite reduction and $Se^0$ production concentration by referring to Supplementary Method 2 and 3 (Supplementary Fig. 3A). *S. oneidensis* MR-1 could quickly reduce selenite within 8 h. Selenite reduction was well fit by the pseudo-first-order model, and the rate constant was estimated to be − 0.2808 $h^{-1}$ (Supplementary Fig. 3B). In addition, an excellent linear relationship ($R^2 = 0.9999$) between $Se^0$ NPs concentration and absorbance at 550 nm was obtained (Supplementary Fig. 3C), providing an approach for the determination of $Se^0$ NPs content in solution. The formation of $Se^0$ NPs was also well fit by the pseudo-first-order model, with a rate constant of about 0.2366 $h^{-1}$ (Supplementary Fig. 3D). The similar rate constant between selenite reduction and $Se^0$ NPs formation indicates that selenite reduction in *S. oneidensis* MR-1 can be used to reflect $Se^0$ NPs production, consistent with previous work[25].

Light absorption of the constructed *S. oneidensis*-$Se^0$ hybrid was measured by ultraviolet-visible diffuse reflection spectroscopy (UV-vis DRS), which revealed that $Se^0$ NPs exhibited a broad and intense absorption in the visible light region (Fig. 1D). According to the UV-vis DRS and corresponding Tauc plot, the bandgap of biogenic $Se^0$ NPs was calculated to be 1.49 eV (Fig. 1E), which is comparable to that of the conventionally chemically synthesized $Se^0$ NPs[26]. Based on the corresponding Mott-Schottky plot (Fig. 1F), the conduction band of biogenic $Se^0$ NPs was −0.708 eV (vs. SHE). The valence band of $Se^0$ semiconductor was calculated to be 0.782 eV (vs. SHE). The photoactivity of the biogenic $Se^0$ NPs was measured by recording the photocurrent with/without light exposure. An apparent photocurrent was generated under visible light illumination and decreased to the baseline value after turning off the light (Fig. 1G). No photocurrent was observed in the cell-only control group. Overall, these results indicate that extracellularly biogenic $Se^0$ NPs are partly attached to the cell surface and possess visible light photoactivity.

### Light-activating $Cu_{2-x}Se$ NPs synthesis by *S. oneidensis*-$Se^0$ hybrid
Having constructed the *S. oneidensis*-$Se^0$ hybrid, light-activated $Cu_{2-x}Se$ NPs biosynthesis by the as-prepared hybrid was further tested with sodium acetate as an electron sacrificial agent under different conditions (Fig. 2A). Acetate was chosen because *S. oneidensis* MR-1 cannot metabolize it[14]. In this case, the electron flow from respiration is almost wholly suppressed, excluding the interference of metabolic electrons. When Cu(II) was injected into *S. oneidensis*-$Se^0$ hybrid under light illumination, copper was continuously transformed into $Cu_{2-x}Se$ NPs, as evidenced by the enhanced accumulation of copper in the precipitate (Fig. 2B and Supplementary Method 4). However, copper bio-transformation was wholly inhibited in the hybrid under dark conditions, indicating that *S. oneidensis*-$Se^0$ hybrid required light to initiate Cu(II) transformation to $Cu_{2-x}Se$ NPs. Concomitantly, the solution of *S. oneidensis*-$Se^0$ hybrid with light exposure turned brown-black (Supplementary Fig. 4A), a feature of $Cu_{2-x}Se$ NPs[27]. Consistently, the UV-vis-NIR spectrum of the hybrid with light exposure shows apparent absorption in the NIR region (Fig. 2C), which might be due to

the local surface plasmon resonance (LSPR) characteristic of $Cu_{2-x}Se$ NPs[27]. Turning off visible light significantly attenuated the NIR absorption of the hybrid, suggesting that light exposure stimulated $Cu_{2-x}Se$ NPs synthesis. Biogenic $Cu_{2-x}Se$ NPs have unique photothermal properties, leading to a linear increase in temperature rise ($\Delta T$) with raised $Cu_{2-x}Se$ NPs content under illumination (Supplementary Fig. 4B and Supplementary Method 5). Therefore, $\Delta T$ was chosen as an indicator of the amount of $Cu_{2-x}Se$ NPs. The $\Delta T$ of controls were comparable to that of water (Fig. 2D). Only *S. oneidensis*-$Se^0$ hybrid exposed to light produced a distinct photothermal signal (Fig. 2D), further demonstrating that light irradiation boosts the transformation of $Cu_{2-x}Se$ NPs. Overall, these results together suggest light-driven Cu(II) biotransformation to $Cu_{2-x}Se$ NPs by *S. oneidensis*-$Se^0$ hybrid.

SEM-EDS mapping of the photocatalytic hybrid revealed that the formed $Cu_{2-x}Se$ NPs were mainly located in the extracellular space and some were closely associated with microbial cells (Fig. 2E–G). The corresponding Raman spectrum also displayed the characteristic signal of Cu-Se bond around 260 $cm^{-1}$ (Supplementary Fig. 4C). The XRD pattern of the as-prepared $Cu_{2-x}Se$ NPs exhibited four distinct peaks, which were assigned to the (111), (200), (311) and (400) facets of cubic berzelianite phase $Cu_{2-x}Se$ (PDF#06-0680) (Supplementary Fig. 4D). X-ray photoelectron spectroscopy (XPS) was used to clarify the valence states of Cu in the resultant $Cu_{2-x}Se$ NPs under light illumination (Supplementary Fig. 5). The full-scan map of XPS displayed that it

contained both Se and Cu elements (Supplementary Fig. 5A). The corresponding Cu $2p$ spectrum exhibited peaks centered at 931.7 and 951.6 eV, which were assigned to Cu(I). Meanwhile, the peaks at 933.5 and 953.3 eV were attributed to Cu(II) (Supplementary Fig. 5B)[28], indicating the co-presence of Cu(II) and Cu(I) in the $Cu_{2-x}Se$ NPs. According to the fitting results, the Cu(I)/Cu(II) ratio in the $Cu_{2-x}Se$ NPs was about 3.57. The Se $3d$ spectrum was deconvoluted into two peaks[29], namely Se $3d_{5/2}$ (52.2 eV) and Se $3d_{3/2}$ (53.1 eV) (Supplementary Fig. 5C), coinciding with $Se^{2-}$. These results further confirmed the successful photocatalysis of $Cu_{2-x}Se$ NPs by *S. oneidensis*-$Se^0$ hybrid.

### Mechanism of light-boosted $Cu_{2-x}Se$ NPs assembly

$Se^0$ NPs and $Cu_{2-x}Se$ NPs are mainly distributed outside the cell. These findings raise a question of whether biological factors are involved in Cu(II) reduction and transformation. To address this issue, the *S. oneidensis*-$Se^0$ hybrid was subjected to ultrasonication to lyse the cells, followed by a similar process for $Cu_{2-x}Se$ NPs synthesis, and the resulting mixture was denoted as a lysed hybrid and the corresponding procedure can be referred to Supplementary Method 6. Inactivating cells in the hybrid completely abolished Cu(II) reduction and $Cu_{2-x}Se$ NPs synthesis (Fig. 2B–D). These results suggest that Cu(II) transformation and $Cu_{2-x}Se$ NPs synthesis are mediated by microorganisms rather than by spontaneous abiotic reactions. Thus, photo-boosted $Cu_{2-x}Se$ NPs synthesis was realized by coupling $Se^0$ semiconductor and *S. oneidensis* MR-1.

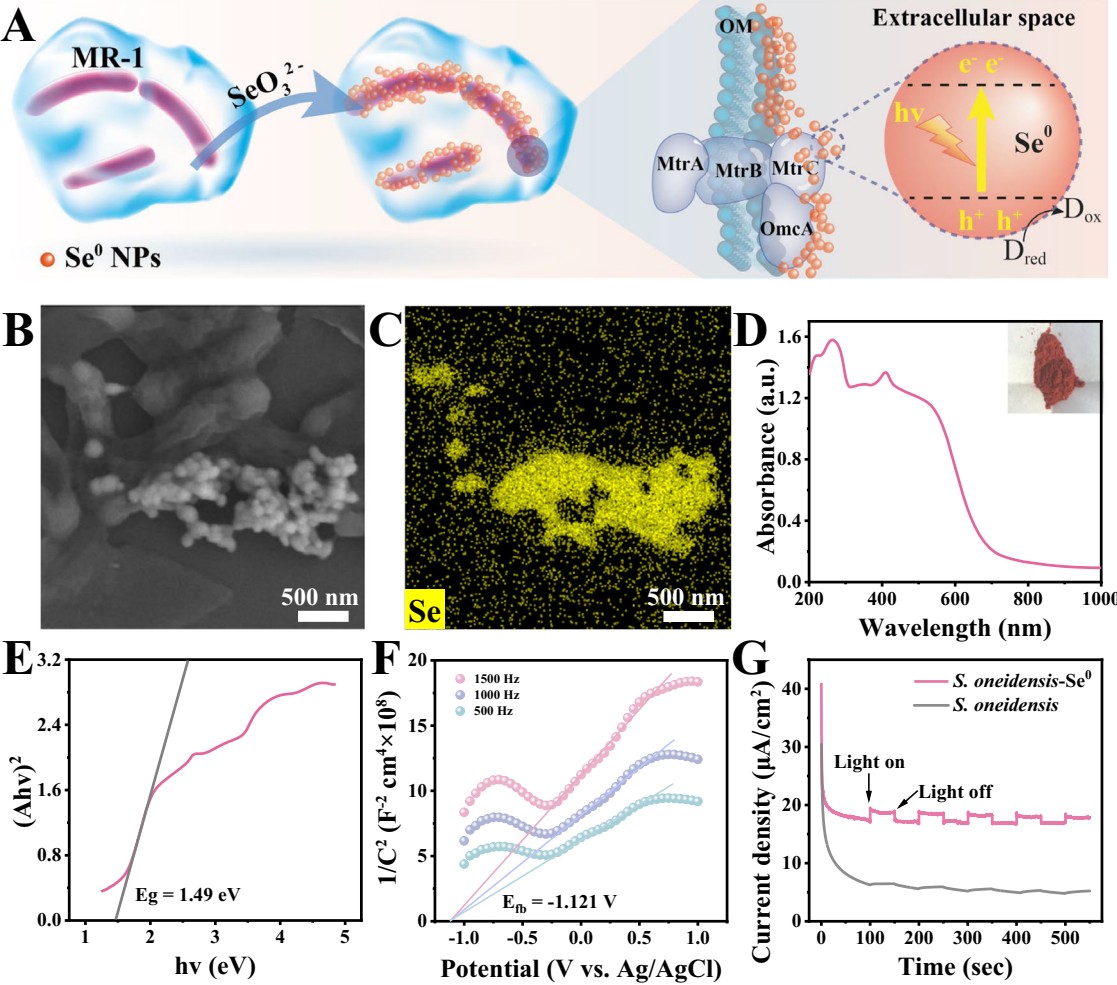

**Fig. 1 | Characteristics of the constructed *S. oneidensis*-$Se^0$ hybrid. A** Schematic diagram of the construction process of *S. oneidensis*-$Se^0$ hybrid. **B** SEM image and (**C**) corresponding EDS mapping image of Se element of the hybrid. **D** UV-vis DRS spectrum, (**E**) Tauc plot, and (**F**) Mott-Schottky plot of *S. oneidensis*-$Se^0$ hybrid. **G** *I-t* curves of the *S. oneidensis*-$Se^0$ hybrid and *S. oneidensis* control with a light on/off cycle (50/50 s). Experiments of (**B**) were repeated three times with similar results. Source data are provided as a Source Data file.

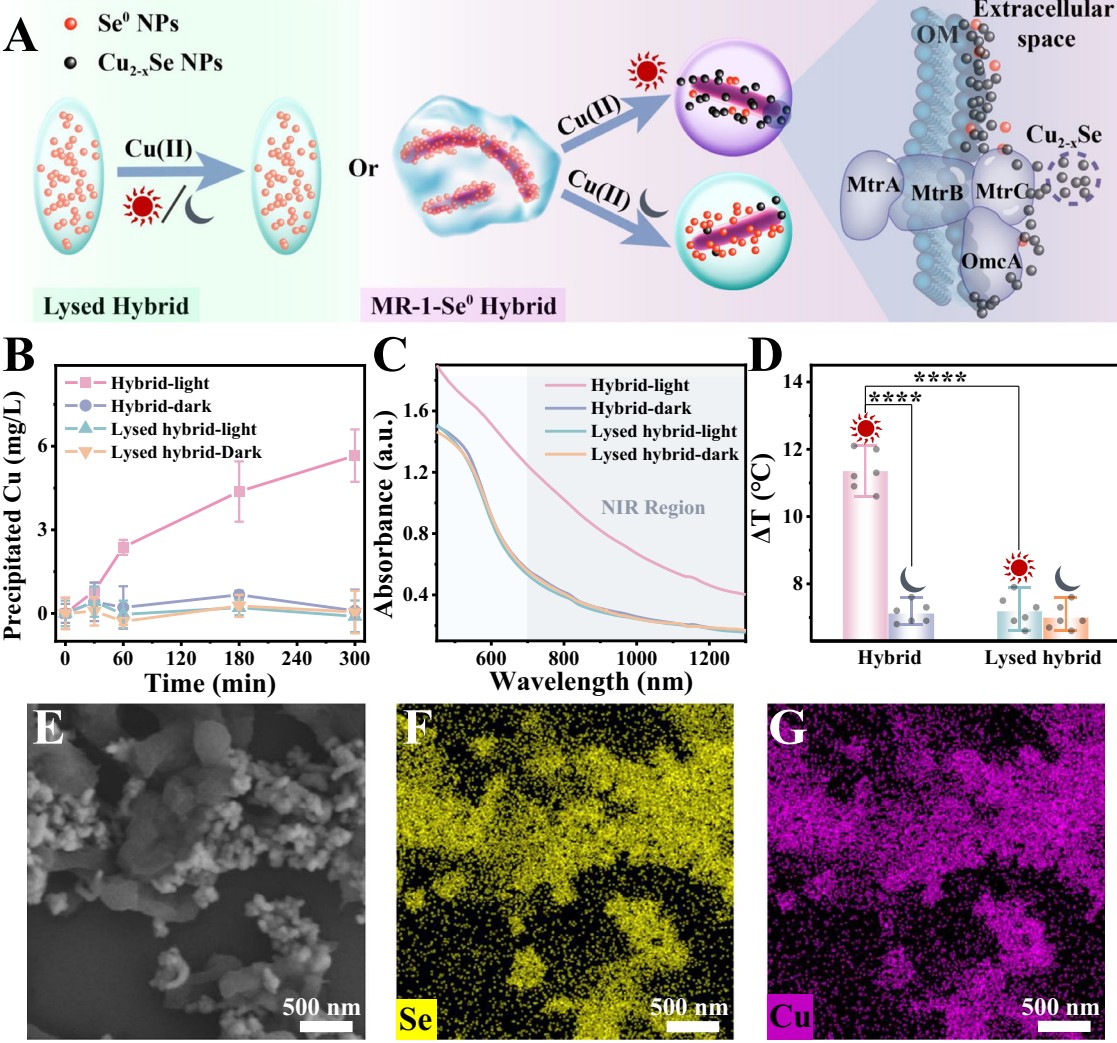

**Fig. 2 | Light-driven Cu$_{2-x}$Se NPs assembly by the *S. oneidensis*-Se$^0$ hybrid and the characteristics of the biogenic Cu$_{2-x}$Se NPs. A** Schematic diagram shows the synthesis of Cu$_{2-x}$Se by the *S. oneidensis*-Se$^0$ hybrid or cell-inactivated lysed hybrid with/without illumination. **B** Time-resolved precipitated copper concentration by the *S. oneidensis*-Se$^0$ hybrid and cell-inactivated lysed hybrid with/without illumination. After 5 h of Cu(II) treatment, (**C**) UV-vis-NIR absorption spectra and (**D**) temperature rise (ΔT) of hybrid and lysed hybrid with/without illumination. **E** SEM image and the corresponding EDS mapping images of (**F**) Se and (**G**) Cu elements of the photosynthesized *S. oneidensis*-Cu$_{2-x}$Se nanoparticles. The above experiments

were performed in the mineral salt medium with 20 mM sodium acetate. The data points represented in (**B**) represent three ($n = 3$) and in (**D**) represent six ($n = 6$) independent experiments for each experimental group and are displayed as mean ± standard deviation (SD). $p$ values of (**D**) were determined by a one-way analysis of variance. "\*\*\*\*" represents $p < 0.0001$. Experiments of (**E**) were repeated three times with similar results. Source data and frequency inference statistics (the value of degrees of freedom, $p$, effect size statistic, 95% confidence intervals) are provided as a Source Data file.

To uncover the pathway of electron flow during normal metabolism, sodium acetate was replaced with sodium lactate, which can act as both a carbon source and an electron donor (Fig. 3). To gain insight into the underlying fundamental mechanism of light-boost Cu$_{2-x}$Se NPs biosynthesis by *S. oneidensis*-Se$^0$ hybrid, we first focused on the fate of copper during photocatalysis. We compared copper accumulation and Cu$_{2-x}$Se NPs production using a cell-lysed-Se$^0$ hybrid, which was metabolically inactive but retained the photoactivity of Se$^0$ semiconductors (Fig. 2A). Copper precipitation and Cu$_{2-x}$Se NPs production were almost negligible in the system that co-incubated lysed hybrid with Cu(II) (Fig. 3B, C), further confirming that the reaction between Se$^0$ NPs and Cu(II) to form Cu$_{2-x}$Se NPs in *S. oneidensis*-Se$^0$ hybrid system was mediated by microorganisms. In contrast, the inactivated hybrid co-incubated with Cu(I) showed apparent copper conversion and Cu$_{2-x}$Se production, supported by the enhanced Cu precipitation and raised ΔT (Fig. 3B, C). Moreover, such a procedure was strengthened by light exposure. These results demonstrate that Cu$_{2-x}$Se NPs

might be formed through a spontaneous abiotic reaction between Se$^0$ NPs and Cu(I), which could be accelerated by the photoactivity of the Se$^0$ semiconductor. Thus, *S. oneidensis* was speculated to be involved in Cu(II) reduction during the photocatalytic synthesis of Cu$_{2-x}$Se NPs. Consistently, the intermediate Cu(I) was detected in the supernatant and increased continuously over time after injecting Cu(II) into the hybrid system with sodium lactic acid as the carbon source (Supplementary Fig. 6A and Supplementary Method 4), suggesting that *S. oneidensis* has the capacity to reduce Cu(II) to Cu(I). Similarly, previous works reported that Cu(II) could be reduced by MR-1[13,14,30]. Together, two steps are coupled in the biosynthetic procedures of Cu$_{2-x}$Se NPs, including Cu(II)-to-Cu(I) and Cu(I)-to-Cu$_{2-x}$Se NPs. The first step is the biological reduction of Cu(II), which leads to the accumulation of intermediate Cu(I), the precursor for further Cu$_{2-x}$Se NPs formation. Subsequently, the resulting Cu(I) reacts with Se$^0$ NPs, undergoing abiotic conversion to synthesize Cu$_{2-x}$Se NPs, which can be stimulated by light irradiation.

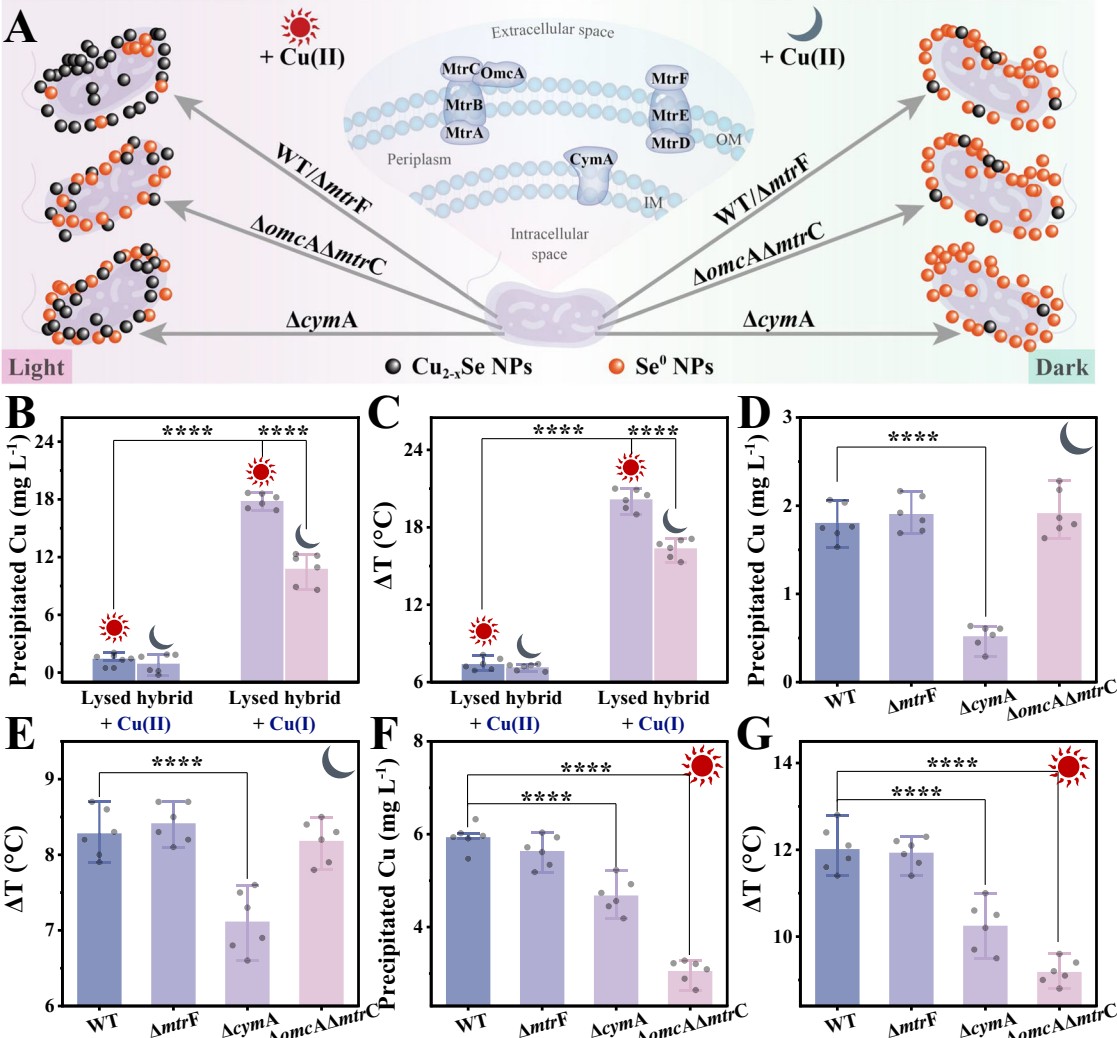

**Fig. 3 | Mechanism of the light-driven Cu$_{2-x}$Se NPs assembly by the *S. oneidensis*-Se$^0$ hybrid. A** Schematic diagram shows the synthesis of Cu$_{2-x}$Se by lysed hybrids that were co-incubated with different strains and Cu(II) with/without illumination. **B** The precipitated copper concentration and (**C**) temperature rise (ΔT) of the lysed hybrids that were co-incubated with Cu(II) or Cu(I) for 5 h with/without illumination. **D** The precipitated copper concentration and (**E**) (ΔT) of the lysed hybrids that were co-incubated with different strains and Cu(II) for 5 h under dark conditions. **F** The precipitated copper concentration and (**G**) (ΔT) of the lysed hybrids that were

co-incubated with different strains and Cu(II) for 5 h under light illumination. The above experiments were performed in the mineral salt medium with 20 mM sodium lactate. The data points represented in (**B**–**G**) represent six ($n = 6$) independent experiments for each experimental group and are displayed as mean ± standard deviation (SD). *p* values of (**B**–**G**) were determined by a one-way analysis of variance. "****" represents $p < 0.0001$. Source data and frequency inference statistics (the value of degrees of freedom, *p*, effect size statistic, 95% confidence intervals) are provided as a Source Data file.

*S. oneidensis* MR-1 owns a unique respiratory pathway to deliver electrons from cytoplasm to periplasm and then to extracellular space. The metal respiratory system (Mtr) pathway is embedded in the outer membrane and is the primary electron transfer conduit for interfacial electron transfer from the periplasm to the extracellular material. The MtrC and OmcA proteins are essential cytochromes in the Mtr pathway (Fig. 3A), and play a critical role in extracellular metal ions reduction[18]. Thus, we speculate that they might be involved in Cu(II) reduction. To test this hypothesis, we employed MtrC- and OmcA-impaired mutant, Δ*omc*AΔ*mtr*C, to construct a hybrid for Cu$_{2-x}$Se NPs synthesis with sodium lactate supplementation. Lactate acts as an electron donor for bacterial respiration and as a sacrificial photoelectron donor for photocatalysis[18]. To rule out changes in Se$^0$ NPs content caused by inactivated proteins, we performed mutation experiments by mixing different strains and cell-lysed hybrid. Inactivation of the MtrC and OmcA proteins in *S. oneidensis* MR-1 had little effect on copper conversion or Cu$_{2-x}$Se NPs production, as supported by the similarity of the precipitated copper content and ΔT between the wild-type (WT)

and Δ*omc*AΔ*mtr*C under dark conditions (Fig. 3D, E). Knocking out MtrF, another important cytochrome in the Mtr pathway(Fig. 3A)[31], resulted in no appreciable change in Cu$_{2-x}$Se NPs formation (Fig. 3D, E), indicating that MtrF is also not required for Cu(II) transformation. Considering that the Mtr pathway, the primary electron transfer conduit for extracellular reduction, is not involved in copper transformation, Cu(II) reduction mainly occurs intracellularly.

To elucidate the exact Cu(II) reduction site, we further coupled CymA-inactivated strain (Δ*cym*A) with a cell-lysed hybrid for Cu$_{2-x}$Se NPs synthesis. CymA is a critical cytoplasmic membrane-anchored c-type cytochrome that transfers electron equivalents from central metabolism to periplasm (Fig. 3A)[32]. Deleting CymA protein severely inhibited copper conversion toward Cu$_{2-x}$Se NPs production under dark conditions (Fig. 3D, E, Supplementary Fig. 6B and Supplementary Method 4), indicating that CymA is an essential protein for Cu(II) reduction and Cu$_{2-x}$Se NPs formation. Interestingly, the elimination of Cu(II) transformation using Δ*cym*A strain could be recovered by light (Fig. 3F, G). When exposed to light, Δ*cym*A exhibited a slightly lower

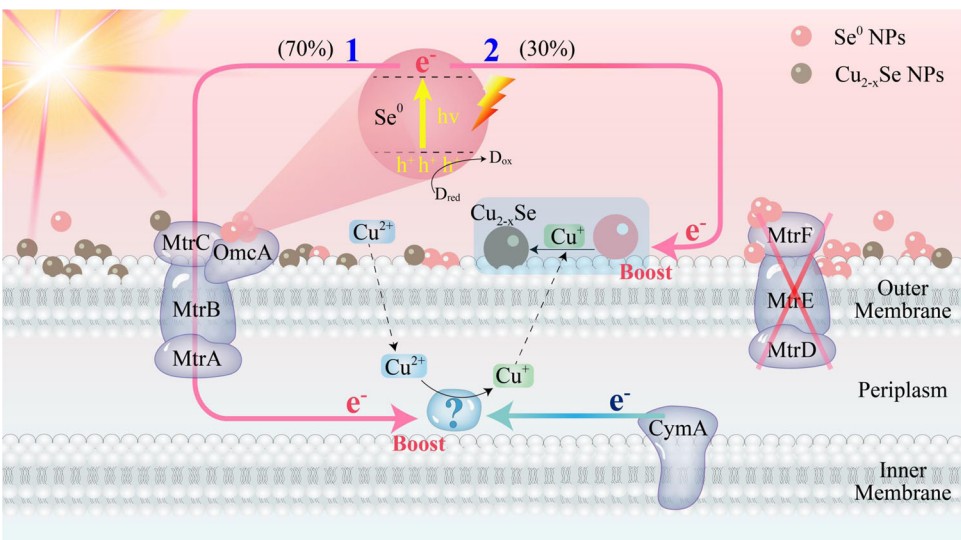

**Fig. 4 | Proposed mechanism of the light-driven Cu$_{2-x}$Se NPs assembly by the *S. oneidensis*-Se$^0$ hybrid.** Numbers 1 and 2 show the two utilization models of photoelectrons, which were generated by Se$^0$ semiconductors.

copper accumulation and transformation than that obtained in WT (Fig. 3F, G), implying that CymA protein is not a Cu(II) terminal reductase, and light exposure activates another electron channel for Cu$_{2-x}$Se NPs photocatalytic synthesis. Based on the above results, CymA protein predominantly serves as an upstream of Cu(II) terminal reductase, working as a relay to transport metabolic electrons for Cu(II) reduction. Altogether, these results suggest that Cu(II) terminal reductases and reduction space are predominant in the periplasm.

Considering the critical role of periplasmic fumarate reductase FccA in selenite reduction in *S. oneidensis*[25], we were curious whether FccA possesses the capacity to reduce Cu(II) and synthesize Cu$_{2-x}$Se NPs. To test this possibility, we measured and compared the Cu$_{2-x}$Se NPs biotransformation by control and mutant with impaired FccA ability (Δ*fcc*A). However, Cu$_{2-x}$Se NPs produced by Δ*fcc*A (reflected by Cu bioaccumulation and ΔT index) was comparable to that produced by WT strain (Supplementary Fig. 7), indicating that FccA did not contribute to Cu(II) reduction or Cu$_{2-x}$Se NPs biosynthesis, with Se$^0$ serving as the precursor. Furthermore, the identification of Cu(II) terminal reductases remains to be fully explored.

Cu(II) reduction space is the periplasm. In comparison, Se$^0$ semiconductor and Cu$_{2-x}$Se NPs are mainly distributed outside the cell. These results raise another question of whether extracellular photoelectrons generated from Se$^0$ NPs can be injected into periplasmic Cu(II) terminal reductases to form Cu(I) and how they are connected. To clarify this, we employed Δ*omc*AΔ*mtr*C mutant to construct a hybrid for photocatalytic biosynthesis, as this strain destroyed the interface electron pathway between periplasm and extracellular space. Compared with the WT, Δ*omc*AΔ*mtr*C displayed a substantial decrease in Cu(II) transformation and Cu$_{2-x}$Se NPs production under light illumination (Fig. 3F, G). Differently, such suppression was not observed in the hybrid constructed using Δ*mtr*F mutant. These results demonstrate that Se$^0$-generated photoelectron flux can be injected into periplasmic Cu(II) reduction via OmcA- and MtrC-involved electron conduit. Given the negative potential of Se$^0$ NPs relative to the outer membrane cytochromes OmcA (−25 to −325 mV vs SHE at PH = 6) and MtrC (−0.5 to −277.5 mV vs SHE at PH = 6)[18], high-energy photoelectrons from Se$^0$ NPs can be injected into periplasms via reversing OmcA- and MtrC-involved electron conduit.

Interestingly, photo-induced Cu$_{2-x}$Se NPs formation was not observed in Δ*omc*AΔ*mtr*C strain, which was supported by the comparable ΔT and precipitated Cu content in Δ*omc*AΔ*mtr*C under light illumination and dark conditions (Fig. 3D−3G). Consistent with the

above-mentioned results, MtrC- and OmcA-composed electron conduit only plays a crucial role in photoelectron delivery but does not participate in metabolic electron transport for Cu$_{2-x}$Se NPs formation. Thus, removing MtrC and OmcA proteins eliminates the light-stimulated synthesis, but does not affect metabolic electron-mediated transformation. The remaining Cu$_{2-x}$Se synthesis capacity of Δ*omc*AΔ*mtr*C strain with/without light illumination was attributed to the metabolically driven synthesis. Furthermore, this similar result further indicates that the MtrC- and OmcA-composed conduit is the main photoelectron transport pathway under light illumination.

In combination with the light-boosting conversion of Cu(I) to Cu$_{2-x}$Se NPs (Fig. 3B, C), *S. oneidensis*-Se$^0$ hybrid possesses a dual mode for photoelectron utilization, including periplasmic Cu(II) reduction and extracellular Cu(I) conversion, favoring the establishment of a balance between photoelectron generation and utilization in whole-cell hybrid photocatalytic biosynthesis. Together, the CymA-delivered metabolic electron channel and Mtr-directed photoelectron conduit are coupled for the photocatalytic biosynthesis of Cu$_{2-x}$Se NPs.

To further determine the direction of electron flow, we compared nitrate reduction by *S. oneidensis* MR-1 with/without light illumination (Supplementary Fig. 8). The selection of nitrate as an electron acceptor was due to the known periplasmic location of nitrate reductase in *S. oneidensis* MR-1 (Supplementary Fig. 8A)[33]. Under light conditions, nitrate reduction by the *S. oneidensis*-Se$^0$ hybrid was significantly higher than that obtained under dark conditions (Supplementary Fig. 8B). Impairing MtrABC cytochrome conduit led to a substantial decrease in nitrate reduction under light illumination, suggesting that the periplasmic reaction can be driven by external photoelectrons through the reverse MtrABC pathway. Meanwhile, the small amount of reduced nitrate in the dark likely comes from the remaining organic carbon in the biomass, which could be oxidized to produce electrons for nitrate reduction. Under these conditions, NO$_2^-$ makes up the majority of the end product (Supplementary Fig. 8C). Overall, the MtrABC pathway works in a reverse direction to take up photoelectrons from the Se$^0$ semiconductor to the periplasmic electron acceptor.

Based on the above results, a schematic diagram for Cu(II) evolution and Cu$_{2-x}$Se NPs photocatalytic biosynthesis by *S. oneidensis*-Se$^0$ hybrid is proposed in Fig. 4. Upon incubation of the *S. oneidensis*-Se$^0$ hybrid with the precursor, Cu(II) is taken up by metabolically active cells and then biologically reduced in the periplasm, forming the intermediate Cu(I). Cu(II) terminal reductase is located in the

periplasm and is a downstream protein of CymA. Normally, enzyme-catalyzed Cu(II) reduction is initiated by intracellular metabolic electrons. Metabolic electrons originate from the metabolism of carbon source and then be transported from the cytoplasm to the periplasmic Cu(II) reductases through CymA. Under light illumination, Cu(II) reduction is activated by another photoelectron flux. $Se^0$ NPs generated photoelectrons cross the outer membrane to activate periplasmic Cu(II) reduction via reversing OmcA- and MtrC-involved electron chain. The intermediate Cu(I) is then exported to the extracellular space, and subsequently spontaneously reacts with $Se^0$ NPs undergoing abiotic reaction to assemble $Cu_{2-x}Se$ NPs, which can be boosted under light irradiation.

Light-driven $Cu_{2-x}Se$ NPs synthesis via a two-step procedure, not only stimulates the periplasmic Cu(II) reduction network but also promotes the evolution of Cu(I) toward $Cu_{2-x}Se$ NPs. Based on the concentration of bio-transformed Cu, as shown in Supplementary Table 1 and Fig. 3D, F, the percentages of extracellular $Se^0$ NPs-generated photoelectron delivery to periplasmic Cu(II) reduction and extracellular Cu(I) evolution were 70% and 30%, respectively. We calculated that $Cu_{2-x}Se$ NPs produced by the *S. oneidensis*-$Se^0$ hybrid at 1 mM lactate consumption with or without light were about 21.80 mg and 11.74 mg, respectively (Supplementary Table. 2 and Supplementary Method 7,). The amount of $Cu_{2-x}Se$ NPs under light illumination was 1.86 times that under dark conditions. Although *S. oneidensis* MR-1 has been reported to be capable of reducing Cu(II)[13,14,30], the specific terminal reductase and detailed reduction site remain elusive. Here, we identified the Cu(II) terminal reductase in the periplasm. Lactate acts as an electron donor for metabolic electron-mediated Cu(II) reduction and a sacrificial electron donor for $Se^0$ NPs photocatalyst under light exposure.

Beyond $Cu_{2-x}Se$ NPs bio-assembly, we also tested light-promoted biological processes, including nitrate reduction through Supplementary Method 8 (Supplementary Fig. 8) and HgSe biosynthesis (Supplementary Fig. 9) by *S. oneidensis*-$Se^0$ hybrid system. As expected, significantly promoted nitrate reduction, Hg biotransformation, and HgSe synthesis under sunlight illumination were observed. Similarly, abundant HgSe NPs were synthesized in *S. oneidensis*-$Se^0$ hybrid under exposure to Hg with light illumination, but were significantly hindered in the dark. The XRD pattern shows distinct 2θ peaks matching the (111), (220), and (311) facets of HgSe (PDF#08-0469) (Supplementary Fig. 9B). Thus, the biological process in *S. oneidensis*-$Se^0$ hybrid system was drastically facilitated by light illumination, indicating the universal applicability of the photo-driven bio-hybrid system.

## Photothermal membranes constructed from S. oneidensis-$Cu_{2-x}Se$ for solar water production

Attracted by the unique photothermal feature of biogenic $Cu_{2-x}Se$ NPs and the naturally hydrophilic properties of bacterial cells, we further explored their potential application in solar vapor generation. The solar steam generation device was fabricated by encapsulating photocatalytic biogenic $Cu_{2-x}Se$ NPs (containing bacterial cells) in a polyvinylidene fluoride (PVDF) substrate through a phase inversion approach referring to Supplementary Method 9 (Fig. 5A, Supplementary Fig. 1B, and Supplementary Table 3)[27]. The pristine PVDF membrane was white (Fig. 5B, insert). In contrast, it turned black with the incorporation of *S. oneidensis*-$Cu_{2-x}Se$ NPs (denoted as Bio-$Cu_{2-x}Se$@MR-1@PVDF) (Fig. 5B, insert), indicating a broad solar energy absorption after $Cu_{2-x}Se$ NPs modification. As expected, Bio-$Cu_{2-x}Se$@MR-1@PVDF displayed excellent solar absorption over a wide wavelength range from 250 to 1500 nm by loading photocatalytic biosynthesized $Cu_{2-x}Se$ NPs as the solar absorber (Fig. 5B). Raman mapping of the Bio-$Cu_{2-x}Se$@MR-1@PVDF clearly shows that the Cu-Se bond signal (260 cm$^{-1}$), corresponding to $Cu_{2-x}Se$ NPs, is almost uniformly distributed in the solar evaporator (Fig. 5C). Meanwhile, SEM

displayed that the membrane exhibited a microchannel structure with a dense surface and finger-like pore (~10 μm in diameter) (Fig. 5D). These results demonstrate that biogenic $Cu_{2-x}Se$ NPs were well encapsulated in the membrane matrix, and the formed pores facilitated water transportation and light absorption.

Compared with the smooth surface of the blank PVDF membrane, the surface of Bio-$Cu_{2-x}Se$@MR-1@PVDF was rougher (Supplementary Fig. 10A). The surface hydrophilicity of Bio-$Cu_{2-x}Se$@MR-1@PVDF membrane was higher than that of the PVDF membrane, as evidenced by the smaller contact angle (57.6° vs. 82.0°) (Supplementary Fig. 10B). To illustrate the enhanced hydrophilicity of PVDF membrane after being incorporated with $Cu_{2-x}Se$ NPs, the corresponding functional groups on photothermal membrane were detected by attenuated total internal reflectance Fourier transform infrared spectroscopy (ATR-FTIR) (Supplementary Fig. 10C). The ATR-FITR spectrum of the pristine membrane shows distinct signals of vibration bonds at 1275, 1178, and 875 cm$^{-1}$, which are considered to be the characteristic peaks of PVDF. After the introduction of Bio-$Cu_{2-x}Se$ NPs, we observed abundant new hydrophilic groups, including amides (1660 cm$^{-1}$ and 1545 cm$^{-1}$) and hydroxyl groups (1395–1440 cm$^{-1}$ and 3500–3700 cm$^{-1}$)[34]. These hydrophilic functional groups may be assigned to the incorporated proteins in bacteria, providing favorable conditions for water transportation and evaporation.

The solar vapor generation performance of Bio-$Cu_{2-x}Se$@MR-1@PVDF was evaluated by recording the real-time water loss under simulated solar irradiation (1 kW m$^{-2}$, 1 sun). Solar steam generation was substantially enhanced by embedding *S. oneidensis*-$Cu_{2-x}Se$ nanoparticles in PVDF membrane (Fig. 5E). The solar evaporation rate of Bio-$Cu_{2-x}Se$@MR-1@PVDF is 1.44 kg m$^{-2}$ h$^{-1}$, which is much higher than that of the pristine PVDF membrane and pure water (Fig. 5F). After 1 h of light illumination, the surface temperature of Bio-$Cu_{2-x}Se$@MR-1@PVDF remarkably increased from 21.8 °C to 41.2 °C (Supplementary Fig. 10D), indicating that the effective localized heating effect of the loaded $Cu_{2-x}Se$ NPs can significantly elevate the temperature for efficient steam generation. The solar evaporation efficiency of Bio-$Cu_{2-x}Se$@MR-1@PVDF was calculated to be 90.55%, which is among the reported top-notch plasmonic systems (Fig. 5G and Supplementary Tables 4 and 5). Additionally, Bio-$Cu_{2-x}Se$@MR-1@PVDF shows a long-term stable and durable evaporation performance over 10 cycles of continuous operation under 1 sun illumination according to Supplementary Method 10, as supported by the steady maximum temperature and photothermal conversion efficiency (Supplementary Fig. 10E, F). The durable solar desalination performance of Bio-$Cu_{2-x}Se$@MR-1@PVDF was investigated by immersing the fabricated photothermal membrane in artificial saline water (0.5 M NaCl solution) and real seawater (Fig. 5E, F and Supplementary Fig. 10E). Salt ions did not influence water evaporation, in terms of solar evaporation rate or photothermal conversion efficiency, and the as-prepared photothermal membrane exhibited good salt resistance. The concentrations of Na$^+$, K$^+$, Ca$^{2+}$, Mg$^{2+}$, B$^{3+}$, and Cu$^{2+}$ content in distilled water were measured according to Supplementary Method 11. They were found to be much lower than the standards of drinkable desalination water set by the World Health Organization (WHO, 2022) (Supplementary Fig. 11A). TOC concentration in the distilled water was below the U.S. Environmental Protection Agency (EPA, 2017) standard for indirect potable reuse of 2 mg/L (Supplementary Fig. 11B)[35]. Meanwhile, protein content in the distilled water was negligible (Supplementary Fig. 11B), satisfying the reusable standards.

To elucidate the underlying mechanism of the excellent performance of the Bio-$Cu_{2-x}Se$@MR-1@PVDF, chemically synthesized $Cu_{2-x}Se$ nanoparticles (Chem-$Cu_{2-x}Se$ NPs) were prepared (Supplementary Method 12)[36]. TEM image revealed that the average size of the Chem-$Cu_{2-x}Se$ NPs was 84.3 ± 6.6 nm (Supplementary Fig. 12A, B). The XRD pattern shows four characteristic peaks corresponding to the (111), (220), (311), and (400) lattice planes of standard $Cu_{2-x}Se$

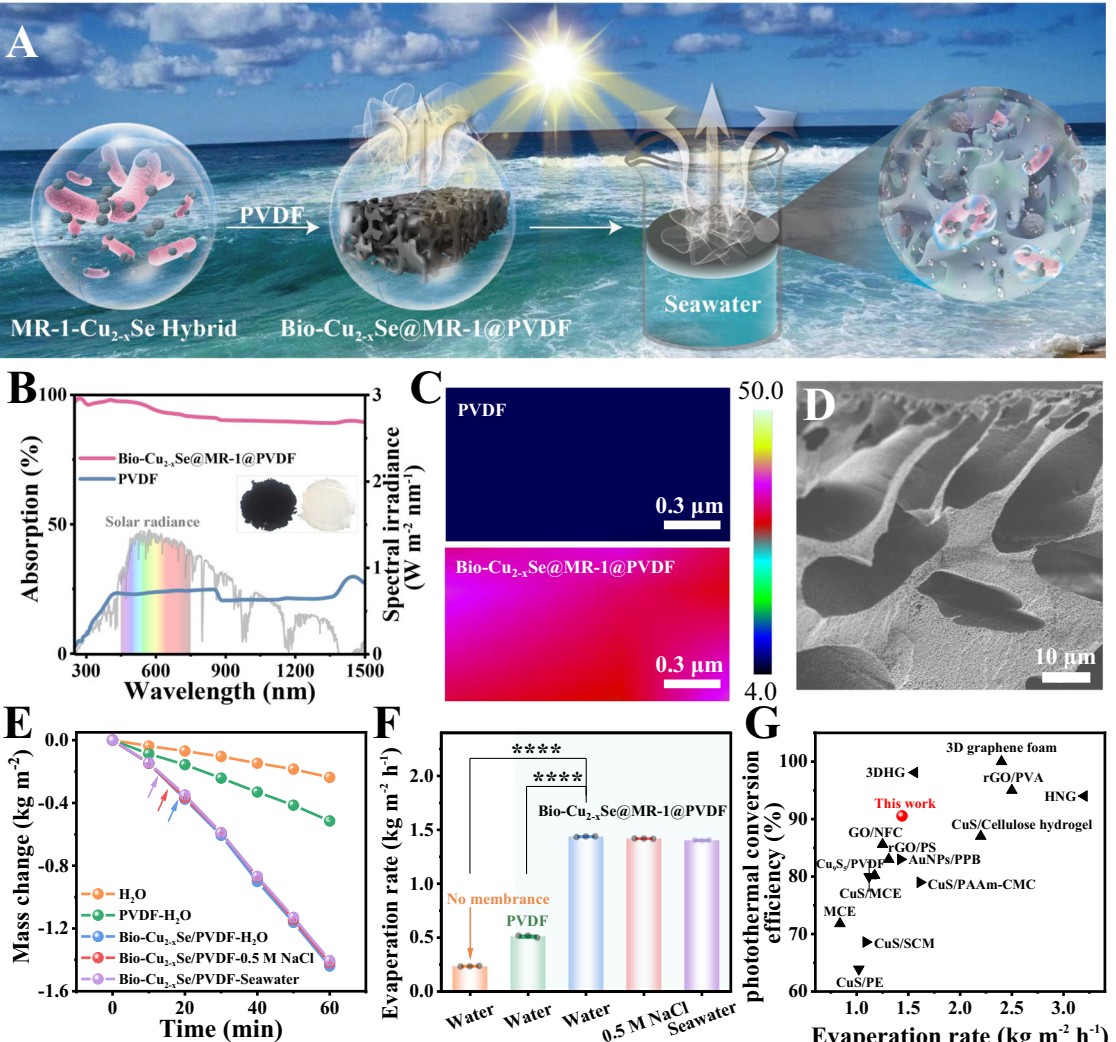

**Fig. 5 | Physicochemical properties and corresponding solar vapor generation performance of the Bio-Cu$_{2-x}$Se@MR-1@PVDF and PVDF membranes.**
**A** Schematic diagram of the preparation of the Bio-Cu$_{2-x}$Se@MR-1@PVDF membrane for solar vapor generation. **B** Solar spectral irradiance (gray, left-hand side axis) and absorption (black, right-hand side axis) and the optical pictures (inset of Fig. 5B. left: Bio-Cu$_{2-x}$Se@MR-1@PVDF membrane. right: PVDF membrane).
**C** Raman images of Cu-Se bond at 260 cm$^{-1}$ for PVDF membrane (top) and Bio-Cu$_{2-x}$Se@MR-1@PVDF membrane (down). **D** SEM image of the cross-section of Bio-Cu$_{2-x}$Se@MR-1@PVDF membrane. **E** Water evaporation induced mass loss curves.
**F** Evaporation rate of membranes under 1 sun irradiation. **G** Comparison of the

photothermal conversion efficiency and evaporation rate of Bio-Cu$_{2-x}$Se@MR-1@PVDF membrane and the reported membranes from Supplementary Table 3-4. The data points represented in (**E**, **F**) represent three ($n = 3$) independent experiments for each experimental group and are displayed as mean ± standard deviation (SD). $p$ values of (**F**) were determined by a one-way analysis of variance. "****" represents $p < 0.0001$. Experiments of (**D**) were repeated three times with similar results. Source data and frequency inference statistics (the value of degrees of freedom, $p$, effect size statistic, 95% confidence intervals) are provided as a Source Data file.

(PDF#06-0680) (Supplementary Fig. 12C). According to the XPS result (Supplementary Fig. 12D), the molar percentages of Cu(I) and Cu(II) in Chem-Cu$_{2-x}$Se NPs were calculated to be 90.69% and 9.41%, respectively. These results confirmed the successful synthesis of Chem-Cu$_{2-x}$Se NPs.

We further investigated and compared the performances of Bio-Cu$_{2-x}$Se@MR-1@PVDF and membrane that only mixed *S. oneidensis* cells with Chem-Cu$_{2-x}$Se NPs (denoted as Chem-Cu$_{2-x}$Se@MR-1@PVDF). Adding *S. oneidensis* cells to Chem-Cu$_{2-x}$Se@PVDF membrane decreased the contact angle of the membrane surface (Supplementary Fig. 13A), suggesting that the hydrophilicity of the membrane was improved by biomass. Meanwhile, Bio-Cu$_{2-x}$Se@MR-1@PVDF owned the best hydrophilicity (Supplementary Fig. 13A), which was mainly attributed to the combined effects of natural capping proteins on Cu$_{2-x}$Se NPs and the inherent hydrophilicity of biomass. Supplementary Fig. 13B shows the average water evaporation rates of the

membranes under 1 sun irradiation, which followed this order: Bio-Cu$_{2-x}$Se@MR-1@PVDF > Chem-Cu$_{2-x}$Se@MR-1@PVDF > Chem-Cu$_{2-x}$Se@PVDF. Consistently, the highest solar evaporation efficiency was obtained in Bio-Cu$_{2-x}$Se@MR-1@PVDF (Supplementary Fig. 13C).

Impressively, after the photothermal evaporation experiment, the color of Chem-Cu$_{2-x}$Se@MR-1@PVDF and Chem-Cu$_{2-x}$Se@PVDF membranes changed from deep black to dark green and green (Supplementary Fig. 14A), respectively, indicating air oxidation of copper in the membrane[36]. In contrast, Bio-Cu$_{2-x}$Se@MR-1@PVDF membrane maintained deep black. Subsequently, the structural, chemical, and performance stabilities of the evaporator membrane were investigated by monitoring copper content in the solution and the valence states of Cu in the membrane before and after 1 sun irradiation for 10 h. After 10 h of treatment, part of Chem-Cu$_{2-x}$Se NPs detached from the supporting material, which was supported by the enhanced copper content in the solution (Supplementary Fig. 14B). XPS of Chem-

$Cu_{2-x}Se@PVDF$ and $Chem-Cu_{2-x}Se@MR-1@PVDF$ membranes showed that the energy spectra of Cu $2p_{3/2}$ and Cu $2p_{1/2}$ shifted to higher binding energy after 10 h of irradiation (Supplementary Fig. 15A-15D), indicating that air oxidation led to an increase in the Cu(II)/Cu(I) ratio.

The evaporation performance of $Chem-Cu_{2-x}Se$ NPs decreased after the evaporator was reused for 10 cycles (Supplementary Fig. 16B, C). Encouragingly, $Bio-Cu_{2-x}Se$ shedding from the membrane was negligible, exhibiting excellent structural stability (Supplementary Fig. 14B). According to XPS spectra, the binding energies of Cu $2p$ in $Bio-Cu_{2-x}Se@MR-1@PVDF$ remained similar after 10 h of irradiation (Supplementary Fig. 15E, F and Supplementary Fig. 16A), which proves that $Bio-Cu_{2-x}Se@MR-1@PVDF$ has excellent chemical stability. $Bio-Cu_{2-x}Se@MR-1@PVDF$ was reused 10 times for the evaporation test under 1 sun irradiation. The corresponding evaporation performance was relatively stable (Supplementary Fig. 16B, C), exhibiting good reusability and durability. The high stability implies that biomass might act as a natural linker, enhancing the adhesion between the photothermal conversion material and the supporting material. Meanwhile, the contents of TOC and protein in the solution were negligible, ruling out potential hazards from the degradation of MR-1 dead cells (Supplementary Fig. 16D). Together, $Bio-Cu_{2-x}Se@MR-1@PVDF$ possesses the best evaporation performance among them, ensuring long-term stable solar steam generation (Supplementary Fig. 16B, C). Such excellent solar evaporation performance should be assigned to the good photothermal conversion efficiency of $Cu_{2-x}Se$, the high hydrophilicity of biomass, and the superior stability of $Bio-Cu_{2-x}Se@MR-1@PVDF$.

To further explore whether the high efficiency of $Bio-Cu_{2-x}Se@MR-1@PVDF$ membrane was attributed to the biogenic $Cu_{2-x}Se$ NPs, we employed $\Delta cymA$ to synthesize $Cu_{2-x}Se$ NPs through Supplementary Method 13, which were then encapsulated in a PVDF substrate to fabricate a solar steam generation device ($\Delta cymA$-NPs@MR-1@PVDF). As expected, destroying the CymA protein synthesis severely inhibited $Cu_{2-x}Se$ NPs production under dark conditions (Supplementary Fig. 17A), resulting in a substantial decrease in photothermal performance (Supplementary Fig. 17B, C). The solar evaporation rate of $\Delta cymA$-NPs@MR-1@PVDF was only 0.89 kg m$^{-2}$ h$^{-1}$, which was much lower than that of $Bio-Cu_{2-x}Se@MR-1@PVDF$ (using WT as bio-nano-factory) (Supplementary Fig. 17B). In comparison, the solar evaporation efficiency of $Bio-Cu_{2-x}Se@MR-1@PVDF$ membrane was 1.73 times that of $\Delta cymA$-NPs@MR-1@PVDF. Thus, the high efficiency of $Bio-Cu_{2-x}Se@MR-1@PVDF$ membrane was tightly associated with the amount of $Cu_{2-x}Se$ NPs.

To uncover the working mechanism of $Bio-Cu_{2-x}Se@MR-1@PVDF$ membrane in the water desalination process, we detected the UV-vis DRS, photoactivity, and UV-vis-NIR spectra of the biogenic $Cu_{2-x}Se$ NPs (Supplementary Method 14). UV-NIS-DRS spectrum displays that biogenic $Cu_{2-x}Se$ NPs exhibit strong light absorption in the wavelength range of 250–2500 nm, which forms the basis for good performance in photothermal distillation (Supplementary Fig. 18A). The photoactivity of the purified $Cu_{2-x}Se$ NPs was evaluated by measuring the photocurrent in the presence and absence of light. An observable photocurrent was produced when the nanoparticles were exposed to light, and it returned to baseline levels after the light was turned off (Supplementary Fig. 18B). No photocurrent was detected in the control group without nanoparticles. These results suggest that biogenic $Cu_{2-x}Se$ NPs act as semiconductors and can generate electron-hole pairs upon exposure to light. Recent studies have found that the free carrier (hole) generated in $Cu_{2-x}Se$ semiconductors can drive thermalization process, resulting in the generation of photogenerated heat. Meanwhile, the free carrier is the response for NIR plasmonic absorption[37]. Thus, we tested the LSPR property of the biogenic $Cu_{2-x}Se$ semiconductor by detecting the UV-vis-NIR spectrum of the purified $Cu_{2-x}Se$ NPs. As expected, we observed apparent absorption in the near-infrared region of the purified $Cu_{2-x}Se$ NPs (Supplementary Fig. 18C). Overall, biogenic $Cu_{2-x}Se$ NPs exhibit broad solar energy absorption and LSPR properties, making $Bio-Cu_{2-x}Se@MR-1@PVDF$ membrane well-suited for water desalination.

Accordingly, a schematic diagram of the working mechanism of $Bio-Cu_{2-x}Se@MR-1@PVDF$ membrane in the water desalination process is plotted in Supplementary Fig. 19. Once the light is turned on, $Cu_{2-x}Se$ NPs are excited and generate electron-hole pairs. Meanwhile, $Cu_{2-x}Se$ NPs have local surface plasmon resonance (LSPR) properties, in which free carriers (holes) can collectively resonate with incident photons and produce hot electrons. By incorporating $Cu_{2-x}Se$ NPs and biomass into the PVDF membrane, $Bio-Cu_{2-x}Se@MR-1@PVDF$ can achieve an apparent increase in surface temperature. Simultaneously, the hydrophilic nature of bacterial cells and the porous structure of PVDF allow efficient water transport to the interface, allowing water to be efficiently transported to the thermally located surface. The excellent evaporation performance should be assigned to the synergistic effect between the good photothermal conversion efficiency of $Cu_{2-x}Se$ NPs, the high hydrophilicity of biomass, the presence of microchannel in PVDF membrane, and the superior stability of $Bio-Cu_{2-x}Se@MR-1@PVDF$.

## Discussion

Indeed, the biological synthesis of Se$^0$ NPs has been realized by various microorganisms (Supplementary Table 6), which mainly focused on synthetic mechanisms and regulatory strategies. However, the in vivo application of biogenic Se$^0$ NPs, that couple the functions of bacteria and nanoparticles, has yet to be reported. Motivated by the photocatalytic function of Se$^0$ NPs, the present study constructed *S. oneidensis*-Se$^0$ hybrid for photo-boosted biological functions. *S. oneidensis* MR-1 possesses powerful nanoparticle assembly capability and diverse respiratory reductases, making it an excellent candidate for exploring *S. oneidensis*-Se$^0$ hybrid system. This work reveals the function of Se$^0$ NPs in terms of accelerating bacterial functions through reversing MtrABC, which is initiated by light illumination. With mutant analyses, we uncover the underlying synergistic mechanisms and elucidate the corresponding electronic circuits.

Recently, we demonstrated the feasibility of extracellular biosynthesis of $Cu_{2-x}Se$ NPs by *S. onedensis* (Supplementary Table 7)[27,38]. Such a biological process can be alleviated by adding anthraquinone-2,6-disulfonate (AQDS)[38], which promotes the transport of metabolic electrons from the outer membrane to the electron acceptor. Meanwhile, the composition and photothermal properties of $Cu_{2-x}Se$ NPs can be regulated by adjusting the precursor concentration[39]. These biogenic processes were found to be driven by metabolic electrons (Supplementary Table 7). In this work, under visible light illumination, periplasmic Cu(II) reduction is initiated by two electron fluxes, either intracellular metabolic electrons through CymA or extracellular photoelectron via reversed MtrABC. Subsequently, the reaction between intermediate Cu(I) and Se$^0$ NPs was facilitated by light irradiation. This photo-promoted biotransformation of $Cu_{2-x}Se$ NPs by *S. oneidensis*-Se$^0$ hybrid is simple and sustainable. Regarding the $Cu_{2-x}Se$ NPs formation site, a previous study showed that $Cu_{2-x}Se$ NPs should be predominantly formed in the cell periplasm and then excreted out[38]. Differently, we find that $Cu_{2-x}Se$ NPs are assembled extracellularly through a reaction between efferent Cu(I) and extracellular Se$^0$ NPs. Such different formation sites may be attributed to the different selenium precursors. The former used sodium selenite as a selenium source related to periplasmic selenite reduction. In contrast, we used Se$^0$ NPs, which are predominately located in extracellular space. In terms of the working direction of EET, previous studies have shown that EET worked normally in the presence of AQDS to deliver intracellular electrons for $Cu_{2-x}Se$ production. We find that under light irradiation, Mtr cytochrome conduit worked in a reversed direction to deliver extracellular photoelectrons from Se$^0$ NPs to periplasmic Cu(II) reduction for subsequent extracellular $Cu_{2-x}Se$ NPs assembly.

Electroactive microorganisms own a unique bidirectional EET between electrically active redox compounds and microorganisms. *S. oneidensis* MR-1 is a particularly well-known model organism with the capacity to deliver internal electrons to the outer surface through the EET pathway consisting of CymA and Mtr cytochrome conduit. Among them, the most striking electron corridor for trafficking periplasmic electrons across the outer membrane is composed of inner-facing cytochrome MtrA, porin protein MtrB, and outer environment-facing cytochromes MtrC and OmcA. Such transmembrane electron conduit has mainly been explored in the context of its native activity, which involves of the efflux of internal electrons to a more reduced external electron acceptor, such as metal oxides and anode electrodes. Interestingly, the direction of the Mtr electron circuit can be reversed to acquire electrons from cathodes and semiconductors to a more oxidized terminal electron acceptor, such as fumarate, oxygen or nitrate (Supplementary Table 8). Although it has previously been demonstrated that photoelectron uptake from extracellular semiconductors to *S. oneidensis* MR-1 for periplasmic $H_2$ generation via a reverse function of the Mtr pathway[16,18], the study reported here couples the reverse Mtr pathway and exterior photocatalysis for nanoparticles assembly. This is a groundbreaking advancement for utilizing inward photoelectrons for biomineralization.

Recently, photoelectrons, cathode electrons, and chemical electrons have been injected into cells to boost the biological process for renewable energy recovery, value-added chemical production, and wastewater treatment (Supplementary Table 8). The function and direction of EET depend not only on the type and location of the upstream electron source but also on the site of the downstream electron acceptor. If the electron acceptor is in extracellular space, metabolic electrons flow out of cells via natural EET[40]. Alternatively, if the electron sink reacts in periplasm or cytoplasm, an inward electron flow is observed[18]. In our system, the electron acceptor Cu(II) is located in the periplasm, resulting in metabolic electrons from the cytoplasm to the periplasm through CymA. At the same time, the Mtr pathway is not involved in this biocatalytic process. Under light illumination, the external $Se^0$ semiconductor can be excited to generate photoelectrons. Mutant experiments and redox potential analysis suggested that a reverse MtrABC pathway was involved in the inward electron flow from extracellular $Se^0$ NPs to cells. The reversal of the MtrABC pathway was further validated by nitrate reduction with acetic acid as an electron donor.

To date, the use of solar energy for the fabrication of nanoparticles has been demonstrated in several pure organisms (Supplementary Table 9). The proposed underlying mechanisms rely on inherently photosensitive groups, implying that such light-induced nanoparticle formation is generally restricted in a particular organism. Furthermore, the detailed mechanisms of related proteins and extracellular polymeric substances remain to be further investigated. Unlike the intrinsic light-use capabilities of specific organisms, in this work, non-photosynthetic bacteria are endowed with advanced light-energy utilization capabilities by in vivo synthesizing $Se^0$ semiconductor, creating an unnatural photoelectronic pathway for $Cu_{2-x}Se$ NPs formation. This hybrid system provides a convenient and universally applicable approach to direct solar energy toward nanoparticle synthesis, and beyond the natural capabilities of living organisms. Undoubtedly, the constructed *S. oneidensis*-$Se^0$ hybrid is a promising and efficient solar-to-chemical technology, that reasonably combines the light-harvesting of inorganic semiconductors with the specific catalytic power of organisms. More importantly, $Cu_{2-x}Se$ NPs biosynthesis is just a model of photo-driven bio-hybrid systems. Such an *S. oneidensis*-$Se^0$ hybrid can be extended to nitrate reduction (Supplementary Fig. 8) and HgSe synthesis (Supplementary Fig. 9), demonstrating its broad applicability. Such functional synergy could not only be important for the biosynthesis of living materials but also contribute to good strategies for solar-to-high-value products.

Biotic-abiotic photosynthetic systems hold great promise to innovate solar-driven chemical transformation. Nevertheless, balancing the generation and utilization of photoelectrons, illustrating the fate of photoelectrons, and diversifying solar energy transduction products are still challenges. Here, we select an electro-active bacterium *S. oneidensis* MR-1 as a model for hybrid photosynthetic system construction, attracted by its multimodal interfacial photoelectrons utilizing channels and inherent extracellular assembly of multifunctional nanoparticles. To the best of our knowledge, biogenic $Se^0$ NPs were first selected as a photosensitizer to construct a biohybrid system (Supplementary Table 10), because of their excellent biocompatibility and appropriate band position. Compared to the single site and single pathway of photoelectron utilization in the reported works (Supplementary Table 10), photoelectrons generated by extracellular $Se^0$ NPs wirelessly activate $Cu_{2-x}Se$ NPs synthesis through a dual catalytic network located in periplasm and extracellular space, respectively. Such a dual catalytic network is a paradigm for balancing the source and sink of photoelectrons. Besides, we expand solar-to-chemical production from organic substances and hydrogen to nanoparticles (Supplementary Table 10), diversifying solar energy conversion products in biotic-abiotic hybrid platforms. Our work possesses several groundbreaking findings that may fundamentally change our perception of biotic-abiotic photocatalytic systems and may bring biotic-abiotic photocatalysis a step forward toward application.

This work improved the sustainability level of the biohybrid system, including the establishment of *S. oneidensis*-$Se^0$ hybrid at the primary level, photo-boosted heavy metal biotransformation at the secondary level, and solar water production at the tertiary level. For solar water production, photothermal conversion material is the most critical factor in determining efficiency. Commonly used photothermal materials include carbon-based materials, noble metal nanomaterials, ceramic-based materials, and plasmonic semiconductors (Supplementary Table 4). Although the extensively used noble metal-based evaporator has achieved an appreciable water evaporation efficiency, it is generally restricted by high cost and easy fusion at high temperatures[41]. The main demerit of graphene-based evaporators is their high price, limiting their large-scale production and practical applications. Recently, Cu-composed materials have been considered as the most promising photothermal materials, owing to their relatively low cost, non-toxicity, eco-friendliness, and broadband light absorption. To date, various Cu-composed materials have been employed to construct solar steam generators and have achieved encouraging performance (Supplementary Table 5). Most of the reported works have focused on traditional chemical synthesis methods, involving aggressive chemical agents, harsh synthesis conditions, intensive energy consumption, and complex synthesis processes (Supplementary Table 5). Besides, the anchored photothermal material can detach from the supporting material during long-term irradiation or high-frequency use, resulting in poor durability[42]. Here, the as-prepared Bio-$Cu_{2-x}Se$@MR-1@PVDF was prepared at room temperature using environmentally benign reagents, providing a more sustainable manufacturing strategy. Additionally, biomass, as a natural linker, enhances the adhesion between the photothermal conversion material and the supporting material. These membranes exhibit good structural, chemical and performance stability. Compared with the typical photothermal conversion materials (Supplementary Tables 4 and 5), the water evaporation conversion efficiency of the resulting membrane is top-ranking. Thus, the Bio-$Cu_{2-x}Se$@MR-1@PVDF has the advantages of being inexpensive, sustainable, simple to fabricate, highly stable, and highly efficient in water evaporation, meeting the practical requirements for solar vapor generation.

In summary, we designed a whole-cell biotic-abiotic hybrid system for $Cu_{2-x}Se$ photocatalytic synthesis and solar water production by integrating the merits of *S. oneidensis* and biogenic nanoparticles. The constructed *S. oneidensis*-$Se^0$ NPs hybrid endows bacterium with the capacity to capture light toward $Cu_{2-x}Se$ NPs synthesis. Intriguingly, we

unveil two catalytic networks that were activated by photoelectrons, ensuring efficient Cu$_{2-x}$Se NPs synthesis, including periplasmic Cu(II) reduction and extracellular Cu(I) conversion. Moreover, the resulting Cu$_{2-x}$Se NPs were applied for photothermal device fabrication, displaying an excellent solar evaporation rate (1.44 kg m$^{-2}$ h$^{-1}$) and a high solar thermal efficiency (90.55%) under 1 sun illumination. Ultimately, the Bio-Cu$_{2-x}$Se@MR-1@PVDF device was successfully employed for seawater desalination. This work provides a good model for a comprehensive understanding of photoelectron transport during hybrid photocatalysis and expands biotic-abiotic hybrid platforms for efficient solar energy harvesting.

## Methods

### S. oneidensis-Se$^0$ NPs hybrid construction

*S. oneidensis* MR-1 wide-type (WT) and several mutants deficient in specific membrane-anchored cytochromes (Δ*cym*A, Δ*mtr*F and Δ*omc*AΔ*mtr*C) were kindly provided by Prof. Shi of China University of Geosciences[43] and Prof. Yong of Jiangsu University[1]. *S. oneidensis*-Se$^0$ NPs hybrid was constructed following the procedure described in Supplementary Fig. 1A. The cryopreserved *S. oneidensis* MR-1 was activated by inoculating in Luria Agar (LA) plate, and aerobically incubated for 48 h at 30 °C. An individual colony was picked from LA and grown in aerobic Luria-Broth (LB) medium overnight. Then the cultures were enlarged by transferring to fresh aerobic LB medium at a ratio of 1:10. After 12 h of incubation, the stationary cells were collected by centrifugation (3743 g, 10 min, 4 °C) and washed three times with LB medium. The collected cells were resuspended in serum bottles containing anaerobic LB medium and 0.5 mM Na$_2$SeO$_3$. The concentration of the resuspended cells was adjusted to an optical density of 1.0 at 600 nm. The anaerobic condition was prepared by purging with 100% N$_2$ for 30 min. After 10 h of incubation under anaerobic conditions, *S. oneidensis*-Se$^0$ NPs hybrid was obtained. All the incubation conditions were conducted at 30 °C and 200 rpm sharking rate.

### Photocatalytic synthesis of Cu$_{2-x}$Se NPs by S. oneidensis-Se$^0$ NPs hybrid

For Cu$_{2-x}$Se NPs photocatalytic synthesis, the constructed *S. oneidensis*-Se$^0$ NPs were collected by centrifugation (3743 g, 10 min, 4 °C) and washed three times with an inorganic salt medium (SMB). The collected *S. oneidensis*-Se$^0$ NPs were resuspended in the anaerobic tube containing SMB medium and 1 mM EDTA-Cu(II). The concentration of resuspended cells was adjusted to an optical density of 1.0 at 600 nm. Under light illumination, Cu$_{2-x}$Se NPs photocatalytic synthesis was initiated. The used illumination source was a collimated Xenon lamp (PLS-SXE300D, Beijing Pophile Technology Co., Ltd., China) with a 420 nm UV-cut filter. The SMB contains (per liter) 1.49 g NH$_4$Cl, 0.09 g KCl, 0.67 g NaH$_2$PO$_4$·2H$_2$O, 0.0002 g flavin, 11.91 g 4-(2-hydroxyethyl)-1-piperazineethanesulfonic acid (HEPES), 29.41 g sodium citrate. In SMB medium, the carbon source was 20 mM sodium lactate or 20 mM sodium acetate. The pH of the SMB was adjusted to 7.2. The anaerobic condition was prepared by purging with 100% N$_2$ for 30 min and sealed with rubber. Before use, Cu(II)-EDTA was added to a final concentration 1 mM. To ensure the complete chelation between EDTA and Cu(II), we pre-paraded the stock solutions of EDTA-Cu chelate solution via the reaction of CuSO$_4$·5H$_2$O (2.5 mol) and ethylenediaminetetraacetic acid disodium salt (EDTA-Na$_2$, 2.5 mol) in 100 mL of deionized water. For Cu$_{2-x}$Se NPs synthesis, the final concentration of EDTA-Cu(II) was 1 mM. The samples of the experiments in Fig. 2B were from three independent biological replicates while Fig. 2D from six. For illumination experiments, the control groups were wrapped in tin foil.

### Fabrication of membranes

Cu(II)-EDTA was directly added to *S. oneidensis*-Se$^0$ NPs anaerobic system in LB and synthesized at 30 °C and 200 rpm for 24 h. Centrifuged, collected and washed *S. oneidensis*-Cu$_{2-x}$Se. Then the membrane making material was obtained by freeze-drying and grinding. The membranes were prepared using phase inversion approach[27]. The casting solution was prepared by mixing polyvinyl pyrrolidone (PVP) and N,N-dimethylformamide (DMF) according to Supplementary Table 3. For Bio-Cu$_{2-x}$Se@MR-1@PVDF membrane, the casting solution was prepared by dissolving 1 g PVP into 75 g DMF. Subsequently, 10 g freeze-dried Bio-Cu$_{2-x}$Se NPs was added into the casting solution and then ultrasonic dispersion. 15 g PVDF was then added to the casting solution and stirred at room temperature to get a uniform mixture solution. The obtained mixture was placed in a vacuum drying (30 °C) to remove bubbles. After 30 min, the mixture was spread onto a clean and dry glass plate. Then use a wet film coater four sided film applicator to obtain a membrane with a thickness of 200 μm. 1 minute later, the glass was immersed in deionized water for 30 min to detach the membrane from the glass. The as-prepared membranes were cut into 4 cm-diameter circles and stored in pure water at 4 °C before use. For control PVDF membrane preparation, the casting solution was prepared by dissolving 1 g PVP into 85 g DMF. Subsequently, 15 g PVDF was added to the casting solution and stirred for at least 24 h.

For comparison, control membranes (Chem-Cu$_{2-x}$Se@MR-1@PVDF, Chem-Cu$_{2-x}$Se@PVDF) were prepared using the same process as Bio-Cu$_{2-x}$Se@MR-1@PVDF membrane. Except for that, Bio-Cu$_{2-x}$Se@MR-1 was replaced by mixing *S. oneidensis* cells with Chem-Cu$_{2-x}$Se NPs. The detailed parameters are listed in Supplementary Table 3.

### Solar evaporation performance of membranes

To evaluate the solar evaporation performance of membranes, a desktop Xenon lamp (PLS-SXE300D, Beijing Pophile Technology Co., Ltd, China) with 1 sunlight intensity and an infrared camera (ICI7320, Infrared Camera Inc., Beaumont, Texas, USA) were utilized to measure the surface temperature of the membrane in the photothermal distillation membrane experiment. The water evaporation rate and photothermal conversion efficiency of each membrane were calculated using the following equation[44]:

$$m = \frac{\Delta m}{S \times t} \quad (1)$$

$$\eta = m' \times h_{Lv}/(3600 \times P_{in}) \quad (2)$$

where $m$ is the water evaporation rate (kg m$^{-2}$ h$^{-1}$), $\eta$ is the light-to-heat conversion rate (%), $\Delta m$ is the mass change of water in 1 h (kg), and $S$ is the membrane evaporation area (m$^2$), $t$ is the irradiation time (h), $m'$ is the corresponding water evaporation rate after subtracting dark evaporation rate (kg m$^{-2}$ h$^{-1}$), $h_{Lv}$ represents the liquid-to-gas phase enthalpy change of water (kJ kg$^{-1}$, ambient temperature = 20 °C), whereas $P_{in}$ represents the incident light power (i.e., 1 kW m$^{-2}$).

### Reporting summary

Further information on research design is available in the Nature Portfolio Reporting Summary linked to this article.

## Data availability

Data supporting the findings of this work are available within the paper and its Supplementary Information files. A reporting summary for this Article is available as a Supplementary Information file. Source data are provided with this paper.

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

## Acknowledgements
This work was supported by the National Natural Science Foundation of China (51821006 and 21907087), the Natural Science Foundation of Anhui Province (1908085MB31), Fundamental Research Funds for the Central Universities and USTC Research Funds of the Double First-Class Initiative. The authors thank Prof. Liang Shi at the China University of Geosciences and Prof. Yang-Chun Yong at Jiangsu University for providing *Shewanella oneidensis* strains.

## Author contributions
S.-L.G. conducted the experiments, L.-J.T. and G.-P.S. conceived the idea. S.-L.G., L.-J.T., Y.-C.T. and G.-P.S. wrote the paper. All authors discussed the results and commented on the manuscript.

## Competing interests
The authors declare no competing interests.
