## [Peer Review File · Nature Communications]

Dual-mode harvest solar energy for photothermal Cu₂-xSe biomineralization and seawater desalination by biotic-abiotic hybridReviewers' Comments:

Reviewer #1:

Remarks to the Author:

In this manuscript, the authors constructed a biotic-abiotic hybrid system composed of *Shewanella oneidensis* MR-1 and biogenic Cu₂-xSe nanoparticles. It's impressive that the authors used multiple characterization techniques to elucidate the pathways for photoelectron utilization and the biogenic mechanisms of Cu₂-xSe nanoparticles. Finally, this hybrid material was incorporated into a solar steam device for seawater desalination application. This work is original and offers a new perspective in the field of photo-driven bio-hybrid systems.

Specific comments:

1. The authors' use of biological terms such as "photosynthesis" and "photosynthetic" to describe the hybrid system consisting of non-photosynthetic cells and nanoparticles is not appropriate. Instead, it can be described as "photocatalytic system" "photocatalytic synthesis" or other suitable phrases.
2. Supplementary Table 4 provides a performance comparison of various photothermal materials in solar vapor generation. It would be beneficial to provide additional comparisons and discussions to highlight the advantages and potential disadvantages of Cu₂-xSe@MR-1@PVDF membrane in terms of performance, stability, scalability, cost, and other relevant factors.
3. There are two photographs in Supplementary Figure 1B that depict the mixture of cells and bio-Cu₂-xSe in. What are the difference between the two photographs?
4. The excellent performance of the Cu₂-xSe@MR-1@PVDF membrane is attributed to the combined effect of the photothermal property of Cu₂-xSe nanoparticles and the hydrophilicity of *Shewanella* cells. Is it possible to obtain this hybrid material simply by mixing *Shewanella* cells with chemically synthesized Cu₂-xSe nanoparticles? A comparison between the performance of these two hybrid materials (biogenic Cu₂-xSe nanoparticles with *Shewanella* cells vs. chemically prepared Cu₂-xSe nanoparticles) is anticipated.
5. In Line 180-181 and 247-248, the authors mention that lactate acts as the sacrificial electron donor for Se₀ nanoparticles photocatalyst. It is known that electrons derived from lactate are typically released in the intracellular space through oxidation reactions. This implies that the extracellular electron transfer (EET) pathway, consisting of MtrCAB, should function in an outward direction to provide electrons for extracellular Se₀ nanoparticles. Therefore, there appears to be a conflict between the outward EET pathway and the inward/reversed EET pathway, and the latter is proposed to transport photoelectrons generated by extracellular Se₀ nanoparticles in Figure 4. Please clarify this conflict.
6. Line 301, it should refer to Supplementary Fig. 7H.
7. Line 427, please give more experimental details about fabrication of Cu₂-xSe@MR-1@PVDF membrane.
8. Please give the complete names of the abbreviations used in Supplementary Table 3 and Supplementary Table 4.

Reviewer #2:

Remarks to the Author:

20231026-Review-Nature Communications

Overview:

The paper entitled "Dual-Mode Harvest Solar Energy for Photothermal Cu₂-xSe Biomineralization and Seawater Desalination by Biotic-Abiotic Hybrid" presents an approach for utilizing photoelectrons for both Cu₂-xSe nanoparticles (NPs) biosynthesis and then for water desalination. To achieve the goals, the authors conducted serial experimental steps. First, the authors synthesized Se₀ NPs from SeO₃²⁻ by utilizing the metal-reducing ability *Shewanella oneidensis* MR-1. Second, Cu₂-xSe NPs were synthesized driven by the photoelectrons generated by illuminating Se₀ NPs via dual pathways: activating the periplasmic Cu(II) reduction network and the direct production of Cu (I). Third, the Cu₂-xSe@MR-1@PVDF membrane was prepared and tested for its performance in water desalination. The authors conducted various analyses to verify the mechanisms. However, based on my comments and questions below on the novelty, the publish-worthiness of this work on Nature Communication should be carefully considered.

Comments:

1. Biosynthesis of Se₀ NPs using *Shewanella oneidensis* MR-1 or other bacteria is well documented (please refer to 10.1021/jacs.7b07460; 10.1038/srep03735; 10.1016/j.procbio.2023.05.016; 10.1271/bbb.90454; 10.1007/s11274-022-03374-6; 10.1016/j.jhazmat.2016.02.035; 10.1016/j.saa.2017.11.050; etc.)
2. Cu₂-xSe NPs biosynthesis by *Shewanella oneidensis* MR-1 or other bacteria is also reported before (please refer to 10.1021/acsami.3c03611; 10.1021/acs.est.2c04130; 10.1016/j.materresbull.2018.11.014; etc.). I understand two new findings in this manuscript: utilizing photoelectrons for biotic and abiotic production of Cu₂-xSe and quantifying each pathway. The finding on reversing electron transfer chain on *Shewanella oneidensis* MR-1 utilizing photoelectrons was also reported before 10.1021/jacs.2c00934.
3. Using Cu₂-xSe NPs containing MR-1 cells to fabricate Cu₂-xSe@MR-1@PVDF membrane. The photothermal property of Cu₂-xSe enhanced water evaporation in the desalination process. However, this application is not very new (please refer to 10.1016/j.jcis.2022.06.028; 10.1002/cssc.202201543; 10.1007/s10853-022-07353-y). Moreover, I worry about the MR-1 dead cell degradation and air oxidation of Cu in Cu₂-xSe NPs during the use of the fabricated membrane.

Questions:

1. Line 119-120: the authors wrote: "acetic acid as an electron sacrifice agent." I could not find the word "acetic acid" in the Methodology section of Cu₂-xSe NPs synthesis. Moreover, can MR-1 cannot use acetate as an electron donor. Did the authors use lactate?
2. The authors sometimes used UV-vis-NIR (Fig.2), and sometimes used UV-vis-DRS (Fig. 1). However, only UV-vis-DRS was described in the Methodology section (Line 362 onward).
3. Can the authors calculate the amount of Cu₂-xSe NPs produced from 1 mole lactate?
4. Line 201-202: "CymA protein is not a Cu(II) terminal reductase and may be upstream of Cu(II) terminal reductase". I suppose the idea is "CymA protein is not ONLY a Cu(II) terminal reductase." Am I correct?
5. About Cu₂-xSe@MR-1@PVDF membrane:
 - Please draw a diagram to explain the working mechanism of this membrane in the water desalination process (can be included in the Supporting Information section)
 - Is it possible to add control without using membrane condition to Fig 5E-F without using membrane condition?
 - The author used WHO standard to compare the quality of treated water Supplementary Figure 7H. If producing drinking water is a target, I worry about the MR-1 dead cells' decomposition during the use of this membrane.
 - In the membrane preparation process and during the use of this membrane for water desalination, how about the air oxidation of Cu in Cu₂-xSe?. Please refer to 10.1021/acs.chemmater.8b03967.
6. Line 336-337: Can the author quantify the Se₀ NPs production rate?
7. Line 342 and Line 350: why did the Cu(II)-EDTA concentration need to be re-adjusted?. Is the total Cu(II)-EDTA concentration higher than 1 mM?
8. Line 358-361: Please add more information on Raman measurement conditions, including laser power, Raman shift range, exposure times, number of accumulations, etc.
9. Fig 5G: The comparison was made based on the information in Supplementary Table 4. Please

include 3 materials for a better comparison: 3DHG, 3D graphene foam, and CTH. Moreover, Fig 5G will be much better if the author included the Evaporation Rate ($\text{kg}\cdot\text{m}^{-2}\cdot\text{h}^{-1}$) on the second Y-axis.

10. Fig 1B: I believe Fig 1B is the same as Supplementary Figure 2B. Please re-check it.

11. Fig 3 B-G: please perform statistical analysis to compare conditions better.

12. Fig 4: Is the question mark in this Figure the limitation of this study? Can it be FccA protein? (Please refer to 10.1038/srep03735)

13. Fig 2: Please conduct XAS or XPS analysis of Cu_{2-x}Se NPs.

14. Supplementary Figure 3: whether Se_0 NPs in this study are in t-Se or m-Se form?

Reviewer #3:

Remarks to the Author:

Please read the attached PDF file.

Response to Reviewer 1's comments

Recommendation: Reconsider after major revisions noted.

Comments:

In this manuscript, the authors constructed a biotic-abiotic hybrid system composed of *Shewanella oneidensis* MR-1 and biogenic Cu_{2-x}Se nanoparticles. It's impressive that the authors used multiple characterization techniques to elucidate the pathways for photoelectron utilization and the biogenic mechanisms of Cu_{2-x}Se nanoparticles. Finally, this hybrid material was incorporated into a solar steam device for seawater desalination application. This work is original and offers a new perspective in the field of photo-driven bio-hybrid systems.

Thanks for the positive comments and constructive suggestions.

Major points:

1. The authors' use of biological terms such as "photosynthesis" and "photosynthetic" to describe the hybrid system consisting of non-photosynthetic cells and nanoparticles is not appropriate. Instead, it can be described as "photocatalytic system" "photocatalytic synthesis" or other suitable phrases.

Thanks. Corrected as suggested.

2. Supplementary Table 4 provides a performance comparison of various photothermal materials in solar vapor generation. It would be beneficial to provide additional comparisons and discussions to highlight the advantages and potential disadvantages of Cu_{2-x}Se@MR-1@PVDF membrane in terms of performance, stability, scalability, cost, and other relevant factors.

Accepting the reviewer's constructive suggestion, we have updated the **Table and Discussion** sections to highlight the novelty of Cu_{2-x}Se@MR-1@PVDF membrane as follows:

"For solar water production, photothermal conversion material is the most critical factor in determining efficiency. Commonly used photothermal materials include carbon-based materials, noble metal nanomaterials, ceramic-based materials, and plasmonic semiconductors (Supplementary Table 4). Although the extensively used noble metal-based evaporator has achieved an appreciable water evaporation efficiency, it is generally restricted by high cost and easy fusion at high temperatures¹. The main demerit of the graphene-based evaporators is their high price, limiting their large-scale production and practical applications. Recently, Cu-composed materials have been considered as the most promising photothermal material, owing to their relatively low cost, non-toxic, eco-friendly and broadband light absorption. To date, various Cu-composed materials have been employed to construct solar steam generators and have achieved encouraging performance (Supplementary Table 5). Most of the reported works focus on traditional chemical synthesis methods, involving aggressive chemical agents, harsh synthesis conditions, intensive energy consumption, and complex synthesis processes (Supplementary Table 5). Besides, the anchored photothermal material can detach from the supporting material during long-term irradiation or high-frequency use, resulting in poor durability². Here, the as-prepared Bio-Cu_{2-x}Se@MR-1@PVDF was prepared at room temperature using environmentally benign reagents, providing a more sustainable manufacturing strategy. Additionally, biomass, as a natural linker, enhances the adhesion between the photothermal conversion material and the supporting material. Compared with the typical photothermal conversion materials (Supplementary Table 4 and 5), the water evaporation conversion efficiency of the resulting membrane is top-ranking. Thus, the Bio-Cu_{2-x}Se@MR-1@PVDF has the advantages of being inexpensive, sustainable, simple to fabricate, highly stable, and highly efficient in water evaporation, meeting the practical requirements for solar vapor generation." (P25-26, L528-551)

Supplementary Table 4. Comparison of solar steam device

Supporting Material	Absorber	Classification	Synthesis	Cost	Stability	Evaporation Rate (kg m ⁻² h ⁻¹)	Conversion Efficiency (%)	Ref.
poly	AuNPs	Plasmonic metal	Using MSA and TFA	high	Easily aggregated and fused together at high temperature	1.424	83	3
Paper substrate	Au NPs	Plasmonic metal	Using HAuCl ₄ , boil for 75 min	high	Easily aggregated and fused together at high temperature	-	77.8	4
PVA	RGO	Carbonaceous	Using H ₂ SO ₄ , KMnO ₄ , N ₂ H ₄ ·H ₂ O, glutaraldehyde	high	-	2.5	95	5
GO/NFC	CNT/GO	Carbonaceous	Using H ₂ SO ₄ , KMnO ₄ , NaClO, 80 °C	high	-	1.25	85.6	6
MCE	RGO	Carbonaceous	Using ascorbic acid, microwave reactor at 95 °C for 8 min	high	-	0.838	71.8	7
MGA	GO	Carbonaceous	Using H ₂ SO ₄ , KMnO ₄ , H ₂ O ₂ , 180 °C for 18 h	high	-	2.0	76.9	8
PS foam	RGO	Carbonaceous	Using H ₂ SO ₄ , KMnO ₄ , H ₂ O ₂ , TEOS, PS	high	A little salt can precipitate on the surface	1.31	83	9

GO foam	RGO	Carbonaceous	Using H ₂ SO ₄ , KMnO ₄ , laser	high	-	2.4	100	10
PS foam	CF	Carbonaceous	Using HNO ₃ at 80 °C for 8 h	low	-	1.56	98.1	11
HNG	PPy	Carbonaceous	Using PVA, pyrrole, glutaraldehyde, APS	high	-	3.2	94	12
CW	TiN NPs	Plasmonic ceramics	Using APTES	high	-	-	> 80	13
PVDFM	Cu _{2-x} Se	Plasmonic semiconductors	Photo-facilitated biosynthesis	low	Stability	1.44	90.55	This work

The above photothermal distillation membrane experiments were performed under 1 sun irradiation. Poly refers to p-phenylene benzobisoxazole. AuNPs refers to gold nanoparticle. MSA refers to methanesulfonic acid. TFA refers to trifluoroacetic acid. PVA refers to polyvinyl alcohol. GO refers to graphene oxide. NFC refers to nanofibrillated cellulose. CNT refers to carbon nanotube. MCE refers to mixed cellulose esters. RGO refers to reduced graphene oxide. MGA refers to modified graphene aerogel. PS refers to polystyrene. TEOS refers to tetraethyl orthosilicate. CF refers to carbon felt. HNG refers to hierarchically nanostructured gel. PPy refers to polypyrrole. PVA refers to polyvinyl alcohol. APS refers to ammonium persulfate. CW refers to ceramic fiber wool. TiN NPs refers to titanium nitride nanoparticles. APTES refers to 3-aminopropyl-triethoxysilane. PVDFM refers to the poly(vinylidene fluoride) membrane. 3DHG refers to 3D hierarchical solar vapor generator.

Supplementary Table 5. Comparison of solar steam device based on Cu-composed nanoparticles

Supporting Material	Absorber	Synthesis of Absorber	Synthesis Conditions of Absorber	Evaporation Rate (kg m ⁻² h ⁻¹)	Conversion Efficiency (%)	Membrane Temperature (°C)	Ref.
Filter paper	Cu _{2-x} Se@polydopamine	-	Using PVP, ascorbic acid, dopamine, H ₂ O ₂	2.71	-	-	14
Glass microfiber	Cu _{2-x} Se/Nb ₂ CT _x	-	Using hydrazine, PVP, H ₂ O ₂ , HF aqueous solution, over one week	1.2	-	39.7	15
PVDFM	Cu ₉ S ₅	Hydrothermal	180 °C for several hours	1.173	80.2 ± 0.6	36.1	16
PVDFM	CuS	Hydrothermal	180 °C for 18 h	1.43	90.4	38.5	1
MCE	CuS	Hydrothermal	140 °C for 12 h	1.12	80±2.5	42.8	17
SCM	CuS nanoflowers	Hydrothermal	120 °C for 18 h	1.09	68.6	35.2	2
Polyethylene	CuS	Hydrothermal	180 °C for 12 h	1.021	63.9	37.6	18
Cellulose hydrogel	CuS	Hydrothermal	120 °C for 12 h	2.2	87	-	19
PAAm-CMC	CuS	Hydrothermal	180 °C for 12 h	1.613	79	47.2	20
PVDFM	Cu _{2-x} Se	Photo-facilitated biosynthesis	Room temperature	1.44	90.55	41.2	This work

The above photothermal distillation membrane experiments were performed under 1 sun irradiation. PVDFM refers to the poly(vinylidene fluoride) membrane. PVP refers to poly(vinyl pyrrolidone). MCE refers to mixed cellulose ester membrane. SCM refers to semipermeable collodion membrane. PAAm-CMC refers to polyacrylamide (PAAm) and carboxymethyl cellulose (CMC).

3. There are two photographs in Supplementary Figure 1B that depict the mixture of cells and bio-Cu_{2-x}Se in. What are the difference between the two photographs?

Thanks. We apologize for the incorrect presentation in Figure S1B. In fact, the upper picture is a freeze-dried mixture. The obtained freeze-dried mixture was subsequently ground and shown in the bottom picture. **Figure S1** has been modified in the supporting information as follows:

Supplementary Figure 1. Schematic diagram of the experimental procedures. (A) Biosynthesis of Cu_{2-x}Se NPs with illumination by *S. oneidensis*-Se⁰ hybrid system. (B) Preparation of Bio-Cu_{2-x}Se@MR-1@PVDF membrane.

4. The excellent performance of the Cu_{2-x}Se@MR-1@PVDF membrane is attributed to the combined effect of the photothermal property of Cu_{2-x}Se nanoparticles and the hydrophilicity of *Shewanella* cells. Is it possible to obtain this hybrid material simply by mixing *Shewanella* cells with chemically synthesized Cu_{2-x}Se nanoparticles? A comparison between the performance of these two hybrid materials (biogenic Cu_{2-x}Se nanoparticles with *Shewanella* cells vs. chemically prepared Cu_{2-x}Se nanoparticles) is anticipated.

Accepting the reviewer's constructive suggestion, we have investigated and compared the solar vapor generation performance of **Bio-Cu_{2-x}Se@MR-1@PVDF** membrane with a membrane that simply by mixing *Shewanella* cells with chemically synthesized Cu_{2-x}Se nanoparticles (**Chem-Cu_{2-x}Se@MR-1@PVDF**). Major demerits of Chem-Cu_{2-x}Se@MR-1@PVDF are the structural, chemical and performance instability after 10 h irradiation, supported by detachment of Chem-Cu_{2-x}Se NPs, change in Cu valance and degradation of photothermal conversion (**Supplementary Figures 12-16**). Differently, Bio-Cu_{2-x}Se NPs exhibited excellent stability. Compared to Chem-Cu_{2-x}Se@MR-

1@PVDF, Bio-Cu_{2-x}Se NPs have good hydrophilicity, stability, and photothermal distillation performance.

A detailed description has been added into the revised manuscript as follows:

“To elucidate the underlying mechanism of the excellent performance of the Bio-Cu_{2-x}Se@MR-1@PVDF, chemically synthesized Cu_{2-x}Se nanoparticles (Chem-Cu_{2-x}Se NPs) were prepared according to the reported procedure²¹. TEM image reveals the average size of the Chem-Cu_{2-x}Se NPs was 84.3± 6.6 nm (Supplementary Fig. 12A-12B). The XRD pattern shows four characteristic peaks corresponding to the lattice plane (111), (220), (311), (400) of standard Cu_{2-x}Se (PDF#06-0680) (Supplementary Fig. 12C). According to the XPS result (Supplementary Fig. 12D), the molar percentages of Cu(I) and Cu(II) in Chem-Cu_{2-x}Se NPs were calculated to be 90.69% and 9.41%, respectively. These results confirm the successful synthesis of Chem-Cu_{2-x}Se NPs.

We further investigated and compared the performance of Bio-Cu_{2-x}Se@MR-1@PVDF and membrane that only mixed *S. oneidensis* cells with Chem-Cu_{2-x}Se NPs (Chem-Cu_{2-x}Se@MR-1@PVDF). Adding *S. oneidensis* cells to Chem-Cu_{2-x}Se@PVDF membrane decreases the contact angle of the membrane surface (Supplementary Fig. 13A), suggesting that the hydrophilicity of the membrane was improved by biomass. Meanwhile, Bio-Cu_{2-x}Se@MR-1@PVDF owned the best hydrophilicity (Supplementary Fig. 13A), mainly attributed to the combining merits of natural capping proteins on Cu_{2-x}Se NPs and the inherent hydrophilicity of biomass. Supplementary Fig. 13B shows the average water evaporation rate of membranes under 1 sun irradiation, which followed this order: Bio-Cu_{2-x}Se@MR-1@PVDF > Chem-Cu_{2-x}Se@MR-1@PVDF > Chem-Cu_{2-x}Se@PVDF. Consistently, the highest solar evaporation efficiency was obtained in Bio-Cu_{2-x}Se@MR-1@PVDF (Supplementary Fig. 13C).

Impressively, after photothermal evaporation experiment, the color of Chem-Cu_{2-x}Se@MR-1@PVDF and Chem-Cu_{2-x}Se@PVDF membranes changed from deep black to dark green and green (Supplementary Fig. 14A), respectively, indicating air oxidation of copper in the membrane²¹. In contrast, Bio-Cu_{2-x}Se@MR-1@PVDF membrane maintains deep black. Subsequently, the structural, chemical and performance stabilities of the evaporator membrane were investigated by monitoring copper content in the solution and the valence states of Cu in the membrane before and after 1 sun irradiation for 10 h. After 10 h treatment, part of Chem-Cu_{2-x}Se NPs detached from the supporting material, supported by the enhanced copper content in the solution (Supplementary Fig. 14B). XPS results of Chem-Cu_{2-x}Se@PVDF and Chem-Cu_{2-x}Se@MR-1@PVDF membranes showed that the energy spectra of Cu 2p_{3/2} and Cu 2p_{1/2} shift to a higher binding energy value after 10 h irradiation (Supplementary Fig. 15A-15D), indicating that air oxidation leads to an increase in the Cu(II)/Cu(I) ratio.

The evaporation performance of Chem-Cu_{2-x}Se NPs was decreased after the evaporator was reused for 10 cycles (Supplementary Fig. 16B-16C). It is encouraging that Bio-Cu_{2-x}Se shedding from the membrane was negligible, exhibiting excellent structural stability (Supplementary Fig. 14B). According to XPS spectra, the binding energies of Cu 2p in Bio-Cu_{2-x}Se@MR-1@PVDF remained similar after 10 h irradiation (Supplementary Fig. 15E-15F, and Supplementary Fig. 16A), which proves that Bio-Cu_{2-x}Se@MR-1@PVDF has excellent chemical stability. Bio-Cu_{2-x}Se@MR-1@PVDF was reused 10 times for the evaporation test under 1 sun irradiation. The corresponding evaporation performance was relatively stable (Supplementary Fig. 16B-16C), exhibiting good reusability and durability. The high stability implies that biomass might act as a natural linker, enhancing the adhesion between the photothermal conversion material and the supporting material.

Meanwhile, the contents of TOC and protein in the solution were negligible, ruling out potential hazards from the degradation of MR-1 dead cells (Supplementary Fig. 16D). Together, Bio-Cu_{2-x}Se@MR-1@PVDF possesses the best evaporation performance among them, ensuring long-term stable solar steam generation (Supplementary Fig. 16B-16C). Such outstanding solar evaporation performance should be assigned to the good photothermal conversion efficiency of Cu_{2-x}Se, the high hydrophilicity of biomass, and the superior stability properties of Bio-Cu_{2-x}Se@MR-1@PVDF.” (P18-20, L370-417)

“Chemical Synthesis and Characterization of Cu_{2-x}Se NPs

According to the previous Cu_{2-x}Se NPs synthesis protocol²¹, 55 mL deionized water and 16 mL polyvinylpyrrolidone K30 solution (PVP-K30, 5 mg mL⁻¹) were mixed evenly in a beaker. Subsequently, 1 mL selenium dioxide (SeO₂, 0.2 M) and 3 mL vitamin C (Vc, 0.4 M) solutions were added. After stirring for 15 minutes, the solution turned red, which is a characteristic of Se⁰ NPs. A fresh mixture of 1 mL CuSO₄·5H₂O (0.4 M) and 4 mL Vc (0.4 M) was added to the above-obtained red Se solution. The obtained mixture was stirred at 30 °C for 8 h. The mixture turned from red to brown black, suggesting that PVP-modified Cu_{2-x}Se NPs were synthesized. The chemically synthesized Cu_{2-x}Se NPs were collected and washed with deionized water for 6 times. The washed Cu_{2-x}Se NPs were freeze-dried for further membrane construction. The as-prepared Cu_{2-x}Se NPs were dripped on a copper screen and then for TEM image (H-7650, Hitachi, Ltd., Japan). The as-prepared Cu_{2-x}Se NPs were freeze-dried (FD-1C-50, Beijing Boyikang Experimental Instrument Co., Ltd, China) and ground into a powder for further X-ray diffractometer (XRD, SmartLab, Rigaku Corporation, Japan)” (SI, P3, L33-46)

“For comparison, control membranes (Chem-Cu_{2-x}Se@MR-1@PVDF, Chem-Cu_{2-x}Se@PVDF) were prepared using the same process as Bio-Cu_{2-x}Se@MR-1@PVDF membrane. Except for that, Bio-Cu_{2-x}Se@MR-1 was replaced by mixing *S. oneidensis* cells with Chem-Cu_{2-x}Se NPs. The detailed parameters were list in Supplementary Table 3.” (P32, L697-700)

“Measurement of Raw and Distilled Water Quality

The concentrations of Na⁺, K⁺, Ca²⁺, Mg²⁺, B³⁺ and Cu²⁺ in the distilled water were quantified by ICP-MS (iCAP RQ, Thermo Fisher Scientific, USA). For TOC measurement, the collected samples were filtered through a 0.22 μm membrane and then mixed with 2 M HCl at a volume ratio of 20:1. The mixed solutions were used for TOC detection by total organic carbon analyzer (Multi N/C 2100 TOC, Analytik Jena, Germany). For protein detection, the collected samples were mixed with NaOH (final concentration, 1 mM). Then, the mixtures were treated at 95 °C for 10 minutes to extract the proteins. The suspensions were for protein detection by an enhanced BCA protein assay kit (P0009, Beyotime Biotechnology, Shanghai, China).” (SI, P4, L59-67)

“Structural, Chemical and Performance Stabilities of Membranes

To investigate the structural, chemical and performance stabilities of the evaporator membrane, membranes were irradiated under 1 solar light intensity for 10 h. Subsequently, the concentrations of copper in the solution were quantified by ICP-MS (iCAP RQ, Thermo Fisher Scientific, USA). For XPS detection, membranes were collected and washed three times with deionized water. The washed membranes were for XPS (Kratos Axis supra⁺, Shimadzu, Japan) measurement. The solar evaporation performance of the membranes was measured according to the method described in the manuscript.” (SI, P4, L697-700)

Supplementary Figure 12. Characteristics of the chemically synthesized Cu_{2-x}Se nanoparticles (Chem- Cu_{2-x}Se NPs). (A) TEM image of the Chem- Cu_{2-x}Se NPs. (B) Average particle size of Chem- Cu_{2-x}Se NPs. (C) XRD pattern of the Chem- Cu_{2-x}Se NPs and standard Cu_{2-x}Se (PDF#06-0680). (D) XPS spectrum with peak fitting for $\text{Cu}2p$.

Supplementary Figure 13. Characteristics and corresponding solar vapor generation performance of the Bio- $\text{Cu}_{2-x}\text{Se}@MR-1@PVDF$, Chem- $\text{Cu}_{2-x}\text{Se}@MR-1@PVDF$ and Chem- $\text{Cu}_{2-x}\text{Se}@PVDF$ membranes. (A) Contact angle of membranes. (B) Evaporation rate of membranes under 1 sun irradiation. (C) Photothermal conversion efficiency of membranes under 1 sun irradiation. The data points represent means \pm SD ($n = 6$). Error bars correspond to standard deviations. “**” represents $p < 0.0001$. “***” represents $p < 0.001$.**

Supplementary Figure 14. Stability of the Bio-Cu_{2-x}Se@MR-1@PVDF, Chem-Cu_{2-x}Se@MR-1@PVDF and Chem-Cu_{2-x}Se@PVDF membranes exposed to 1 sun. (A) Optical pictures of different membranes before/after 10 h photothermal distillation experiments. (B) The Cu leaching concentration in the original water body of different membranes exposed to 1 sun for 10 h of pure water. The data points represent means \pm SD (n = 6). Error bars correspond to standard deviations. “*****” represents p < 0.0001. “***” represents p < 0.001.

Supplementary Figure 15. XPS results of different membranes before (A, C and E) / after (B, D and F) the photothermal distillation experiments exposed to 1 sun for 10 h. XPS spectra with peak fitting of Cu 2p for (A, B) Chem-Cu_{2-x}Se@PVDF, (C, D) Chem-Cu_{2-x}Se@MR-1@PVDF and (E, F) Bio-Cu_{2-x}Se@MR-1@PVDF membranes.

Supplementary Figure 16. Characterization of the stability of different membrane structures and properties. (A) The proportion of Cu(II) or Cu(I) in the membranes before/after exposing to 1 sun irradiation for 10 h. (B) The evaporation rate of membranes before/after exposing to 1 sun irradiation for 10 h. (C) The photothermal conversion efficiency of membranes before/after exposing to 1 sun irradiation for 10 h. (D) TOC concentration and protein content in the original water body of Bio-Cu_{2-x}Se@MR-1@PVDF membrane exposed to 1 sun for 10 h of pure water. The data points represent means \pm SD (n = 6). Error bars correspond to standard deviations. “****” represents $p < 0.0001$.

5. In Line 180-181 and 247-248, the authors mention that lactate acts as the sacrificial electron donor for Se⁰ nanoparticles photocatalyst. It is known that electrons derived from lactate are typically released in the intracellular space through oxidation reactions. This implies that the extracellular electron transfer (EET) pathway, consisting of MtrCAB, should function in an outward direction to provide electrons for extracellular Se⁰ nanoparticles. Therefore, there appears to be a conflict between the outward EET pathway and the inward/reversed EET pathway, and the latter is proposed to transport photoelectrons generated by extracellular Se⁰ nanoparticles in Figure 4. Please clarify this conflict.

Thanks. For a more accurate description, the reverse EET has been modified as a reverse MtrABC complexes. *S. oneidensis* MR-1 is a particularly well-known model organism with the capacity to deliver internal electrons to the outer surface through the EET pathway consisting of CymA and Mtr cytochrome conduit. The electron direction is closely associated with the electron source and sink (Supplementary Table 8). The location of electron acceptor Cu(II) is in the periplasm in this work, resulting in metabolic electrons from the cytoplasm to the periplasm through CymA, while the Mtr pathway is not involved in this biocatalytic process. Under light illumination, the external Se⁰ semiconductor can be excited and generate photoelectrons. Mutant experiments and redox potential

analysis suggested that a reverse MtrABC pathway was observed to realize the inward electron flow from extracellular Se⁰ NPs to cells. The reversal of the MtrABC pathway was further validated by nitrate reduction with acetic acid as an electron donor (**Supplementary Figure 8**). The selection of nitrate as an electron acceptor was due to the known periplasmic location of nitrate reductase in *S. oneidensis* MR-1²².

We have updated the manuscript to highlight the direction of electron flow as follows:

“To further determine the direction of electron flow, we compared nitrate reduction by *S. oneidensis* MR-1 with/without light illumination (Supplementary Fig. 8). The selection of nitrate as an electron acceptor was due to the known periplasmic location of nitrate reductase in *S. oneidensis* MR-1 (Supplementary Fig. 8A)²². Under light conditions, nitrate reduction by *S. oneidensis*-Se⁰ hybrid was significantly higher than that obtained in dark conditions (Supplementary Fig. 8B). Impairing MtrABC cytochrome conduit led to a substantial decrease in nitrate reduction under light illumination, suggesting that periplasm reaction can be driven by external photoelectrons through the reverse MtrABC pathway. Meanwhile, the small amount of reduced nitrate in dark conditions likely comes from the remaining organic carbon in the biomass, which could be oxidized to produce electrons for nitrate reduction. Under these conditions, NO₂⁻ makes up the majority of the end product (Supplementary Fig. 8C). Overall, the MtrABC pathway works in a reverse direction to uptake photoelectrons from the Se⁰ semiconductor to the periplasmic electron acceptor.” (**P13-14, L271-280**)

“Recently, photoelectrons, cathode electrons, and chemical electrons have been injected into cells to boost the biological process for renewable energy recovery, value-added chemical production, and wastewater treatment (Supplementary Table 8). Notably, the function and direction of EET depend not only on the type and location of the upstream electron source, but also on the site of the downstream electron acceptor. If the electron acceptor is in extracellular space, metabolic electrons flow out of cells with natural EET²³. Alternatively, if the electron sink reacted in periplasm or cytoplasm, an inward electron flow was observed²⁴. In our system, the location of electron acceptor Cu(II) is in the periplasm, resulting in metabolic electrons from the cytoplasm to the periplasm through CymA. At the same time, the Mtr pathway is not involved in this biocatalytic process. Under light illumination, the external Se⁰ semiconductor can be excited and generate photoelectrons. Mutant experiments and redox potential analysis suggested that a reverse MtrABC pathway was observed to realize the inward electron flow from extracellular Se⁰ NPs to cells. The reversal of the MtrABC pathway was further validated by nitrate reduction with acetic acid as an electron donor.” (**P22-23, L479-492**)

“Nitrate Reduction Mediated by *S. oneidensis*-Se⁰ NPs

For nitrate reduction, the constructed *S. oneidensis*-Se⁰ NPs were collected, washed and resuspended in the anaerobic tube containing a modified SMB medium. The concentration of resuspended cells was adjusted to an optical density of 1.0 at 600 nm. The modified SMB contains (per liter) 0.09 g KCl, 0.67 g NaH₂PO₄·2H₂O, 11.91 g 4-(2-hydroxyethyl)-1-piperazineethanesulfonic acid (HEPES), 1 mM sodium nitrate and 20 mM acetate. The pH of the SMB was adjusted to 7.2. The anaerobic condition was prepared by purging with 100% N₂ for 30 min and sealed with rubber. The used illumination source was a collimated Xenon lamp (PLS-SXE300D, Beijing Pophile Technology Co., Ltd., China) with a 420 nm UV-cut filter. The reaction time was 10 h and the light intensity was 20 mW/cm². The concentrations of nitrate (NO₃⁻-N), nitrite (NO₂⁻-N) and ammonia (NH₄⁺-N) were measured according to the standard methods²⁵.”(**SI, P2, L22-32**)

Supplementary Table 8. The direction of EET in *Shewanella oneidensis* MR-1 cells

Electron Source	Electron Sink	Reaction site of electron acceptor	Electron flow	Ref.
Cathodic electrons (-360 mV vs SHE)	Fumarate (30 mV vs SHE)	Periplasm	Reverse MtrABC	26
Cathodic electrons (-303 mV vs SHE)	Oxygen reduction, NADH, FMNH ₂ generation	Cytoplasm	Reverse MtrABC	27
Cathodic electrons (-300 mV vs SHE)	2,3-butanediol	Cytoplasm	Reverse EET Reverse NADH dehydrogenases	28
Cathodic electrons (-400 mV vs SHE)	Nitrate reduction	Periplasm	Reverse MtrABC	22
Photoelectron from Cu ₂ O/RGO (-510 mV vs SHE)	H ₂	Periplasm	Reverse MtrABC	29
Photoelectron from CdS	Degradation of trypan blue	Extracellular space	Native EET	23
Photoelectron from CdS (-680 mV vs SHE)	H ₂	Periplasm	Reverse MtrABC	24
Chemical electron from iron-containing metals	Nitrate reduction	Periplasm	Reverse MtrABC	30
Photoelectron from Se ⁰ (-736 mV vs SHE)	Cu(II) reduction Cu _{2-x} Se NPs formation	Periplasm	Reverse MtrABC	This work

EET refers to extracellular electron transfer. refers to anthraquinone-2,6-disulfonate. RGO refers to reduced graphene oxide.

Supplementary Figure 8. Light-driven nitrate reduction by the *S. oneidensis*-Se⁰ hybrid. (A) Proposed mechanism of the light-driven nitrate reduction by the *S. oneidensis*-Se⁰ hybrid. (B) Concentrations of reduced nitrate (NO₃⁻-N) of the lysed hybrids that were co-incubated with WT/ΔomcAΔmtrC and nitrate with/without illumination for 10 h. (C) Concentrations of produced nitrite (NO₂⁻-N) and ammonia (NH₄⁺-N) of the lysed hybrids that were co-incubated with WT and nitrate with illumination for 10 h. The above experiments were performed in the mineral salt medium with 20 mM sodium acetate. The data points represent means ± SD (n = 3). Error bars correspond to standard deviations. “****” represents p < 0.0001. “ns” indicates not significant (p > 0.05).

6. Line 301, it should refer to Supplementary Fig. 7H.

Thanks. Corrected as suggested.

7. Line 427, please give more experimental details about fabrication of Cu_{2-x}Se@MR-1@PVDF membrane.

Thanks. A detailed description has been added to the revised manuscript as follows:

“Fabrication of Membranes

The Bio-Cu_{2-x}Se@MR-1@PVDF membrane was prepared using phase inversion approach³¹. The casting solution was prepared by mixing polyvinyl pyrrolidone (PVP) and N,N-dimethylformamide (DMF) according to Supplementary Table 3. For Bio-Cu_{2-x}Se@MR-1@PVDF membrane, the casting solution was prepared by dissolving 1 g PVP into 75 g DMF. Subsequently, 10 g freeze-dried Bio-Cu_{2-x}Se NPs was added into the casting solution and then sonicated for 20 min. 15 g PVDF was then added to the casting solution and stirred at room temperature for at least 24 h to get a uniform mixture solution. The obtained mixture was placed in a vacuum drying (30 °C) to remove bubbles. After 30 min, the mixture was spread onto a clean and dry glass plate. Then use a wet film coater four sided film applicator to obtain a membrane with a thickness of 200 μm. 1 minute later, the glass was immersed in deionized water for 30 min to detach the membrane from the glass. The as-prepared membranes

were cut into 4 cm-diameter circles and stored in pure water at 4 °C before use. For control PVDF membrane preparation, the casting solution was prepared by dissolving 1 g PVP into 85 g DMF. Subsequently, 15 g PVDF was added to the casting solution and stirred for at least 24 h.” (P31-32, L683-696)

8. Please give the complete names of the abbreviations used in Supplementary Table 3 and Supplementary Table 4.

Thanks. Corrected as suggested.

REFERENCES

1. Tao, F. et al. Copper sulfide-based plasmonic photothermal membrane for high-efficiency solar vapor generation. *ACS Appl. Mater. Interfaces* **10**, 35154-35163 (2018).
2. Qin, Y. et al. Dyeable PAN/CuS nanofiber membranes with excellent mechanical and photothermal conversion properties via electrospinning. *ACS Appl. Polym. Mater.* **4**, 9144-9150 (2022).
3. Chen, M. et al. Plasmonic nanoparticle-embedded poly(p-phenylene benzobisoxazole) nanofibrous composite films for solar steam generation. *Nanoscale* **10**, 6186-6193 (2018).
4. Liu, Y. et al. A bioinspired, reusable, paper-based system for high-performance large-scale evaporation. *Adv. Mater.* **27**, 2768-2774 (2015).
5. Zhou, X., Zhao, F., Guo, Y., Zhang, Y. & Yu, G. A hydrogel-based antifouling solar evaporator for highly efficient water desalination. *Energy Environ. Sci.* **11**, 1985-1992 (2018).
6. Li, Y. et al. 3D-printed, all-in-one evaporator for high-efficiency solar steam generation under 1 Sun illumination. *Adv. Mater.* **29** (2017).
7. Wang, G. et al. Reusable reduced graphene oxide based double-layer system modified by polyethylenimine for solar steam generation. *Carbon* **114**, 117-124 (2017).
8. Fu, Y. et al. Oxygen plasma treated graphene aerogel as a solar absorber for rapid and efficient solar steam generation. *Carbon* **130**, 250-256 (2018).
9. Shi, L., Wang, Y., Zhang, L. & Wang, P. Rational design of a bi-layered reduced graphene oxide film on polystyrene foam for solar-driven interfacial water evaporation. *J. Mater. Chem. A* **5**, 16212-16219 (2017).
10. Liang, H. et al. Thermal efficiency of solar steam generation approaching 100 % through capillary water transport. *Angew Chem. Int. Ed. Engl.* **58**, 19041-19046 (2019).
11. Yu, Z., Cheng, S., Li, C., Li, L. & Yang, J. Highly efficient solar vapor generator enabled by a 3D hierarchical structure constructed with hydrophilic carbon felt for desalination and wastewater treatment. *ACS Appl. Mater. Interfaces* **11**, 32038-32045 (2019).
12. Zhao, F. et al. Highly efficient solar vapour generation via hierarchically nanostructured gels. *Nat. Nanotechnol.* **13**, 489-495 (2018).
13. Kaur, M., Ishii, S., Shinde, S.L. & Nagao, T. All-ceramic microfibrillar solar steam generator: TiN plasmonic nanoparticle-loaded transparent microfibers. *ACS Sustainable Chem. Eng.* **5**, 8523-8528 (2017).
14. Cheng, H. et al. Tailoring core@shell structure of Cu_{2-x}Se@PDAs for synergistic solar-driven water evaporation. *J. Mater. Sci.* **57**, 11725-11734 (2022).
15. Xia, W., Cheng, H., Zhou, S., Yu, N. & Hu, H. Synergy of copper selenide/MXenes composite with enhanced solar-driven water evaporation and seawater desalination. *J. Colloid. Interface Sci.*

- 625**, 289-296 (2022).
16. Tao, F. et al. A plasmonic interfacial evaporator for high-efficiency solar vapor generation. *Sustain. Energy Fuels* **2**, 2762-2769 (2018).
 17. Guo, Z. et al. Super-hydrophilic copper sulfide films as light absorbers for efficient solar steam generation under one sun illumination. *Semicond. Sci. Technol.* **33**, 025008 (2018).
 18. Shang, M. et al. Full-spectrum solar-to-heat conversion membrane with interfacial plasmonic heating ability for high-efficiency desalination of seawater. *ACS Appl. Energy Mater.* **1**, 56-61 (2017).
 19. Wang, Z., Zhang, X.F., Shu, L. & Yao, J. Copper sulfide integrated functional cellulose hydrogel for efficient solar water purification. *Carbohydr. Polym.* **319**, 121161 (2023).
 20. Chen, J. et al. Photothermal membrane of CuS/polyacrylamide-carboxymethyl cellulose for solar evaporation. *ACS Appl. Polym. Mater.* **3**, 2402-2410 (2021).
 21. Zhang, S. et al. Vacancy engineering of Cu_{2-x}Se nanoparticles with tunable LSPR and magnetism for dual-modal imaging guided photothermal therapy of cancer. *Nanoscale* **10**, 3130-3143 (2018).
 22. Li, Y. et al. Microbial electrosynthetic nitrate reduction to ammonia by reversing the typical electron transfer pathway in *Shewanella oneidensis*. *Cell Reports Physical Science* **4**, 101433 (2023).
 23. Xiao, X. et al. Anaerobically photoreductive degradation by CdS nanocrystal: Biofabrication process and bioelectron-driven reaction coupled with *Shewanella oneidensis* MR-1. *Biochem. Eng. J.* **154**, 107466 (2020).
 24. Han, H.X. et al. Reversing electron transfer chain for light-driven hydrogen production in biotic-abiotic hybrid systems. *J. Am. Chem. Soc.* **144**, 6434-6441 (2022).
 25. E.W. Rice, R.B.B., A.D. Eaton Standard methods for the examination of water and wastewater. *Am. Phys. Educ. Rev.* **24**, (9) (1995).
 26. Ross, D.E., Flynn, J.M., Baron, D.B., Gralnick, J.A. & Bond, D.R. Towards electrosynthesis in *Shewanella*: energetics of reversing the mtr pathway for reductive metabolism. *PLoS One* **6**, e16649 (2011).
 27. Rowe, A.R. et al. Tracking electron uptake from a cathode into *Shewanella* cells: implications for energy acquisition from solid-substrate electron donors. *mBio* **9**, e02203-02217 (2018).
 28. Tefft, N.M. & TerAvest, M.A. Reversing an extracellular electron transfer pathway for electrode-driven acetoin reduction. *ACS Synth. Biol.* **8**, 1590-1600 (2019).
 29. Shen, H. et al. A whole-cell inorganic-biohybrid system integrated by reduced graphene oxide for boosting solar hydrogen production. *ACS Catal.* **10**, 13290-13295 (2020).
 30. Zhou, E. et al. Direct microbial electron uptake as a mechanism for stainless steel corrosion in aerobic environments. *Water Res.* **219**, 118553 (2022).
 31. Wang, X.-M. et al. Highly efficient near-infrared photothermal antibacterial membrane with incorporated biogenic CuSe nanoparticles. *Chem. Eng. J.* **405**, 126711 (2021).

Response to Reviewer 2's comments

Overview:

The paper entitled “Dual-Mode Harvest Solar Energy for Photothermal Cu_{2-x}Se Biomineralization and Seawater Desalination by Biotic-Abiotic Hybrid” presents an approach for utilizing photoelectrons for both Cu_{2-x}Se nanoparticles (NPs) biosynthesis and then for water desalination. To achieve the goals, the authors conducted serial experimental steps. First, the authors synthesized Se⁰ NPs from SeO₃²⁻ by utilizing the metal-reducing ability *Shewanella oneidensis* MR-1. Second, Cu_{2-x}Se NPs were synthesized driven by the photoelectrons generated by illuminating Se⁰ NPs via dual pathways: activating the periplasmic Cu(II) reduction network and the direct production of Cu(I). Third, the Cu_{2-x}Se@MR-1@PVDF membrane was prepared and tested for its performance in water desalination. The authors conducted various analyses to verify the mechanisms. However, based on my comments and questions below on the novelty, the publish-worthiness of this work on Nature Communication should be carefully considered.

Thanks. The novelty of this work has been highlighted by comparing this work with the reported works about Se⁰ NPs biosynthesis, Cu_{2-x}Se NPs biosynthesis, biotic-abiotic hybrids and water desalination (**Table R1**). The substantial difference between this work and the reported works has been elucidated by providing more convincing evidence and deepening the discussion section.

Table R1. Summary of the differences between recent studies and this work

Topic	Previous studies	This work	Table Panels
Biogenic Se ⁰	Synthetic mechanisms and regulatory strategies	In vivo application	Supplementary Table 6
Biogenic Cu _{2-x} Se NPs	Metabolic electron-driven	Metabolic electron & photoelectron-driven	Supplementary Table 7
	Regulating culture conditions	Sunlight-boost (simple and sustainable)	
	Formed in periplasm	Extracellular assembly	
	Normal Mtr pathway	Reversed Mtr pathway	
Reversed EET	For nitrate, fumarate, trypan blue reduction	For nanoparticles assembly	Supplementary Table 8
Photo-induced NPs biosynthesis	Inherent function (specific organisms)	Creating an unnatural photoelectronic pathway	Supplementary Table 9
Biotic-abiotic hybrid	Photoelectron transport along single pathway	Dual catalytic network	Supplementary Table 10
	Single site for photoelectron utilization	Two site for photoelectron utilization	
	For organic substance & H ₂ synthesis	For nanoparticles assembly	
Solar water production	Chemical synthesis (Harsh condition)	Biosynthesis (simple and sustainable)	Supplementary Table 4 and Table 5
	Instability	Good structural, chemical and performance stabilities (Biomass acts as natural linker)	

Here we report **several groundbreaking findings that may fundamentally change our perception of biotic-abiotic photocatalytic systems and hopefully open up a promising new horizon in this field.** 1) Solar energy was harvested in a **dual mode** for Cu_{2-x}Se NPs biosynthesis and seawater desalination by integrating the functionality synergism between bacterium and nanoparticles. 2) Photoelectrons generated by extracellular Se⁰ NPs wirelessly activate Cu_{2-x}Se synthesis through a **dual catalytic network** located in periplasm and extracellular space, respectively. 3) Under visible light, periplasmic Cu(II) reduction is initiated by **two electron fluxes**, either intracellular metabolic electron or extracellular photoelectron. 4) The unique photothermal feature of the photosynthetic Cu_{2-x}Se NPs and the natural hydrophilic and linking properties of bacterium offers a convenient way to tailor photothermal membranes for **solar water production**.

Furthermore, The structural, chemical and performance stabilities of the Bio-Cu_{2-x}Se@MR-1@PVDF membrane have been investigated by monitoring TOC content and protein concentration in solution, as well as the valence states of Cu in the membrane before and after 1 sun irradiation for 10 h. Encouragingly, the results rule out potential hazards from the degradation of MR-1 dead cells and prove the excellent stability of Bio-Cu_{2-x}Se@MR-1@PVDF. Impressively, compared with a chemically synthesized membrane (Chem-Cu_{2-x}Se@MR-1@PVDF), Bio-Cu_{2-x}Se NPs not only have better hydrophilicity and photothermal distillation performance, but also have excellent structural, chemical and performance stabilities (**Supplementary Figure 12-16**).

The data interpretation and literature discussion have been strengthened. We believe that the reviewers' concerns have been adequately addressed and the quality of the manuscript has been substantially improved to meet the high-standards of *Nature Communications*.

Comments:

1. Biosynthesis of Se⁰ NPs using *Shewanella oneidensis* MR-1 or other bacteria is well documented (please refer to [10.1021/jacs.7b07460](https://doi.org/10.1021/jacs.7b07460); [10.1038/srep03735](https://doi.org/10.1038/srep03735); [10.1016/j.procbio.2023.05.016](https://doi.org/10.1016/j.procbio.2023.05.016); [10.1271/bbb.90454](https://doi.org/10.1271/bbb.90454); [10.1007/s11274-022-03374-6](https://doi.org/10.1007/s11274-022-03374-6); [10.1016/j.jhazmat.2016.02.035](https://doi.org/10.1016/j.jhazmat.2016.02.035); [10.1016/j.saa.2017.11.050](https://doi.org/10.1016/j.saa.2017.11.050); etc.)

Thanks. This work substantially differed from the reported works. Indeed, the biological synthesis of Se⁰ NPs has been realized by various microorganisms (**Supplementary Table 6**). Previous works mainly focused on the synthetic mechanisms and regulatory strategies of biogenic Se⁰ NPs. However, the in vivo application of biogenic Se⁰ NPs, that couple the functions of bacteria and nanoparticles, has not been reported. Motivated by the photocatalytic function of Se⁰ NPs, the present study constructed *S. oneidensis*-Se⁰ hybrid for photo-boosted biological functions. *S. oneidensis* MR-1 possesses powerful nanoparticle assembly capability and diverse respiratory reductases, making it an excellent candidate for exploring *S. oneidensis*-Se⁰ hybrid system. This work reveals a previously unrecognized function of Se⁰ NPs in terms of accelerating bacterial functions through reversing EET, which is initiated by light illumination. With mutant analyses, we uncover the underlying synergistic mechanisms and elucidate the corresponding electronic circuits.

Accepting the reviewer's suggestion, we have updated the manuscript to highlight the novelty of this work as follows:

“Indeed, the biological synthesis of Se⁰ NPs has been realized by various microorganisms (Supplementary Table 6), which mainly focused on the synthetic mechanisms and regulatory strategies. However, the in vivo application of biogenic Se⁰ NPs, that couple the functions of bacteria and nanoparticles, has yet to be reported. Motivated by the photocatalytic function of Se⁰ NPs, the present

study constructed *S. oneidensis*-Se⁰ hybrid for photo-boosted biological functions. *S. oneidensis* MR-1 possesses powerful nanoparticle assembly capability and diverse respiratory reductases, making it an excellent candidate for exploring *S. oneidensis*-Se⁰ hybrid system. This work reveals a previously unrecognized function of Se⁰ NPs in terms of accelerating bacterial functions through reversing MtrABC, which is initiated by light illumination. With mutant analyses, we uncover the underlying synergistic mechanisms and elucidate the corresponding electronic circuits.” (P20-21, L433-443)

Supplementary Table 6. Mechanism, regulation strategies and applications of biogenic Se⁰ nanoparticles

Organism	Mechanism	Regulation strategies	Applications	Ref.
Stenotrophomonas maltophilia SeITE02	Alcohol dehydrogenase homolog	-	-	1
Azospirillum thiophilum	Structural and compositional properties of Se ⁰ NPs.	-	-	2
Shewanella sp. HN-41	-	Reaction time, biomass, selenite concentration	-	3
Shewanella sp. HN-41	-	Reaction time and biomass	-	4
Pseudoalteromonas shioyasakiensis	-	-	Antimicrobial, antifouling and cytotoxic activities	5
S. oneidensis MR-1 S. putrefaciens	-	Adding riboflavin	-	6
S. oneidensis MR-1	-	Regulate extracellular electron transfer	-	7
S. oneidensis MR-1	Fumarate reductase FccA	-	-	8
S. oneidensis MR-1	-	-	S. oneidensis -Se ⁰ hybrid for solar energy conversion	This work

2. Cu_{2-x}Se NPs biosynthesis by *Shewanella oneidensis* MR-1 or other bacteria is also reported before (please refer to 10.1021/acsami.3c03611; 10.1021/acs.est.2c04130; 10.1016/j.materresbull.2018.11.014; etc.). I understand two new findings in this manuscript: utilizing photoelectrons for biotic and abiotic production of Cu_{2-x}Se and quantifying each pathway. The finding on reversing electron transfer chain on *Shewanella oneidensis* MR-1 utilizing photoelectrons was also reported before 10.1021/jacs.2c00934.

Thanks for the reviewer’s constructive suggestion. This work substantially differed from the reported works^{9,10} and our previous works¹¹⁻¹³ in several aspects:

- 1) Different regulation strategies are adopted. Previous work optimized the biosynthesis process by regulating culture conditions, such as adding anthraquinone-2,6-disulfonate (AQDS) and adjusting the precursor concentration^{10, 11} (**Supplementary Table 7**). The present study proposed a much simpler and sustainable regulation approach that is sunlight-boost Cu_{2-x}Se NPs synthesis. Thus, the regulation mechanisms also differ. Previous studies find that EET combines with AQDS to facilitate intracellular electron efflux for extracellular Cu_{2-x}Se production. We find that under light irradiation, EET worked in a reversed direction to deliver extracellular photoelectron to periplasmic.
- 2) The electron source is different (**Supplementary Table 7**). Only the metabolic electron was harvested for Cu_{2-x}Se NP biosynthesis in previous work. Here, we coordinate metabolic electrons and photoelectrons to boost biological function.
- 3) Regarding the Cu_{2-x}Se NPs formation site, the previous study demonstrated that Cu_{2-x}Se NPs should be predominantly formed in the cell periplasm and then excreted out. Differently, we find that Cu_{2-x}Se NPs assembled extracellularly through efflux Cu(I) reacted with extracellular Se⁰ NPs. Such different formation sites may be attributed to the different Se precursors, the former used Na₂SeO₃ as Se source relating to periplasmic Se reduction, while we used Se⁰ NPs, that is predominately located in the extracellular space.
- 4) More importantly, Cu_{2-x}Se NPs biosynthesis is just a model product of the photo-driven bio-hybrid systems. Such *S. oneidensis*-Se⁰ hybrid can be applied for nitrate reduction (**Supplementary Fig. 8**) and HgSe synthesis (**Supplementary Fig. 9**), indicating the universal applicability of a photo-driven *S. oneidensis*-Se⁰ hybrid.
- 5) In terms of reverse EET (**Supplementary Table 8**). To the best of our knowledge, the study reported here is the first to couple the reverse Mtr pathway and exterior photocatalysis for nanoparticle assembly. This represents a proof of concept for inward photoelectron transfer for nanoparticle assembly and, with further development, can be used to upgrade bioproducts.
- 6) Compared to the single site and single pathway of photoelectron utilization in the reported works (**Supplementary Table 10**), photoelectrons generated by extracellular Se⁰ NPs wirelessly activate Cu_{2-x}Se NPs synthesis through a dual catalytic network located in periplasm and extracellular space, respectively. Such a dual catalytic network is a novel paradigm for balancing the source and sink of photoelectrons.

Accepting the reviewer's suggestion, we have updated the **RESULTS and DISCUSSION** section to highlight the novelty of this work as follows:

“Beyond Cu_{2-x}Se NPs bio-assembly, we also tested light-promoted biological processes, including nitrate reduction (Supplementary Fig. 8) and HgSe biosynthesis (Supplementary Fig. 9) by *S. oneidensis*-Se⁰ hybrid system. As expected, a significantly promoted nitrate reduction, Hg biotransformation and HgSe synthesis by sunlight illumination were observed. Similarly, abundant HgSe NPs were synthesized in *S. oneidensis*-Se⁰ hybrid under exposure to Hg with light illumination, but significantly hindered in dark conditions. The XRD pattern shows distinct 2θ peaks matching the (111), (220), and (311) facets of HgSe (PDF#08-0469) (Supplementary Fig. 9B). Thus, the biological process in *S. oneidensis*-Se⁰ hybrid system was drastically facilitated by light illumination, indicating the universal applicability of the photo-driven bio-hybrid system.” (**P15, L307-315**)

“Recently, we demonstrated the feasibility of extracellular biosynthesis of Cu_{2-x}Se NPs by *S. onedensis* (Supplementary Table 7)^{11, 12}. Such a biological process can be alleviated by adding anthraquinone-2,6-disulfonate (AQDS)¹¹, which promotes a metabolic electron from the outer

membrane to the electron acceptor. Meanwhile, the composition and photothermal properties of Cu_{2-x}Se NPs can be regulated by adjusting the precursor concentration¹⁰. These biogenic processes were found to be driven by metabolic electrons (Supplementary Table 7). In this work, under visible light illumination, periplasmic Cu(II) reduction is initiated by two electron fluxes, either intracellular metabolic electrons through CymA or extracellular photoelectron via reversed MtrABC. Subsequently, the reaction between intermediate Cu(I) and Se^0 NPs was facilitated by light irradiation. This photo-promoted biotransformation of Cu_{2-x}Se NPs by *S. oneidensis*- Se^0 hybrid is simple and sustainable. Regarding the Cu_{2-x}Se NPs formation site, the previous study showed that Cu_{2-x}Se NPs should be predominantly formed in the cell periplasm and then excreted out¹¹. Differently, we find that Cu_{2-x}Se NPs assembled extracellularly through efflux Cu(I) reacted with extracellular Se^0 NPs. Such different formation sites may attributed to the different selenium precursors. The former used sodium selenite as a selenium source relating to periplasmic selenite reduction. In contrast, we used Se^0 NPs, which are predominately located in extracellular space. In terms of EET working direction, previous studies find that EET worked in a normal direction in the presence of AQDS to deliver intracellular electrons for Cu_{2-x}Se production. We find that under light irradiation, EET worked in a reversed direction to deliver extracellular photoelectrons from Se^0 NPs to periplasmic Cu(II) reduction for subsequent extracellular Cu_{2-x}Se NPs assembly.” (P21-22, L444-463)

“Biotic-abiotic photosynthetic systems hold great promise to innovate solar-driven chemical transformation. Nevertheless, balancing the generation and utilization of photoelectrons, illustrating the fate of photoelectrons, and diversifying solar energy transduction products are still challenges. Here, we select an electro-active bacterium *S. oneidensis* MR-1 as a model for hybrid photosynthetic system construction, attracted by its multimodal interfacial photoelectrons utilizing channels and inherent extracellular assembly of multi-functional nanoparticles. To the best of our knowledge, biogenic Se^0 NPs were first selected as a photosensitizer to construct a biohybrid system (Supplementary Table 10), attracted by their excellent biocompatibility and appropriate band position. Compared to the single site and single pathway of photoelectron utilization in the reported works (Supplementary Table 10), photoelectrons generated by extracellular Se^0 NPs wirelessly activate Cu_{2-x}Se NPs synthesis through a dual catalytic network located in periplasm and extracellular space, respectively. Such a dual catalytic network is a novel paradigm for balancing the source and sink of photoelectrons. Besides, we expand the solar-to-chemical production from organic substance and hydrogen to nanoparticles (Supplementary Table 10), diversifying solar energy conversion products in biotic-abiotic hybrid platforms. Our work possesses several groundbreaking findings that may fundamentally change our perception of biotic-abiotic photocatalytic systems and may bring biotic-abiotic photocatalysis a step forward toward application.” (P24-25, L508-525)

Supplementary Figure 8. Light-driven nitrate reduction by the *S. oneidensis*-Se⁰ hybrid. (A) Proposed mechanism of the light-driven nitrate reduction by the *S. oneidensis*-Se⁰ hybrid. (B) Concentrations of reduced nitrate (NO₃⁻-N) of the lysed hybrids that were co-incubated with WT/ΔomcAΔmtrC and nitrate with/without illumination for 10 h. (C) Concentrations of produced nitrite (NO₂⁻-N) and ammonia (NH₄⁺-N) of the lysed hybrids that were co-incubated with WT and nitrate with illumination for 10 h. The above experiments were performed in the mineral salt medium with 20 mM sodium acetate. The data points represent means ± SD (n = 3). Error bars correspond to standard deviations. “****” represents p < 0.0001. “ns” indicates not significant (p > 0.05).

Supplementary Figure 9. Light-driven HgSe NPs assembly by the *S. oneidensis*-Se⁰ hybrid and the characteristics of the biogenic HgSe NPs. (A) The precipitated mercury concentration by the *S. oneidensis*-Se⁰ hybrid or cell-inactivated lysed hybrid with/without illumination. (B) XRD patterns of the formed *S. oneidensis*-HgSe NPs and standard HgSe (PDF#08-0469). (C) SEM image and the corresponding EDS mapping images of (D) Se and (E) Hg elements of the photosynthesized *S. oneidensis*-HgSe nanoparticles. The above experiments were performed in the mineral salt medium with 20 mM sodium lactate. The data points represent means \pm SD (n = 6). Error bars correspond to standard deviations. “*****” represents p < 0.0001.

1

Supplementary Table 7. Biosynthesis and the corresponding applications of Cu_{2-x}Se nanoparticles

Organism	Precursors	Formation Site	Electron Source	Regulation Strategies	Mtr pathway Direction	Applications	Ref.
Pantoea agglomerans	SeO ₃ ²⁻ & EDTA-Cu ²⁺	-	Metabolic electron	-	-	Photocatalytic degradation of methylene blue	9
S. onedensis	SeO ₃ ²⁻ & Cu ²⁺	-	Metabolic electron	-	Normal	Photothermal antibacterial membrane	12
S. onedensis	SeO ₃ ²⁻ & Cu ²⁺	-	Metabolic electron	Adding AQDS	Normal	-	11
S. onedensis	SeO ₃ ²⁻ & Cu ²⁺	Periplasmic synthesis and then efflux out	Metabolic electron	Adjusting precursor concentration	Normal	Photothermal therapy	10
S. onedensis	Se ⁰ & EDTA-Cu ²⁺	Extracellular fabrication	Metabolic electron & Photoelectron	Solar energy	Reversed	Solar water production	This work

2 EDTA refers to ethylenediaminetetraacetic acid tetrasodium salt dihydrate. AQDS refers to anthraquinone-2,6-disulfonate.

Supplementary Table 10. Comparison of whole-cell biotic-abiotic hybrid systems for solar energy conversion

Living organism	Materials	Synthesis methods	Location of semiconductor	Photoelectron transport to Product	Photoelectron utilization site	Products	Ref.
Engineered Saccharomyces cerevisiae	InP	Chemical	On cell surface	One pathway	Cytoplasm	Shikimic acid	14
Moorella thermoacetica	Gold nanoclusters	Chemical	Cytoplasm	One pathway	Cytoplasm	Acetic acid	15
M. thermoacetica	CdS	Biosynthesis	On cell surface	One pathway	Cytoplasm	Acetic acid	16
Escherichia coli	CdS nanocluster	Biosynthesis	Periplasm	One pathway	Cytoplasm	Malate	17
Methanosarcina barkeri	NiCu@CdS	Chemical	On cell surface	One pathway	Cytoplasm	Methane	18
Engineered E. coli	CdS	Biosynthesis	On cell surface	One pathway	Cytoplasm	Hydrogen	19
Desulfovibrio desulfuricans	CdS	Biosynthesis	On cell surface	One pathway	-	Hydrogen	20
S. oneidensis MR-1	CuInS ₂ /ZnS QDs	Chemical	Periplasm	One pathway	Cytoplasm	Hydrogen	21
S. oneidensis MR-1	RGO and Cu ₂ O	Chemical	Around cell	One pathway	Periplasm	Hydrogen	22
S. oneidensis MR-1	CdS	Biosynthesis	On cell surface	One pathway	Periplasm	Hydrogen	13
S. oneidensis MR-1	Se ⁰	Biosynthesis	On cell surface & Extracellular	Dual-Mode	Periplasm & Extracellular	Cu _{2-x} Se NPs	This work

RGO refers to reduced graphene oxide. QDs refers to quantum dots.

- Using Cu_{2-x}Se NPs containing MR-1 cells to fabricate $\text{Cu}_{2-x}\text{Se}@MR-1@PVDF$ membrane. The photothermal property of Cu_{2-x}Se enhanced water evaporation in the desalination process. However, this application is not very new (please refer to 10.1016/j.jcis.2022.06.028; 10.1002/cssc.202201543; 10.1007/s10853-022-07353-y). Moreover, I worry about the MR-1 dead cell degradation and air oxidation of Cu in Cu_{2-x}Se NPs during the use of the fabricated membrane.

Thanks for the reviewer's constructive suggestion. This work substantially differed from the reported works using Cu-composed materials to construct solar steam generators in several aspects (**Supplementary Table 5**). Most of the reported works focus on traditional chemical synthesis methods, involving aggressive chemical agents, harsh synthesis conditions, intensive energy consumption, and complex synthesis processes. Besides, the anchored photothermal material can detach from the supporting material during long-term irradiation or high-frequency use, resulting in poor durability²³. Here, the as-prepared Bio- $\text{Cu}_{2-x}\text{Se}@MR-1@PVDF$ was prepared at room temperature using environmentally benign reagents, providing a more sustainable manufacturing strategy. Additionally, biomass, as a natural linker, enhances the adhesion between the photothermal conversion material and the supporting material. Compared with the typical photothermal conversion materials (**Fig. 5G and Supplementary Table 4 and Table 5**), the water evaporation conversion efficiency of the resulting membrane is top-ranking. Thus, the Bio- $\text{Cu}_{2-x}\text{Se}@MR-1@PVDF$ has the advantages of being inexpensive, sustainable, simple to fabricate, highly stable, and highly efficient in water evaporation, meeting the practical requirements for solar vapor generation.

The structural and chemical stabilities of the Bio- $\text{Cu}_{2-x}\text{Se}@MR-1@PVDF$ membrane have been investigated by monitoring TOC content and protein concentration in solution, as well as the valence states of Cu in the membrane before and after 1 sun irradiation for 10 h. Encouragingly, the contents of TOC and protein in the solution were negligible, ruling out potential hazards from the degradation of MR-1 dead cells (**Supplementary Figure 16D**). According to XPS spectra, the binding energies of Cu 2p in Bio- $\text{Cu}_{2-x}\text{Se}@MR-1@PVDF$ remained similar after 10 h irradiation (**Supplementary Figure 15E-15F**), which proves that Bio- $\text{Cu}_{2-x}\text{Se}@MR-1@PVDF$ has excellent chemical stability.

Accepting the reviewer's suggestion, we have updated the Results and Discussion section to highlight the novelty of this work as follows:

“Recently, Cu-composed materials have been considered as the most promising photothermal material, owing to their relatively low cost, non-toxic, eco-friendly and broadband light absorption. To date, various Cu-composed materials have been employed to construct solar steam generators and have achieved encouraging performance (**Supplementary Table 5**). Most of the reported works focus on traditional chemical synthesis methods, involving aggressive chemical agents, harsh synthesis conditions, intensive energy consumption, and complex synthesis processes (**Supplementary Table 5**). Besides, the anchored photothermal material can detach from the supporting material during long-term irradiation or high-frequency use, resulting in poor durability²³. Here, the as-prepared Bio- $\text{Cu}_{2-x}\text{Se}@MR-1@PVDF$ was prepared at room temperature using environmentally benign reagents, providing a more sustainable manufacturing strategy. Additionally, biomass, as a natural linker, enhances the adhesion between the photothermal conversion material and the supporting material. Compared with the typical photothermal conversion materials (**Supplementary Table 4 and 5**), the water evaporation conversion efficiency of the resulting membrane is top-ranking. Thus, the Bio- $\text{Cu}_{2-x}\text{Se}@MR-1@PVDF$ has the advantages of being inexpensive, sustainable, simple to fabricate, highly stable, and highly efficient in water evaporation, meeting the practical requirements for solar vapor generation.” (**P25-26, L535-551**)

“Impressively, after photothermal evaporation experiment, the color of Chem- $\text{Cu}_{2-x}\text{Se}@MR-1@PVDF$ and Chem- $\text{Cu}_{2-x}\text{Se}@PVDF$ membranes changed from deep black to dark

green and green (Supplementary Fig. 14A), respectively, indicating air oxidation of copper in the membrane²⁴. In contrast, Bio-Cu_{2-x}Se@MR-1@PVDF membrane maintains deep black. Subsequently, the structural, chemical and performance stabilities of the evaporator membrane were investigated by monitoring copper content in the solution and the valence states of Cu in the membrane before and after 1 sun irradiation for 10 h. After 10 h treatment, part of Chem-Cu_{2-x}Se NPs detached from the supporting material, supported by the enhanced copper content in the solution (Supplementary Fig. 14B). XPS results of Chem-Cu_{2-x}Se@PVDF and Chem-Cu_{2-x}Se@MR-1@PVDF membranes showed that the energy spectra of Cu 2p_{3/2} and Cu 2p_{1/2} shift to a higher binding energy value after 10 h irradiation (Supplementary Fig. 15A-15D), indicating that air oxidation leads to an increase in the Cu(II)/Cu(I) ratio.

The evaporation performance of Chem-Cu_{2-x}Se NPs was decreased after the evaporator was reused for 10 cycles (Supplementary Fig. 16B-16C). It is encouraging that Bio-Cu_{2-x}Se shedding from the membrane was negligible, exhibiting excellent structural stability (Supplementary Fig. 14B). According to XPS spectra, the binding energies of Cu 2p in Bio-Cu_{2-x}Se@MR-1@PVDF remained similar after 10 h irradiation (Supplementary Fig. 15E-15F, and Supplementary Fig. 16A), which proves that Bio-Cu_{2-x}Se@MR-1@PVDF has excellent chemical stability. Bio-Cu_{2-x}Se@MR-1@PVDF was reused 10 times for the evaporation test under 1 sun irradiation. The corresponding evaporation performance was relatively stable (Supplementary Fig. 16B-16C), exhibiting good reusability and durability. The high stability implies that biomass might act as a natural linker, enhancing the adhesion between the photothermal conversion material and the supporting material. Meanwhile, the contents of TOC and protein in the solution were negligible, ruling out potential hazards from the degradation of MR-1 dead cells (Supplementary Fig. 16D). Together, Bio-Cu_{2-x}Se@MR-1@PVDF possesses the best evaporation performance among them, ensuring long-term stable solar steam generation (Supplementary Fig. 16B-16C). Such outstanding solar evaporation performance should be assigned to the good photothermal conversion efficiency of Cu_{2-x}Se, the high hydrophilicity of biomass, and the superior stability properties of Bio-Cu_{2-x}Se@MR-1@PVDF.” (P18-20, L370-417)

“Measurement of Raw and Distilled Water Quality

The concentrations of Na⁺, K⁺, Ca²⁺, Mg²⁺, B³⁺ and Cu²⁺ in the distilled water were quantified by ICP-MS (iCAP RQ, Thermo Fisher Scientific, USA). For TOC measurement, the collected samples were filtered through a 0.22 μm membrane and then mixed with 2 M HCl at a volume ratio of 20:1. The mixed solutions were used for TOC detection by total organic carbon analyzer (Multi N/C 2100 TOC, Analytik Jena, Germany). For protein detection, the collected samples were mixed with NaOH (final concentration, 1 mM). Then, the mixtures were treated at 95 °C for 10 minutes to extract the proteins. The suspensions were for protein detection by an enhanced BCA protein assay kit (P0009, Beyotime Biotechnology, Shanghai, China).” (SI, P4, L59-67)

“Structural, Chemical and Performance Stabilities of Membranes

To investigate the structural, chemical and performance stabilities of the evaporator membrane, membranes were irradiated under 1 solar light intensity for 10 h. Subsequently, the concentrations of copper in the solution were quantified by ICP-MS (iCAP RQ, Thermo Fisher Scientific, USA). For XPS detection, membranes were collected and washed three times with deionized water. The washed membranes were for XPS (Kratos Axis supra⁺, Shimadzu, Japan) measurement. The solar evaporation performance of the membranes was measured according to the method described in the manuscript.” (SI, P4, L697-700)

Supplementary Figure 14. Stability of the Bio-Cu_{2-x}Se@MR-1@PVDF, Chem-Cu_{2-x}Se@MR-1@PVDF and Chem-Cu_{2-x}Se@PVDF membranes exposed to 1 sun. (A) Optical pictures of different membranes before/after 10 h photothermal distillation experiments. (B) The Cu leaching concentration in the original water body of different membranes exposed to 1 sun for 10 h of pure water. The data points represent means \pm SD (n = 6). Error bars correspond to standard deviations. “**” represents p < 0.0001. “***” represents p < 0.001.**

Supplementary Figure 15. XPS results of different membranes before (A, C and E) / after (B, D and F) the photothermal distillation experiments exposed to 1 sun for 10 h. XPS spectra with peak fitting of Cu 2p for (A, B) Chem-Cu_{2-x}Se@PVDF, (C, D) Chem-Cu_{2-x}Se@MR-1@PVDF and (E, F) Bio-Cu_{2-x}Se@MR-1@PVDF membranes.

Supplementary Figure 16. Characterization of the stability of different membrane structures and properties. (A) The proportion of Cu(II) or Cu(I) in the membranes before/after exposing to 1 sun irradiation for 10 h. (B) The evaporation rate of membranes before/after exposing to 1 sun irradiation for 10 h. (C) The photothermal conversion efficiency of membranes before/after exposing to 1 sun irradiation for 10 h. (D) TOC concentration and protein content in the original water body of Bio-Cu_{2-x}Se@MR-1@PVDF membrane exposed to 1 sun for 10 h of pure water. The data points represent means \pm SD (n = 6). Error bars correspond to standard deviations. “****” represents $p < 0.0001$.

Supplementary Table 5. Comparison of solar steam device based on Cu-composed nanoparticles

Supporting Material	Absorber	Synthesis of Absorber	Synthesis Conditions of Absorber	Evaporation Rate (kg m ⁻² h ⁻¹)	Conversion Efficiency (%)	Membrane Temperature (°C)	Ref.
Filter paper	Cu _{2-x} Se@polydopamine	-	Using PVP, ascorbic acid, dopamine, H ₂ O ₂	2.71	-	-	25
Glass microfiber	Cu _{2-x} Se/Nb ₂ CT _x	-	Using hydrazine, PVP, H ₂ O ₂ , HF aqueous solution, over one week	1.2	-	39.7	26
PVDFM	Cu ₉ S ₅	Hydrothermal	180 °C for several hours	1.173	80.2 ± 0.6	36.1	27
PVDFM	CuS	Hydrothermal	180 °C for 18 h	1.43	90.4	38.5	28
MCE	CuS	Hydrothermal	140 °C for 12 h	1.12	80±2.5	42.8	29
SCM	CuS nanoflowers	Hydrothermal	120 °C for 18 h	1.09	68.6	35.2	23
Polyethylene	CuS	Hydrothermal	180 °C for 12 h	1.021	63.9	37.6	30
Cellulose hydrogel	CuS	Hydrothermal	120 °C for 12 h	2.2	87	-	31
PAAm-CMC	CuS	Hydrothermal	180 °C for 12 h	1.613	79	47.2	32
PVDFM	Cu _{2-x} Se	Photo-facilitated biosynthesis	Room temperature	1.44	90.55	41.2	This work

The above photothermal distillation membrane experiments were performed under 1 sun irradiation; PVDFM refers to the poly(vinylidene fluoride) membrane. PVP refers to poly(vinyl pyrrolidone). MCE refers to mixed cellulose ester membrane. SCM refers to semipermeable collodion membrane. PAAm-CMC refers to polyacrylamide (PAAm) and carboxymethyl cellulose (CMC).

Questions:

1. Line 119-120: the authors wrote: “acetic acid as an electron sacrifice agent.” I could not find the word “acetic acid” in the Methodology section of Cu_{2-x}Se NPs synthesis. Moreover, can MR-1 cannot use acetate as an electron donor. Did the authors use lactate?

Thanks. In fact, we added 20 mM sodium lactate or 20 mM sodium acetate as the electron donor. In the study of biotic-abiotic hybrid, a common question is the interference of internal metabolism, making it difficult to judge the coupling performance and the direction of electron flow. To solve this problem, we initially compared the performance of strains with acetate as an electron sacrifice agent. Acetate was chosen because *S. oneidensis* MR-1 cannot metabolize it. In this case, the electron flow from respiration is wholly suppressed, excluding the interference of metabolic electron flow. Subsequently, to uncover the pathway of electron flow during normal metabolism, acetate was replaced with lactate, which can act as both a carbon source and an electron donor.

Accepting the reviewer’s suggestion, we have updated the manuscript as follows:

“Having constructed the *S. oneidensis*-Se⁰ hybrid, the light-activating Cu_{2-x}Se NPs biosynthesis by the as-prepared hybrid was further tested with sodium acetate as an electron sacrifice agent under different conditions (Fig. 2A). Acetate was chosen because *S. oneidensis* MR-1 cannot metabolize it³³. In this case, the electron flow from respiration is almost wholly suppressed, excluding the interference of metabolic electrons.” (P7, L126-130)

“To uncover the pathway of electron flow during normal metabolism, sodium acetate was replaced with sodium lactate in Figure 3, which can act as both a carbon source and an electron donor.” (P14, L172-174)

“The SMB contains (per liter) 1.49 g NH₄Cl, 0.09 g KCl, 0.67 g NaH₂PO₄·2H₂O, 0.0002 g flavin, 11.91 g 4-(2-hydroxyethyl)-1-piperazineethanesulfonic acid (HEPES), 29.41 g sodium citrate. In SMB medium, the carbon source was 20 mM sodium lactate or 20 mM sodium acetate.” (P27, L588-591)

2. The authors sometimes used UV-vis-NIR (Fig.2), and sometimes used UV-vis-DRS (Fig. 1). However, only UV-vis-DRS was described in the Methodology section (Line 362 onward).

Thanks. We have added a UV-Vis-NIR description in the **Methodology** section.

“For UV-visible-near-infrared spectrophotometer (UV-vis-NIR) measurement, the treated cells were collected by centrifugation (6000 g, 5 min) and washed with deionized water for three times. The collected precipitate was resuspended in deionized water for UV-vis-NIR detection (UV-Vis-NIR, Shimadzu’s SolidSpec-3700/3700DUV spectrophotometers). The test range was 450-1300 nm.” (P29, L624-628)

3. Can the authors calculate the amount of Cu_{2-x}Se NPs produced from 1 mole lactate?

Thanks. We have detected consumed lactate, biotransformed copper content, and the Cu(I) proportion in the Cu_{2-x}Se NPs to calculate the amount of Cu_{2-x}Se NPs produced from 1 mole lactate. The proportion of Cu(I) in Cu_{2-x}Se NPs was obtained from XPS result (**Supplementary Fig. 5B and 5E**). We calculated that Cu_{2-x}Se NPs produced by the *S. oneidensis*-Se⁰ hybrid at 1 mM lactate consumption with or without light were about 21.80 mg and 11.74 mg, respectively (**Supplementary Table. 2**). The amount of Cu_{2-x}Se NPs under light illumination was 1.866 times that under dark conditions.

Accepting the reviewer’s suggestion, we have updated the manuscript as follows:

“We calculated that Cu_{2-x}Se NPs produced by the *S. oneidensis*-Se⁰ hybrid at 1 mM lactate consumption with or without light were about 21.80 mg and 11.74 mg, respectively (Supplementary Table. 2). The amount of Cu_{2-x}Se NPs under light illumination was 1.86 times that under dark conditions.” (P15, L298-301)

“**Detection of Lactate**

Lactate was measured by high-performance liquid chromatography (HPLC, Agilent 1260 Infinity, Germany) equipped with carbohydrate columns (Agilent Hi-Plex PL1170-6830, Germany). The mobile phase is 5 mM H₂SO₄ at a flow rate of 0.5 mL/min. The lactate was detected with a differential refractive index detector. The samples were centrifuged at 10000 rpm for 5 min. The collected supernatant was filtered through a 0.22 μm membrane and then for HPLC detection.” (SI, P3, L48-53)

Supplementary Figure 5. XPS results of the biogenic Cu_{2-x}Se NPs by *S. oneidensis*-Se⁰ hybrid (A-C) with / (D-F) without illumination for 5 h. (A, D) XPS survey spectrum, (B, E) Cu 2p spectrum and (C, F) Se 3d spectrum.

Supplementary Table 2. The amount of Cu_{2-x}Se NPs produced by *S. oneidensis*-Se⁰ hybrid with lactate supplementation and incubated for 5 h.

Groups	Consumed Lactate (mM)	Cu _{2-x} Se NPs Synthesis					
		Precipitated Cu (mg)	Cu(I) Ratio (%)	Produced Cu ₂ Se (mg)	Precipitated Cu (mg)	Produced Cu _{2-x} Se NPs (mg)	Ratio: produced Cu _{2-x} Se NPs / consumed lactate (mg /mM)
Light	1.03	12.78	78.15	16.19	6.26	22.45	21.80
Dark	0.86	5.84	82.61	7.82	2.28	10.10	11.74

The proportion of Cu(I) in Cu_{2-x}Se NPs was obtained from XPS result (Supplementary Fig. 5B and 5E).

4. Line 201-202: “CymA protein is not a Cu(II) terminal reductase and may be upstream of Cu(II) terminal reductase”. I suppose the idea is “CymA protein is not ONLY a Cu(II) terminal reductase.” Am I correct?

With mutant analysis, we found that CymA protein is not a Cu(II) terminal reductase. Additionally, it predominantly serves as an upstream of Cu(II) terminal reductase, working as a relay to transport metabolic electron to Cu(II) reduction. Our results showed that Cu_{2-x}Se NPs synthesis was retained in $\Delta cymA$ mutant but not completely inhibited under light illumination (Fig. 3F-3G), indicating that the CymA was not a Cu(II) terminal reductase. To the best of our knowledge, the Cu(II) terminal reductase in *S. oneidensis* remains unrevealed yet.

We have updated the manuscript to clarify the function of CymA as follows:

“To elucidate the exact Cu(II) reduction site, we further coupled CymA-inactivated strain ($\Delta cymA$) with a cell-lysed hybrid for Cu_{2-x}Se NPs synthesis. CymA is a critical cytoplasmic membrane-anchored c-type cytochrome that transfers electron equivalents from central metabolism to periplasm (Fig. 3A)³⁴. Deleting CymA protein severely inhibited copper conversion toward Cu_{2-x}Se NPs production under dark conditions (Fig. 3D-3E, Supplementary Fig. 6B), indicating that CymA is an essential protein for Cu(II) reduction and Cu_{2-x}Se NPs formation. Interestingly, the elimination of Cu(II) transformation using $\Delta cymA$ strain could be recovered by light (Fig. 3F-3G). When exposed to light, $\Delta cymA$ exhibited a slightly lower copper accumulation and transformation than that obtained in WT (Fig. 3F-3G), implying that CymA protein was not a Cu(II) terminal reductase, and light exposure activates another electron channel for Cu_{2-x}Se NPs photocatalytic synthesis. Based on the above results, CymA protein predominantly serves as an upstream of Cu(II) terminal reductase, working as a relay to transport metabolic electrons to Cu(II) reduction. Altogether, these results suggest that Cu(II) terminal reductases and reduction space predominantly in the periplasm.” (P11-12, L216-229)

Fig. 3 | Mechanism of the light-driven Cu_{2-x}Se NPs assembly by the *S. oneidensis*- Se^0 hybrid. (A) Schematic diagram shows the synthesis of Cu_{2-x}Se by lysed hybrids that were co-incubated with different strains and $\text{Cu}(\text{II})$ with/without illumination. (B) The precipitated copper concentration and (C) temperature rise (ΔT) of the lysed hybrids that were co-incubated with $\text{Cu}(\text{II})$ or $\text{Cu}(\text{I})$ for 5 h with/without illumination. (D) The precipitated copper concentration and (E) (ΔT) of the lysed hybrids that were co-incubated with different strains and $\text{Cu}(\text{II})$ for 5 h under dark conditions. (F) The precipitated copper concentration and (G) (ΔT) of the lysed hybrids that were co-incubated with different strains and $\text{Cu}(\text{II})$ for 5 h under light illumination. The above experiments were performed in the mineral salt medium with 20 mM sodium lactate. The data points represent means \pm SD ($n = 6$). Error bars correspond to standard deviations. “****” represents $p < 0.0001$.

5. About $\text{Cu}_{2-x}\text{Se}@MR-1@PVDF$ membrane:

- Please draw a diagram to explain the working mechanism of this membrane in the water desalination process (can be included in the Supporting Information section)

Accepting the reviewer’s constructive suggestion, we have drawn a diagram to explain the working mechanism of this membrane in the water desalination process as follows:

“Accordingly, a schematic diagram of the working mechanism of $\text{Bio-Cu}_{2-x}\text{Se}@MR-1@PVDF$ membrane in the water desalination process is plotted in Supplementary Fig. 17. Once the light is turned on, Cu_{2-x}Se NPs are excited and generate electron-hole pairs. The photoexcited electrons and holes relax to their band edges before recombining, along with converting excess irradiative energy into thermal energy³⁵. Meanwhile, Cu_{2-x}Se NPs have local surface plasmon resonance (LSPR) properties, in which free carriers can collectively resonate with incident photons and produce hot electrons. By incorporating Cu_{2-x}Se NPs and biomass into the PVDF membrane, $\text{Bio-Cu}_{2-x}\text{Se}@MR-1@PVDF$ can achieve an apparent increase in surface temperature. Simultaneously, the hydrophilic nature of bacterial cells and the porous structure of PVDF allow efficient water transport to the interface, allowing water to be efficiently transported to the thermally located surface. The excellent evaporation performance should be assigned to the synergistic effect between the good photothermal conversion efficiency of Cu_{2-x}Se NPs, the high hydrophilicity of biomass, the microchannel PVDF membrane, and the superior stability properties of $\text{Bio-Cu}_{2-x}\text{Se}@MR-1@PVDF$.” (P20, L418-431)

Supplementary Figure 17. Schematic illustration of the working mechanism of $\text{Bio-Cu}_{2-x}\text{Se}@MR-1@PVDF$ membrane in the water desalination process.

- Is it possible to add control without using membrane condition to Fig 5E-F without using membrane condition?

Thanks. We have investigated the performance of pure water (Fig. 5E-5F). The evaporation rate of pure water was only $0.24 \text{ kg m}^{-2} \text{ h}^{-1}$, which is much lower than that of the Bio-Cu_{2-x}Se@MR-1@PVDF. The relevant information has been added to the manuscript as follows:

“The solar evaporation rate of Bio-Cu_{2-x}Se@MR-1@PVDF is $1.44 \text{ kg m}^{-2} \text{ h}^{-1}$, which is much higher than that of the pristine PVDF membrane and pure water (Fig. 5F).” (P17, L349-350)

Fig. 5 | Physicochemical properties and corresponding solar vapor generation performance of the Bio-Cu_{2-x}Se@MR-1@PVDF and PVDF membranes. (A) Schematic diagram of the preparation of the Bio-Cu_{2-x}Se@MR-1@PVDF membrane for solar vapor generation. (B) Solar spectral irradiance (gray, left-hand side axis) and absorption (black, right-hand side axis) and the optical pictures (inset of Fig. 5B). left: Bio-Cu_{2-x}Se@MR-1@PVDF membrane. right: PVDF membrane. (C) Raman images of Cu-Se bond at 260 cm^{-1} for PVDF membrane (top) and Bio-Cu_{2-x}Se@MR-1@PVDF membrane (down). (D) SEM image of the cross-section of Bio-Cu_{2-x}Se@MR-1@PVDF membrane. (E) Water evaporation induced mass loss curves. (F) Evaporation rate of membranes under 1 sun irradiation. (G) Comparison of the photothermal conversion efficiency and evaporation rate of Bio-Cu_{2-x}Se@MR-1@PVDF membrane and the reported membranes. The data points represent means \pm SD ($n = 6$). Error bars correspond to standard deviations. “****” represents $p < 0.0001$.

- The author used WHO standard to compare the quality of treated water Supplementary Figure 7H. If producing drinking water is a target, I worry about the MR-1 dead cells' decomposition during the use of this membrane.

Accepting the reviewer's constructive suggestion, we have detected TOC content and protein concentration in distilled water (Supplementary Fig. 11B), which were much lower than the standards of the U.S. Environmental Protection Agency (EPA, 2017) standard. The relevant information has been added to the manuscript as follows:

“And the concentrations of Na^+ , K^+ , Ca^{2+} , Mg^{2+} , B^{3+} , and Cu^{2+} content in distilled water were much lower than the standards of drinkable desalination water set by the World Health Organization (WHO) (Supplementary Fig. 11A). TOC concentration in the distilled water was below the U.S. Environmental Protection Agency (EPA, 2017) standard for indirect potable reuse of 2 mg/L (Supplementary Fig. 11B)³⁶. Meanwhile, protein contents in the distilled water were negligible (Supplementary Fig. 11B), satisfying the reusable standards.” (P17-18, L364-369)

“Measurement of Raw and Distilled Water Quality

The concentrations of Na^+ , K^+ , Ca^{2+} , Mg^{2+} , B^{3+} and Cu^{2+} in the distilled water were quantified by ICP-MS (iCAP RQ, Thermo Fisher Scientific, USA). For TOC measurement, the collected samples were filtered through a 0.22 μm membrane and then mixed with 2 M HCl at a volume ratio of 20:1. The mixed solutions were used for TOC detection by total organic carbon analyzer (Multi N/C 2100 TOC, Analytik Jena, Germany). For protein detection, the collected samples were mixed with NaOH (final concentration, 1 mM). Then, the mixtures were treated at 95 °C for 10 minutes to extract the proteins. The suspensions were for protein detection by an enhanced BCA protein assay kit (P0009, Beyotime Biotechnology, Shanghai, China).” (SI, P3-4, L54-67)

Supplementary Figure 11. The quality of distilled water produced by Bio-Cu_{2-x}Se@MR-1@PVDF membrane exposed to 1 sun for 10 h in seawater system. (A) Concentrations of Na⁺, K⁺, Ca²⁺, Mg²⁺, B³⁺, Cu²⁺ in distilled water. (B) TOC concentration and protein content in distilled water. The data points represent means \pm SD (n = 6). Error bars correspond to standard deviations.

- In the membrane preparation process and during the use of this membrane for water desalination, how about the air oxidation of Cu in Cu_{2-x}Se?. Please refer to 10.1021/acs.chemmater.8b03967.

Thanks. The chemical stability of the Bio-Cu_{2-x}Se@MR-1@PVDF membrane has been investigated by monitoring copper valence states in the membrane before and after 1 sun irradiation for 10 h. According to XPS spectra, the binding energies of Cu 2p in Bio-Cu_{2-x}Se@MR-1@PVDF remained similar after 10 h irradiation (Supplementary Fig. 15E-15F), which proves that Bio-Cu_{2-x}Se@MR-

1@PVDF has excellent chemical stability. The corresponding description has been added to the revised manuscript as follows:

“According to XPS spectra, the binding energies of Cu 2p in Bio-Cu_{2-x}Se@MR-1@PVDF remained similar after 10 h irradiation (Supplementary Fig. 15E-15F, and Supplementary Fig. 16A), which proves that Bio-Cu_{2-x}Se@MR-1@PVDF has excellent chemical stability.” (P19, L404-406)

Supplementary Figure 15. XPS results of different membranes before (A, C and E) / after (B, D and F) the photothermal distillation experiments exposed to 1 sun for 10 h. XPS spectra with peak fitting of Cu 2p for (A, B) Chem-Cu_{2-x}Se@PVDF, (C, D) Chem-Cu_{2-x}Se@MR-1@PVDF and (E, F) Bio-Cu_{2-x}Se@MR-1@PVDF membranes.

6. Line 336-337: Can the author quantify the Se⁰ NPs production rate?

Following the reviewer's constructive suggestion, we have quantified the production rate of Se⁰ NPs by testing time-resolved selenite reduction and Se⁰ concentration (Supplementary Fig. 3). Previous work pointed out that the selenite reduction in *S. oneidensis* MR-1 can be used to reflect Se⁰ NPs production, as the reduced selenite fitted well with the produced Se⁰ NPs content.⁸ In this work, selenite reduction was well fit by the pseudo-first-order model, and the rate constant was estimated to be -0.2808 h⁻¹ (Supplementary Fig. 3A-3B). Additionally, an excellent linear relationship (R² = 0.9999) between Se⁰ NPs concentration and absorbance at 550 nm was obtained (Supplementary Fig. 3C), providing an approach for the determination of Se⁰ NPs content in solution. The formation of Se⁰ NPs was also well fit by the pseudo-first-order model with a rate constant about 0.2366 h⁻¹ (Supplementary Fig. 3A and 3C-3D). The corresponding description has been added to the revised manuscript as follows:

“Furthermore, we quantified the production rate of Se⁰ NPs by testing time-resolved selenite reduction and Se⁰ production concentration (Supplementary Fig. 3A). *S. oneidensis* MR-1 could quickly reduce selenite within 8 h. Selenite reduction was well fit by the pseudo-first-order model, and

the rate constant was estimated to be -0.2808 h^{-1} (Supplementary Fig. 3B). Additionally, an excellent linear relationship ($R^2 = 0.9999$) between Se^0 NPs concentration and absorbance at 550 nm was obtained (Supplementary Fig. 3C), providing an approach for the determination of Se^0 NPs content in solution. The formation of Se^0 NPs was also well fit by the pseudo-first-order model with a rate constant of about 0.2366 h^{-1} (Supplementary Fig. 3D). The similar rate constant between selenite reduction and Se^0 NPs formation indicates that selenite reduction in *S. oneidensis* MR-1 can be used to reflect Se^0 NPs production, consistent with previous work⁸.” (P6, L103-112)

“Selenite Reduction Analysis

Selenite reduction was monitored by detecting selenium in the supernatant using inductively coupled plasma mass spectrometry (ICP-MS, iCAP RQ, Thermo Fisher Scientific, USA). To quantify selenium content in the supernatant, biological samples were collected at the sampling time point and then centrifuged. Subsequently, the obtained supernatant was digested in a mixture of HNO_3 and HClO_4 (volume ratio = 4:1). The resulting digestive fluid was diluted to 5 mL with deionized water for further ICP-MS assay.

Assay of Se^0 NPs Content

To quantify time-resolved Se^0 content, biological samples were collected at the sampling time point. The obtained samples were centrifuged and washed three times with deionized water. Subsequently, the washed precipitate was re-suspended in deionized water. The corresponding absorption at 550 nm wavelength was measured using UV-VIS spectroscopy (UV-2600, Shimadzu, Japan).” (SI, P2, L8-20)

Supplementary Figure 3. The dynamics of selenite reduction and Se^0 NPs formation. (A) Time-resolved selenite reduction and Se^0 NPs formation concentration by *S. oneidensis* MR-1. (B) Kinetic curve of selenite reduction ($R^2 = 0.9876$). (C) A linear relationship ($R^2 = 0.9999$) between Se^0 NPs concentration and absorbance at 550 nm. (D) Kinetic curve of Se^0 NPs formation ($R^2 = 0.9940$). The above experiments were performed in LB mediums. The data points represent means \pm SD ($n = 3$). Error bars correspond to standard deviations.

7. Line 342 and Line 350: why did the Cu(II)-EDTA concentration need to be re-adjusted?. Is the total Cu(II)-EDTA concentration higher than 1 mM?

Thanks. To ensure the complete chelation between EDTA and Cu(II), we pre-parated the stock

solutions of EDTA-Cu(II) chelate solution via the reaction of $\text{CuSO}_4 \cdot 5\text{H}_2\text{O}$ (2.5 mol) and ethylenediaminetetraacetic acid disodium salt (EDTA-Na_2 , 2.5 mol) in 100 mL of deionized water. For Cu_{2-x}Se NPs synthesis, the final concentration of EDTA-Cu(II) was 1 mM. A modified description has been added to the revised manuscript as follows:

“To ensure the complete chelation between EDTA and Cu(II), we pre-parated the stock solutions of EDTA-Cu chelate solution via the reaction of $\text{CuSO}_4 \cdot 5\text{H}_2\text{O}$ (2.5 mol) and ethylenediaminetetraacetic acid disodium salt (EDTA-Na_2 , 2.5 mol) in 100 mL of deionized water. For Cu_{2-x}Se NPs synthesis, the final concentration of EDTA-Cu(II) was 1 mM.” (P27-28, L593-597)

8. Line 358-361: Please add more information on Raman measurement conditions, including laser power, Raman shift range, exposure times, number of accumulations, etc.

Thanks. Updated.

“Raman spectra and mapping of the sample were obtained using a micro-Raman spectrometer (LabRAM HR Evolution, Horiba Co., Japan) excited by a 532 nm laser. The as-prepared hybrids were dripped on a tinfoil plate and then for measurements. For Raman mapping, $50 \times$ objective was used. The used laser power was 10 mW, and the Raman shift range was $150\text{-}600\text{ cm}^{-1}$. The exposure times was 5 s. The number of accumulations was two.” (P28, L606-610)

9. Fig 5G: The comparison was made based on the information in Supplementary Table 4. Please include 3 materials for a better comparison: 3DHG, 3D graphene foam, and CTH. Moreover, Fig 5G will be much better if the author included the Evaporation Rate ($\text{kg}\cdot\text{m}^{-2}\cdot\text{h}^{-1}$) on the second Y-axis.

Thanks. Updated.

Fig. 5 | Physicochemical properties and corresponding solar vapor generation performance of the Bio-Cu_{2-x}Se@MR-1@PVDF and PVDF membranes. (A) Schematic diagram of the preparation of the Bio-Cu_{2-x}Se@MR-1@PVDF membrane for solar vapor generation. (B) Solar spectral irradiance (gray, left-hand side axis) and absorption (black, right-hand side axis) and the optical pictures (inset of Fig. 5B. left: Bio-Cu_{2-x}Se@MR-1@PVDF membrane. right: PVDF membrane). (C) Raman images of Cu-Se bond at 260 cm⁻¹ for PVDF membrane (top) and Bio-Cu_{2-x}Se@MR-1@PVDF membrane (down). (D) SEM image of the cross-section of Bio-Cu_{2-x}Se@MR-1@PVDF membrane. (E) Water evaporation induced mass loss curves. (F) Evaporation rate of membranes under 1 sun irradiation. (G) Comparison of the photothermal conversion efficiency and evaporation rate of Bio-Cu_{2-x}Se@MR-1@PVDF membrane and the reported membranes. The data points represent means ± SD (n = 6). Error bars correspond to standard deviations. “****” represents p < 0.0001.

10. Fig 1B: I believe Fig 1B is the same as Supplementary Figure 2B. Please re-check it.
Thanks. We deleted Supplementary Figure 2B.

11. Fig 3 B-G: please perform statistical analysis to compare conditions better.
Thanks. Updated.

“Statistical Analysis.

All experiments were performed in at least three parallel groups, and the results are shown as means ± standard deviation (SD). Independent samples t-test and one-way analysis of variance (GraphPad Prism version 9.5) were used to test the significant differences between two groups of data and the significant differences between multiple groups of data, respectively. Bonferroni was applied when performing an analysis of variance. The level of difference depends on the calculated p-value, and p < 0.05 indicates a significant difference.” (P4, L77-82)

Fig. 3 | Mechanism of the light-driven Cu_{2-x}Se NPs assembly by the *S. oneidensis*-Se⁰ hybrid. (A) Schematic diagram shows the synthesis of Cu_{2-x}Se by lysed hybrids that were co-incubated with different strains and Cu(II) with/without illumination. (B) The precipitated copper concentration and (C) temperature rise (ΔT) of the lysed hybrids that were co-incubated with Cu(II) or Cu(I) for 5 h with/without illumination. (D) The precipitated copper concentration and (E) (ΔT) of the lysed hybrids that were co-incubated with different strains and Cu(II) for 5 h under dark conditions. (F) The precipitated copper concentration and (G) (ΔT) of the lysed hybrids that were co-incubated with different strains and Cu(II) for 5 h under light illumination. The above experiments were performed in the mineral salt medium with 20 mM sodium lactate. The data points represent means \pm SD (n = 6). Error bars correspond to standard deviations. “****” represents p < 0.0001.

12. Fig 4: Is the question mark in this Figure the limitation of this study? Can it be FccA protein? (Please refer to 10.1038/srep03735)

Thanks for the constructive suggestions. To test the possible contribution of FccA to Cu_{2-x}Se NPs formation by *S. oneidensis*-Se⁰ hybrid, we have measured and compared the Cu_{2-x}Se NPs biotransformation by control and mutant with impaired FccA ability ($\Delta fccA$) (**Supplementary Fig. 7**). The biogenic Cu_{2-x}Se NPs (reflected by Cu bioaccumulation and ΔT index) by $\Delta fccA$ -Se⁰ hybrid was comparable with that of WT-Se⁰ system, excluding Cu(II) reduction and Cu_{2-x}Se NPs biosynthesis mediated by FccA protein with Se⁰ as the precursor.

A detailed description has been added to the revised manuscript as follows:

“Considering the critical role of periplasmic fumarate reductase FccA in selenite reduction in *S. oneidensis*⁸, we were curious whether FccA possesses the capacity to reduce Cu(II) and synthesize Cu_{2-x}Se NPs. To test this possibility, we measured and compared the Cu_{2-x}Se NPs biotransformation by control and mutant with impaired FccA ability ($\Delta fccA$). However, the biogenic Cu_{2-x}Se NPs (reflected by Cu bioaccumulation and ΔT index) by $\Delta fccA$ was comparable with that of the WT strain (Supplementary Fig. 7), indicating that FccA had no contribution to Cu(II) reduction and Cu_{2-x}Se NPs biosynthesis, with Se⁰ as the precursor. Furthermore, the identification of Cu(II) terminal reductases remains to be fully explored.” (P12, L230-237)

Supplementary Figure 7. The function of fumarate reductase FccA in Cu_{2-x}Se NPs synthesis. (A) The precipitated copper concentration and (B) temperature rise (ΔT) of the lysed hybrids that were co-incubated with WT/ $\Delta fccA$ strains for 5 h with/without illumination. The above experiments were performed in the mineral salt medium with 20 mM sodium lactate. The data points represent means \pm SD (n = 6). Error bars correspond to standard deviations. “ns” indicates not significant (p > 0.05).

13. Fig 2: Please conduct XAS or XPS analysis of Cu_{2-x}Se NPs.

Accept the reviewer's suggestion. We have performed XPS analysis of Cu_{2-x}Se NPs in **Supplementary Fig 5**. The relevant information has been added to the manuscript as follows:

“X-ray photoelectron spectroscopy (XPS) was used to clarify the valence states of Cu in the resultant Cu_{2-x}Se NPs under light illumination (Supplementary Fig. 5). The full-scan map of XPS displayed that it contains both Se and Cu elements (Supplementary Fig. 5A). The corresponding Cu 2p spectrum exhibited peaks centered at 931.7 and 951.6 eV were assigned to Cu(I) and peaks at 933.5 and 953.3 eV were attributed to Cu(II) (Supplementary Fig. 5B)³⁷, indicating the co-presence of Cu(II) and Cu(I) in the Cu_{2-x}Se NPs. According to the fitting results, the Cu(I)/Cu(II) ratio in the Cu_{2-x}Se NPs was about 3.57. The Se 3d spectrum was deconvoluted into two peaks³⁸, including Se 3d_{5/2} (52.2 eV) and Se 3d_{3/2} (53.1 eV) (Supplementary Fig. 5C), coinciding with the Se^{2-} . These results further confirmed the successful photocatalysis of Cu_{2-x}Se NPs by *S. oneidensis*- Se^0 hybrid.” (P8-9, L153-162)

“For X-ray photoelectron spectroscopy (XPS) detection, *S. oneidensis*- Cu_{2-x}Se NPs were collected and washed three times with deionized water. The washed *S. oneidensis*- Cu_{2-x}Se NPs were freeze-dried and then ground for XPS (Kratos Axis supra+, Shimadzu) measurement.” (P29, L640-642)

Supplementary Figure 5. XPS results of the biogenic Cu_{2-x}Se NPs by *S. oneidensis*- Se^0 hybrid (A-C) with / (D-F) without illumination for 5 h. (A, D) XPS survey spectrum, (B, E) Cu 2p spectrum and (C, F) Se 3d spectrum.

14. Supplementary Figure 3: whether Se^0 NPs in this study are in t-Se or m-Se form?

Thanks. The result shows that Se^0 NPs in this study are in the crystalline t-Se phase (**Supplementary Fig. 2E**). The corresponding description has been added to the revised manuscript as follows:

“The X-ray powder diffraction (XRD) displays two major diffraction peaks that coincide well with the trigonal Se standard phase (COD-9008579) (Supplementary Fig. 2E)³⁹.” (P6, L100-101)

Supplementary Figure 2. Characteristics of the constructed *S. oneidensis*-Se⁰ hybrid. (A) Optical images of *S. oneidensis*-Se⁰ hybrid. (B) Average particle size of Se nanoparticles. The corresponding EDS mapping images of (C) N and (D) C elements of the biosynthesized *S. oneidensis*-Se⁰ hybrid in Figure 1B. (E) XRD patterns of the formed *S. oneidensis*-Se⁰ hybrid and standard Se (COD-9008579).

REFERENCES

1. Lampis, S. et al. Selenite biotransformation and detoxification by *Stenotrophomonas maltophilia* SeITE02: Novel clues on the route to bacterial biogenesis of selenium nanoparticles. *J. Hazard. Mater.* **324**, 3-14 (2017).
2. Tugarova, A.V., Mamchenkova, P.V., Dyatlova, Y.A. & Kamnev, A.A. FTIR and Raman spectroscopic studies of selenium nanoparticles synthesised by the bacterium *Azospirillum thiophilum*. *Spectrochim Acta A: Mol. Biomol. Spectrosc.* **192**, 458-463 (2018).
3. Tam, K. et al. Growth mechanism of amorphous selenium nanoparticles synthesized by *Shewanella* sp. HN-41. *Biosci. Biotechnol. Biochem.* **74**, 696-700 (2010).
4. Ho, C.T. et al. Biogenic synthesis of selenium nanoparticles by *Shewanella* sp. HN-41 using a modified bioelectrochemical system. *Electronic J. Biotechnol.* **54**, 1-7 (2021).
5. Beleneva, I.A. et al. Biogenic synthesis of selenium and tellurium nanoparticles by marine bacteria and their biological activity. *World J. Microbiol. Biotechnol.* **38**, 188 (2022).
6. Zhang, X., Zhong, M., Zhou, R., Qin, W. & Si, Y. Se(IV) reduction and extracellular biosynthesis of Nano-Se(0) by *Shewanella oneidensis* MR-1 and *Shewanella putrefaciens*. *Process Biochem.* **130**, 481-491 (2023).
7. Tian, L.J. et al. Directed biofabrication of nanoparticles through regulating extracellular electron transfer. *J. Am. Chem. Soc.* **139**, 12149-12152 (2017).
8. Li, D.B. et al. Selenite reduction by *Shewanella oneidensis* MR-1 is mediated by fumarate reductase in periplasm. *Sci. Rep.* **4**, 3735 (2014).
9. Qi, S. et al. Extracellular biosynthesis of Cu_{2-x}Se nanocrystallites with photocatalytic activity.

Materi. Res. Bull. **111**, 126-132 (2019).

10. Wang, X.M. et al. Biogenic copper selenide nanoparticles for near-infrared photothermal therapy application. *ACS Appl. Mater. Interfaces* **15**, 27638-27646 (2023).
11. Wang, X.M. et al. AQDS activates extracellular synergistic biodegradation of copper and selenite via altering the coordination environment of outer-membrane proteins. *Environ. Sci. Technol.* **56**, 13786-13797 (2022).
12. Wang, X.-M. et al. Highly efficient near-infrared photothermal antibacterial membrane with incorporated biogenic CuSe nanoparticles. *Chem. Eng. J.* **405** (2021).
13. Han, H.X. et al. Reversing electron transfer chain for light-driven hydrogen production in biotic-abiotic hybrid systems. *J. Am. Chem. Soc.* **144**, 6434-6441 (2022).
14. Guo, J. et al. Light-driven fine chemical production in yeast biohybrids. *Science* **362**, 813-816 (2018).
15. Zhang, H. et al. Bacteria photosensitized by intracellular gold nanoclusters for solar fuel production. *Nat. Nanotechnol.* **13**, 900-905 (2018).
16. Sakimoto, K.K., Wong, A.B. & Yang, P. Self-photosensitization of nonphotosynthetic bacteria for solar-to-chemical production. *Science* **351**, 74-77 (2016).
17. Lin, Y. et al. Periplasmic biomineralization for semi-artificial photosynthesis. *Sci. Adv.* **9**, eadg5858 (2023).
18. Ye, J. et al. Solar-driven methanogenesis with ultrahigh selectivity by turning down H₂ production at biotic-abiotic interface. *Nat. Commun.* **13**, 6612 (2022).
19. Wei, W. et al. A surface-display biohybrid approach to light-driven hydrogen production in air. *Sci. Adv.* **4**, eaap9253 (2018).
20. Martins, M., Toste, C. & Pereira, I.A.C. Enhanced light-driven hydrogen production by self-photosensitized biohybrid systems. *Angew Chem. Int. Ed. Engl.* **60**, 9055-9062 (2021).
21. Luo, B.F. et al. A periplasmic photosensitized biohybrid system for solar hydrogen production. *Adv. Energy Mater.* **11**, 2100256 (2021).
22. Shen, H. et al. A whole-cell inorganic-biohybrid system integrated by reduced graphene oxide for boosting solar hydrogen production. *ACS Catal.* **10**, 13290-13295 (2020).
23. Qin, Y. et al. Dyeable PAN/CuS nanofiber membranes with excellent mechanical and photothermal conversion properties via electrospinning. *ACS Appl. Polym. Mater.* **4**, 9144-9150 (2022).
24. Zhang, S. et al. Vacancy engineering of Cu_{2-x}Se nanoparticles with tunable LSPR and magnetism for dual-modal imaging guided photothermal therapy of cancer. *Nanoscale* **10**, 3130-3143 (2018).
25. Cheng, H. et al. Tailoring core@shell structure of Cu_{2-x}Se@PDAs for synergistic solar-driven water evaporation. *J. Mater. Sci.* **57**, 11725-11734 (2022).
26. Xia, W., Cheng, H., Zhou, S., Yu, N. & Hu, H. Synergy of copper selenide/MXenes composite with enhanced solar-driven water evaporation and seawater desalination. *J. Colloid. Interface Sci.* **625**, 289-296 (2022).
27. Tao, F. et al. A plasmonic interfacial evaporator for high-efficiency solar vapor generation. *Sustain. Energy Fuels* **2**, 2762-2769 (2018).
28. Tao, F. et al. Copper sulfide-based plasmonic photothermal membrane for high-efficiency solar vapor generation. *ACS Appl. Mater. Interfaces* **10**, 35154-35163 (2018).
29. Guo, Z. et al. Super-hydrophilic copper sulfide films as light absorbers for efficient solar steam generation under one sun illumination. *Semicond. Sci. Technol.* **33**, 025008 (2018).

30. Shang, M. et al. Full-spectrum solar-to-heat conversion membrane with interfacial plasmonic heating ability for high-efficiency desalination of seawater. *ACS Appl. Energy Mater.* **1**, 56-61 (2017).
31. Wang, Z., Zhang, X.F., Shu, L. & Yao, J. Copper sulfide integrated functional cellulose hydrogel for efficient solar water purification. *Carbohydr. Polym.* **319**, 121161 (2023).
32. Chen, J. et al. Photothermal membrane of CuS/polyacrylamide–carboxymethyl cellulose for solar evaporation. *ACS Appl. Polym. Mater.* **3**, 2402-2410 (2021).
33. Fan, G., Dundas, C.M., Graham, A.J., Lynd, N.A. & Keitz, B.K. *Shewanella oneidensis* as a living electrode for controlled radical polymerization. *Proc. Natl. Acad. Sci. U. S. A.* **115**, 4559-4564 (2018).
34. Shi, L. et al. Extracellular electron transfer mechanisms between microorganisms and minerals. *Nat. Rev. Microbiol.* **14**, 651-662 (2016).
35. Razaqpur, A.G., Wang, Y., Liao, X., Liao, Y. & Wang, R. Progress of photothermal membrane distillation for decentralized desalination: A review. *Water Res.* **201**, 117299 (2021).
36. Arnold, M., Batista, J., Dickenson, E. & Gerrity, D. Use of ozone-biofiltration for bulk organic removal and disinfection byproduct mitigation in potable reuse applications. *Chemosphere* **202**, 228-237 (2018).
37. Jeong, Y. et al. Roles of heterojunction and Cu vacancies in the Au@Cu_{2-x}Se for the enhancement of electrochemical nitrogen reduction performance. *ACS Appl. Mater. Interfaces* **15**, 52342–52357 (2023).
38. Liu, S., Wang, R., Wang, Q., Tian, Q. & Cui, X. A facile synthesis of Ni_{0.85}Se@Cu_{2-x}Se nanorods as high-performance supercapacitor electrode materials. *Dalton Trans.* **50**, 13543-13553 (2021).
39. Ruiz Fresneda, M.A. et al. Green synthesis and biotransformation of amorphous Se nanospheres to trigonal 1D Se nanostructures: impact on Se mobility within the concept of radioactive waste disposal. *Environ. Sci.: Nano* **5**, 2103-2116 (2018).

Response to Reviewer 3's comments

Comments: The paper presents a study on the biosynthesis of Cu_{2-x}Se nanoparticles by the *Shewanella oneidensis* MR-1. This bacteria was used to reduce Cu ions on Se^0 nanoparticles, resulting in Cu_{2-x}Se nanoparticles. These were then used for solar-to-vapor generation. The authors aimed to investigate the underlying chemical synthesis mechanism under photon induction. The topic is interesting and the study is comprehensive. However, after a quick search from the literature, there are several similar reports already available (Scientific Reports volume 9, Article number: 7589(2019), *J. Am. Chem. Soc.* 2017, 139, 35, 12149-12152). Additionally, the manuscript's presentation should be improved (see my remarks below).

Thanks for the reviewer's constructive suggestion. This work substantially differed from the reported work (*Sci. Rep.* 2019, 9 (1), 7589) and our previous work (*J. Am. Chem. Soc.* 2017, 139, 12149-12152) in several aspects:

- 1) Indeed, the biological synthesis of Se^0 NPs has been realized by various microorganisms (**Supplementary Table 6**). Previous works mainly focused on the **synthetic mechanisms and regulatory strategies** of biogenic Se^0 NPs, including our previous work (*J. Am. Chem. Soc.* 2017, 139, 12149-12152). However, the **in vivo application of biogenic Se^0 NPs, that couple the functions of bacteria and nanoparticles**, has not been reported. Motivated by the photocatalytic function of Se^0 NPs, the present study constructed *S. oneidensis*- Se^0 hybrid for photo-boosted biological functions. *S. oneidensis* MR-1 possesses powerful nanoparticle assembly capability and diverse respiratory reductases, making it an excellent candidate for exploring *S. oneidensis*- Se^0 hybrid system. This work reveals a previously unrecognized function of Se^0 NPs in terms of accelerating bacterial functions through reversing Mtr pathway, which is initiated by light illumination. With mutant analyses, we uncover the underlying synergistic mechanisms and elucidate the corresponding electronic circuits.
- 2) Regarding utilizing photoelectrons for nanoparticle bio-assembly, several pure organisms have been demonstrated (**Supplementary Table 9**), which relied on the **inherently photosensitive groups**. Thus, such light-induced nanoparticle formation was generally restricted in a particular organism. Furthermore, the detailed mechanisms of related proteins and extracellular polymeric substances (EPS) remain to be further investigated. Unlike the intrinsic light-use capabilities of specific organisms, in this work, non-photosynthetic bacteria are endowed with advanced light-energy utilization capabilities by in vivo synthesizing Se^0 semiconductor, creating an **unnatural photoelectronic pathway** for Cu_{2-x}Se NPs formation. This hybrid system provides a convenient and universally applicable approach to direct solar energy toward nanoparticle synthesis, and beyond the natural capabilities of living organisms

Accepting the reviewer's suggestion, we have updated the **Discussion** section to highlight the novelty of this work as follows:

“Indeed, the biological synthesis of Se^0 NPs has been realized by various microorganisms (Supplementary Table 6), which mainly focused on the synthetic mechanisms and regulatory strategies. However, the in vivo application of biogenic Se^0 NPs, that couple the functions of bacteria and nanoparticles, has yet to be reported. Motivated by the photocatalytic function of Se^0 NPs, the present study constructed *S. oneidensis*- Se^0 hybrid for photo-boosted biological functions. *S. oneidensis* MR-1 possesses powerful nanoparticle assembly capability and diverse respiratory reductases, making it

an excellent candidate for exploring *S. oneidensis*-Se⁰ hybrid system. This work reveals a previously unrecognized function of Se⁰ NPs in terms of accelerating bacterial functions through reversing MtrABC, which is initiated by light illumination. With mutant analyses, we uncover the underlying synergistic mechanisms and elucidate the corresponding electronic circuits.” (P20-21, L433-443)

“So far, utilizing solar energy for the fabrication of nanoparticles has been demonstrated in several pure organisms (Supplementary Table 9). The proposed underlying mechanisms relied on the inherently photosensitive groups, implying that such light-induced nanoparticle formation was generally restricted in a particular organism. Furthermore, the detailed mechanisms of related proteins and extracellular polymeric substances remain to be further investigated. Unlike the intrinsic light-use capabilities of specific organisms, in this work, non-photosynthetic bacteria are endowed with advanced light-energy utilization capabilities by in vivo synthesizing Se⁰ semiconductor, creating an unnatural photoelectronic pathway for Cu_{2-x}Se NPs formation. This hybrid system provides a convenient and universally applicable approach to direct solar energy toward nanoparticle synthesis, and beyond the natural capabilities of living organisms. Undoubtedly, the constructed *S. oneidensis*-Se⁰ hybrid is a promising and efficient solar-to-chemical technology, that reasonably combines the light-harvesting of inorganic semiconductors with the specific catalytic power of organisms. Such functional synergy could not only be important for the biosynthesis of living materials, but also contribute to new strategies for solar-to-high-value products.” (P23-24, L493-507)

Supplementary Table 9. Photo-induced biosynthesis of nanoparticles

Organism	Mechanism for photo-induced biosynthesis	Nanoparticles synthesis	Ref.
S. oneidensis MR-1	Certain active groups and extracellular polymeric substances (EPS)	Gold NPs	1
Eucalyptus	-	Gold NPs	2
Lolium perenne	-	CdS _x Se _{1-x} quantum dots	3
Chlorella	Electrons from photosynthetic chain	Gold NPs	4
Phormidium ambiguum Desertifilum tharense	-	Silver NPs	5
S. oneidensis MR-1	Bacterium uses photogenerated electrons from illuminated Se ⁰ semiconductor	Cu _{2-x} Se NPs	This work

Supplementary Table 6. Mechanism, regulation strategies and applications of biogenic Se⁰ nanoparticles

Organism	Mechanism	Regulation strategies	Applications	Ref.
Stenotrophomonas maltophilia SeITE02	Alcohol dehydrogenase homolog	-	-	6
Azospirillum thiophilum	Structural and compositional properties of Se ⁰ NPs.	-	-	7
Shewanella sp. HN-41	-	Reaction time, biomass, selenite concentration	-	8
Shewanella sp. HN-41	-	Reaction time and biomass	-	9
Pseudoalteromonas shioyasakiensis	-	-	Antimicrobial, antifouling and cytotoxic activities	10
S. oneidensis MR-1 S. putrefaciens	-	Adding riboflavin	-	11
S. oneidensis MR-1	-	Regulate extracellular electron transfer	-	12
S. oneidensis MR-1	Fumarate reductase FccA	-	-	13
S. oneidensis MR-1	-	-	S. oneidensis -Se ⁰ hybrid for solar energy conversion	This work

Remarks:

1. There is a contradiction between line 151, which states that "Cu_{2-x}Se NPs synthesis were mediated by microorganisms rather than a spontaneous abiotic reaction," and line 160, which suggests that "Cu_{2-x}Se NPs might be formed through a spontaneous abiotic reaction." The authors should clarify why there is an opposite conclusion for Cu (I) and Cu (II).

Thanks. After a detailed re-examination, we have found the opposite conclusion for Cu (I) and Cu (II) was correct. The resulting conclusion was attributed to the two coupled steps in the biosynthetic procedures of Cu_{2-x}Se NPs, including Cu(II)-to-Cu(I) and Cu(I)-to-Cu_{2-x}Se NPs. The first step is the biological reduction of Cu(II), which leads to the accumulation of Cu(I), the precursor for further Cu_{2-x}Se NPs formation. Subsequently, the resulting Cu(I) reacts with Se⁰ to form Cu_{2-x}Se NPs, which is a spontaneous reaction. Such abiotic conversion can be accelerated by the photoactivity of Se⁰ semiconductor. Thus, Cu_{2-x}Se NPs synthesis is mediated by microorganisms rather than a spontaneous abiotic reaction, as the key intermediate Cu(I) can only be formed by biocatalysts (previous manuscript, line 151). Meanwhile, Cu_{2-x}Se NPs may be formed through a spontaneous abiotic reaction between Se⁰ NPs and Cu(I), which can be accelerated by the photoactivity of Se⁰ semiconductor (previous manuscript, line 160).

Accepting the reviewer's suggestion, we have updated the corresponding section to clarify the two steps that are coupled in the Cu_{2-x}Se NPs formation as follows:

“Copper precipitation and Cu_{2-x}Se NPs production were almost negligible in the system that co-incubated lysed hybrid with Cu(II) (Fig. 3B-3C), further confirming the reaction between Se⁰ NPs and Cu(II) to form Cu_{2-x}Se NPs in *S. oneidensis*-Se⁰ hybrid system were mediated by microorganisms. In contrast, the inactivated hybrid co-incubated with Cu(I) showed apparent copper conversion and Cu_{2-x}Se production, supported by the enhanced Cu precipitation and raised ΔT (Fig. 3B-3C). Moreover, such a procedure was strengthened by light exposure. These results demonstrate that Cu_{2-x}Se NPs might be formed through a spontaneous abiotic reaction between Se⁰ NPs and Cu(I), which could be accelerated by the photoactivity of Se⁰ semiconductor. Thus, *S. oneidensis* was speculated to be involved in Cu(II) reduction during Cu_{2-x}Se NPs photocatalytic synthesis. Consistently, the intermediate Cu(I) was detected in the supernatant and increased continuously over time after injecting Cu(II) into the hybrid system with sodium lactic acid as the carbon source (Supplementary Fig. 6A), suggesting that *S. oneidensis* has the capacity to reduce Cu(II) to Cu(I). Similarly, previous works reported that Cu(II) could be reduced by MR-1, and the resulting Cu(I) served as a catalyst for atom transfer radical polymerization¹⁴⁻¹⁶. Together, there are two steps that are coupled in the biosynthetic procedures of Cu_{2-x}Se NPs, including Cu(II)-to-Cu(I) and Cu(I)-to-Cu_{2-x}Se NPs. The first step is the biological reduction of Cu(II), which leads to the accumulation of intermediate Cu(I), the precursor for further Cu_{2-x}Se NPs formation. Subsequently, the resulting Cu(I) reacts with Se⁰ NPs, undergoing abiotic conversion to synthesize Cu_{2-x}Se NPs, which could be stimulated by light irradiation.” (P9-10, L178-196)

2. Figure 3, which illustrates the mechanism of light-driven biosynthesis of Cu_{2-x}Se, should be the main focus of this manuscript. However, its presentation and explanation need substantial improvement. For instance, the "ΔmtrF" arrow is missing in Figure 3A. In Figure 3G, the "ΔT" of "ΔomcAΔmtrC" is quite comparable to "ΔomcAΔmtrC" in Figure 3E, but a detailed discussion is missing in the main text.

Accepting the reviewer's suggestion, **Figure 3A** has been revised. In the presence/absence of

illumination, the comparable " ΔT " of $\Delta omcA\Delta mtrC$ is reasonable, and the corresponding detailed discussion has been added to the revised manuscript as follows:

“Interestingly, photo-induced $Cu_{2-x}Se$ NPs formation was not observed in $\Delta omcA\Delta mtrC$ strain, supported by a comparable ΔT and precipitated Cu content in $\Delta omcA\Delta mtrC$ under light illumination and dark conditions (Fig. 3D-3G). Consistent with the above-mentioned results, MtrC- and OmcA-composed electron conduit only plays a crucial role in photoelectron delivery but does not participate in metabolic electron transport for $Cu_{2-x}Se$ NPs formation. Thus, removing MtrC and OmcA proteins would eliminate the light-stimulated synthesis, but does not affect metabolic electron-mediated transformation. The remaining $Cu_{2-x}Se$ synthesis capacity by $\Delta omcA\Delta mtrC$ strain with/without light illumination was attributed to the metabolically driven synthesis. Furthermore, this similar result further indicates that the MtrC- and OmcA-composed conduit is the main photoelectron transport pathway under light illumination.” (P13, L252-261)

Fig. 3 | Mechanism of the light-driven $Cu_{2-x}Se$ NPs assembly by the *S. oneidensis*- Se^0 hybrid. (A) Schematic diagram shows the synthesis of $Cu_{2-x}Se$ by lysed hybrids that were co-incubated with different strains and Cu(II) with/without illumination. (B) The precipitated copper concentration and (C) temperature rise (ΔT) of the lysed hybrids that were co-incubated with Cu(II) or Cu(I) for 5 h

with/without illumination. **(D)** The precipitated copper concentration and **(E)** (ΔT) of the lysed hybrids that were co-incubated with different strains and Cu(II) for 5 h under dark conditions. **(F)** The precipitated copper concentration and **(G)** (ΔT) of the lysed hybrids that were co-incubated with different strains and Cu(II) for 5 h under light illumination. The above experiments were performed in the mineral salt medium with 20 mM sodium lactate. The data points represent means \pm SD ($n = 6$). Error bars correspond to standard deviations. “****” represents $p < 0.0001$.

3. The authors should clarify the advantage of their proposed method over the traditional wet chemistry approach, especially if they are only discussing the photo-to-vapor applications where the Cu_{2-x}Se NPs are encapsulated in PVDF.

Accepting the reviewer’s constructive suggestion, we have updated the **Table and Discussion** sections to highlight the novelty of $\text{Cu}_{2-x}\text{Se}@MR-1@PVDF$ membrane as follows:

“To date, various Cu-composed materials have been employed to construct solar steam generators and have achieved encouraging performance (Supplementary Table 5). Most of the reported works focus on traditional chemical synthesis methods, involving aggressive chemical agents, harsh synthesis conditions, intensive energy consumption, and complex synthesis processes (Supplementary Table 5). Besides, the anchored photothermal material can detach from the supporting material during long-term irradiation or high-frequency use, resulting in poor durability¹⁷. Here, the as-prepared $\text{Bio-Cu}_{2-x}\text{Se}@MR-1@PVDF$ was prepared at room temperature using environmentally benign reagents, providing a more sustainable manufacturing strategy. Additionally, biomass, as a natural linker, enhances the adhesion between the photothermal conversion material and the supporting material. Compared with the typical photothermal conversion materials (Supplementary Table 4 and 5), the water evaporation conversion efficiency of the resulting membrane is top-ranking. Thus, the $\text{Bio-Cu}_{2-x}\text{Se}@MR-1@PVDF$ has the advantages of being inexpensive, sustainable, simple to fabricate, highly stable, and highly efficient in water evaporation, meeting the practical requirements for solar vapor generation.” **(P25-26, L537-551)**

Supplementary Table 5. Comparison of solar steam device based on Cu-composed nanoparticles

Supporting Material	Absorber	Synthesis of Absorber	Synthesis Conditions of Absorber	Evaporation Rate (kg m ⁻² h ⁻¹)	Conversion Efficiency (%)	Membrane Temperature (°C)	Ref.
Filter paper	Cu _{2-x} Se@polydopamine	-	Using PVP, ascorbic acid, dopamine, H ₂ O ₂	2.71	-	-	18
Glass microfiber	Cu _{2-x} Se/Nb ₂ CT _x	-	Using hydrazine, PVP, H ₂ O ₂ , HF aqueous solution, over one week	1.2	-	39.7	19
PVDFM	Cu ₉ S ₅	Hydrothermal	180 °C for several hours	1.173	80.2 ± 0.6	36.1	20
PVDFM	CuS	Hydrothermal	180 °C for 18 h	1.43	90.4	38.5	21
MCE	CuS	Hydrothermal	140 °C for 12 h	1.12	80±2.5	42.8	22
SCM	CuS nanoflowers	Hydrothermal	120 °C for 18 h	1.09	68.6	35.2	17
Polyethylene	CuS	Hydrothermal	180 °C for 12 h	1.021	63.9	37.6	23
Cellulose hydrogel	CuS	Hydrothermal	120 °C for 12 h	2.2	87	-	24
PAAm-CMC	CuS	Hydrothermal	180 °C for 12 h	1.613	79	47.2	25
PVDFM	Cu _{2-x} Se	Photo-facilitated biosynthesis	Room temperature	1.44	90.55	41.2	This work

The above photothermal distillation membrane experiments were performed under 1 sun irradiation; PVDFM refers to the poly(vinylidene fluoride) membrane. PVP refers to poly(vinyl pyrrolidone). MCE refers to mixed cellulose ester membrane. SCM refers to semipermeable collodion membrane. PAAm-CMC refers to polyacrylamide (PAAm) and carboxymethyl cellulose (CMC).

4. The authors should explain why the Cu_{2-x}Se NPs synthesized by *S. oneidensis* have a higher conversion efficiency (Figure 5G) compared to the " ΔT " (below 20 degrees C, which is not very impressive) as shown in Figure 3.

Thanks. The variation comes from the different detection conditions, including the content of photothermal material, the light region and intensity of solar energy and the different calculation methods. The photothermal effect of Bio- Cu_{2-x}Se in solution (Figure 3) was characterized using a 1064 nm NIR laser (Inter-Diff Co., China) with a power density of 1 W/cm². After 6 min irradiation, the temperature increase (ΔT) was recorded. The content of Cu_{2-x}Se is very low, leading to a small ΔT . For the photothermal conversion efficiency of $\text{Cu}_{2-x}\text{Se}@MR-1@PVDF$ membrane (Figure 5G), a desktop Xenon lamp (PLS-SXE300D, Beijing Pophile Technology Co., Ltd, China) with 1 sun light intensity was utilized to measure the surface temperature of the membrane in the photothermal distillation membrane experiment. Meanwhile, the amount of incorporated photothermal materials in the membrane is much higher than that in the Bio- Cu_{2-x}Se aqueous solution.

5. The mechanism proposed in this paper is quite similar to those reported in the literature. It might be more suitable for publication in a chemistry-related journal rather than Nature Communications.

Thanks. The novelty of this work has been highlighted by comparing this work with the reported works about Se^0 NPs biosynthesis, Cu_{2-x}Se NPs biosynthesis, biotic-abiotic hybrids and water desalination (**Table R1**). The substantial difference between this work and the reported works has been elucidated by providing more convincing evidence and deepening the discussion section.

Here we report **several groundbreaking findings that may fundamentally change our perception of biotic-abiotic photocatalytic systems and hopefully open up a promising new horizon in this field.** 1) Solar energy was harvested in a **dual mode** for Cu_{2-x}Se NPs biosynthesis and seawater desalination by integrating the functionality synergism between bacterium and nanoparticles. 2) Photoelectrons generated by extracellular Se^0 NPs wirelessly activate Cu_{2-x}Se synthesis through a **dual catalytic network** located in periplasm and extracellular space, respectively. 3) Under visible light, periplasmic Cu(II) reduction is initiated by **two electron fluxes**, either intracellular metabolic electron or extracellular photoelectron. 4) The unique photothermal feature of the photosynthetic Cu_{2-x}Se NPs and the natural hydrophilic and linking properties of bacterium offers a convenient way to tailor photothermal membranes for **solar water production**.

Furthermore, The structural, chemical and performance stabilities of the Bio- $\text{Cu}_{2-x}\text{Se}@MR-1@PVDF$ membrane have been investigated by monitoring TOC content and protein concentration in solution, as well as the valence states of Cu in the membrane before and after 1 sun irradiation for 10 h. Encouragingly, the results rule out potential hazards from the degradation of MR-1 dead cells and prove the excellent stability of Bio- $\text{Cu}_{2-x}\text{Se}@MR-1@PVDF$. Impressively, compared with a chemically synthesized membrane (Chem- $\text{Cu}_{2-x}\text{Se}@MR-1@PVDF$), Bio- Cu_{2-x}Se NPs not only have better hydrophilicity and photothermal distillation performance, but also have excellent structural, chemical and performance stabilities (**Supplementary Figure 12-16**).

The data interpretation and literature discussion have been strengthened. We believe that the reviewers' concerns have been adequately addressed and the quality of the manuscript has been substantially improved to meet the high-standards of *Nature Communications*.

Table R1. Summary of the differences between recent studies and this work

Topic	Previous studies	This work	Table Panels
Biogenic Se ⁰	Synthetic mechanisms and regulatory strategies	In vivo application	Supplementary Table 6
Biogenic Cu ₂ - _x Se NPs	Metabolic electron-driven	Metabolic electron & photoelectron -driven	Supplementary Table 7
	Regulating culture conditions	Sunlight -boost (simple and sustainable)	
	Formed in periplasm	Extracellular assembly	
	Normal Mtr pathway	Reversed Mtr pathway	
Reversed EET	For nitrate, fumarate, trypan blue reduction	For nanoparticles assembly	Supplementary Table 8
Photo-induced NPs biosynthesis	Inherent function (specific organisms)	Creating an unnatural photoelectronic pathway	Supplementary Table 9
Biotic-abiotic hybrid	Photoelectron transport along single pathway	Dual catalytic network	Supplementary Table 10
	Single site for photoelectron utilization	Two site for photoelectron utilization	
	For organic substance & H ₂ synthesis	For nanoparticles assembly	
Solar water production	Chemical synthesis (Harsh condition)	Biosynthesis (simple and sustainable)	Supplementary Table 4 and Table 5
	Instability	Good structural, chemical and performance stabilities (Biomass acts as natural linker)	

REFERENCES

- Huang, B.C., Yi, Y.C., Chang, J.S. & Ng, I.S. Mechanism study of photo-induced gold nanoparticles formation by *Shewanella oneidensis* MR-1. *Sci. Rep.* **9**, 7589 (2019).
- Lintern, M., Anand, R., Ryan, C. & Paterson, D. Natural gold particles in *Eucalyptus* leaves and their relevance to exploration for buried gold deposits. *Nat. Commun.* **4**, 2614 (2013).
- Tian, L.-J. et al. Bio-assembly of CdS_xSe_{1-x} quantum dots in ryegrass. *Green Chem.* **21**, 6727-6730 (2019).
- Zhu, X. et al. Photosynthesis-mediated intracellular biomineralization of gold nanoparticles inside *Chlorella* cells towards hydrogen boosting under green light. *Angew Chem. Int. Ed. Engl.*, e202308437 (2023).
- Hanna, A.L. et al. Biosynthesis and characterization of silver nanoparticles produced by phormidium ambiguum and desertifilum tharense cyanobacteria. *Bioinorg. Chem. Appl.* **2022**, 9072508 (2022).
- Lampis, S. et al. Selenite biotransformation and detoxification by *Stenotrophomonas maltophilia* SeITE02: Novel clues on the route to bacterial biogenesis of selenium nanoparticles. *J. Hazard. Mater.* **324**, 3-14 (2017).
- Tugarova, A.V., Mamchenkova, P.V., Dyatlova, Y.A. & Kamnev, A.A. FTIR and Raman

- spectroscopic studies of selenium nanoparticles synthesised by the bacterium *Azospirillum thiophilum*. *Spectrochim Acta A: Mol. Biomol. Spectrosc.* **192**, 458-463 (2018).
8. Tam, K. et al. Growth mechanism of amorphous selenium nanoparticles synthesized by *Shewanella* sp. HN-41. *Biosci. Biotechnol. Biochem.* **74**, 696-700 (2010).
 9. Ho, C.T. et al. Biogenic synthesis of selenium nanoparticles by *Shewanella* sp. HN-41 using a modified bioelectrochemical system. *Electronic J. Biotechnol.* **54**, 1-7 (2021).
 10. Beleneva, I.A. et al. Biogenic synthesis of selenium and tellurium nanoparticles by marine bacteria and their biological activity. *World J. Microbiol. Biotechnol.* **38**, 188 (2022).
 11. Zhang, X., Zhong, M., Zhou, R., Qin, W. & Si, Y. Se(IV) reduction and extracellular biosynthesis of Nano-Se(0) by *Shewanella oneidensis* MR-1 and *Shewanella putrefaciens*. *Process Biochem.* **130**, 481-491 (2023).
 12. Tian, L.J. et al. Directed biofabrication of nanoparticles through regulating extracellular electron transfer. *J. Am. Chem. Soc.* **139**, 12149-12152 (2017).
 13. Li, D.B. et al. Selenite reduction by *Shewanella oneidensis* MR-1 is mediated by fumarate reductase in periplasm. *Sci. Rep.* **4**, 3735 (2014).
 14. Fan, G., Graham, A.J., Kolli, J., Lynd, N.A. & Keitz, B.K. Aerobic radical polymerization mediated by microbial metabolism. *Nat. Chem.* **12**, 638-646 (2020).
 15. Fan, G., Dundas, C.M., Graham, A.J., Lynd, N.A. & Keitz, B.K. *Shewanella oneidensis* as a living electrode for controlled radical polymerization. *Proc. Natl. Acad. Sci. U. S. A.* **115**, 4559-4564 (2018).
 16. Kimber, R.L. et al. Biosynthesis and characterization of copper nanoparticles using *Shewanella oneidensis*: application for click chemistry. *Small* **14**, 1703145 (2018).
 17. Qin, Y. et al. Dyeable PAN/CuS nanofiber membranes with excellent mechanical and photothermal conversion properties via electrospinning. *ACS Appl. Polym. Mater.* **4**, 9144-9150 (2022).
 18. Cheng, H. et al. Tailoring core@shell structure of Cu_{2-x}Se@PDAs for synergistic solar-driven water evaporation. *J. Mater. Sci.* **57**, 11725-11734 (2022).
 19. Xia, W., Cheng, H., Zhou, S., Yu, N. & Hu, H. Synergy of copper selenide/MXenes composite with enhanced solar-driven water evaporation and seawater desalination. *J. Colloid. Interface Sci.* **625**, 289-296 (2022).
 20. Tao, F. et al. A plasmonic interfacial evaporator for high-efficiency solar vapor generation. *Sustain. Energy Fuels* **2**, 2762-2769 (2018).
 21. Tao, F. et al. Copper sulfide-based plasmonic photothermal membrane for high-efficiency solar vapor generation. *ACS Appl. Mater. Interfaces* **10**, 35154-35163 (2018).
 22. Guo, Z. et al. Super-hydrophilic copper sulfide films as light absorbers for efficient solar steam generation under one sun illumination. *Semicond. Sci. Technol.* **33**, 025008 (2018).
 23. Shang, M. et al. Full-spectrum solar-to-heat conversion membrane with interfacial plasmonic heating ability for high-efficiency desalination of seawater. *ACS Appl. Energy Mater.* **1**, 56-61 (2017).
 24. Wang, Z., Zhang, X.F., Shu, L. & Yao, J. Copper sulfide integrated functional cellulose hydrogel for efficient solar water purification. *Carbohydr. Polym.* **319**, 121161 (2023).
 25. Chen, J. et al. Photothermal membrane of CuS/polyacrylamide-carboxymethyl cellulose for solar evaporation. *ACS Appl. Polym. Mater.* **3**, 2402-2410 (2021).

Reviewers' Comments:

Reviewer #1:

Remarks to the Author:

The authors have addressed my concerns reasonably well. I do not have further comments.

Reviewer #2:

Remarks to the Author:

After reviewing the manuscript and considering the comments from other reviewers, it is clear that the work presents novel content; however, its innovative aspect and impact on a broad audience seems limited. The discovery regarding how nanoparticles are biosynthesized by bacteria function and their electronic pathways is indeed new. However, these properties could be anticipated from existing literature, and asserting that the findings indicate innovative phenomena or functionalities is challenging. Furthermore, the results of PVDF and the proposed mechanisms require more rigorous examination. The validity of the mechanisms suggested by the authors remains uncertain. Also, the high efficiency should be verified by the use of mutant strains and explained quantitatively. It would be advisable to consider publishing the initial part of the study in a chemistry-focused journal. The latter part should also have a more suitable platform once more data for the mechanism.

Reviewer #3:

Remarks to the Author:

Dear Editor,

Significant improvements have been made on the revised manuscript, and the majority of the reviewers' comments have been addressed. After careful consideration of the revised manuscript, I recommend its publication in Nature Communications.

Response to Reviewer 1's comments

Recommendation: The authors have addressed my concerns reasonably well. I do not have further comments.

Thanks.

Response to Reviewer 2's comments

Comments/Overview: After reviewing the manuscript and considering the comments from other reviewers, it is clear that the work presents novel content; however, its innovative aspect and impact on a broad audience seem limited. The discovery regarding how nanoparticles are biosynthesized by bacteria function and their electronic pathways is indeed new. However, these properties could be anticipated from existing literature, and asserting that the findings indicate innovative phenomena or functionalities is challenging. Furthermore, the results of PVDF and the proposed mechanisms require more rigorous examination. The validity of the mechanisms suggested by the authors remains uncertain. Also, the high efficiency should be verified by the use of mutant strains and explained quantitatively. It would be advisable to consider publishing the initial part of the study in a chemistry-focused journal. The latter part should also have a more suitable platform once more data for the mechanism.

Thanks for recognizing the novelty of our manuscript. Following the review's valuable and constructive suggestions, we have conducted additional experiments to improve our manuscript.

The major modifications include:

- 1) We have detected the photoactivity, UV-vis DRS and UV-vis-NIR spectra of the biogenic Cu_{2-x}Se NPs to uncover the working mechanism of Bio- $\text{Cu}_{2-x}\text{Se}@MR-1@PVDF$ membrane in the water desalination process.
- 2) To explore whether the high efficiency of Bio- $\text{Cu}_{2-x}\text{Se}@MR-1@PVDF$ membrane was attributed to the biogenic Cu_{2-x}Se NPs, we employed ΔcymA to synthesize Cu_{2-x}Se NPs, which were then encapsulated in PVDF substrate to fabricate a solar steam generation device

Comments:

1. After reviewing the manuscript and considering the comments from other reviewers, it is clear that the work presents novel content; however, its innovative aspect and impact on a broad audience seem limited.

Thanks. The novelty and essential implications for the broad fields of this work have been highlighted by deepening the discussion section. Briefly, here we report **several groundbreaking** findings that may fundamentally change our perception of **biomineralization, solar energy harvesting, biotic-abiotic hybrid, seawater desalination, green chemistry and bio-catalysis** (Figure R1, Table R1), making it suitable for publication in Nature Communications and have potential to open up new frontiers in these fields.

- (1) Solar energy was harvested in a **dual mode** for Cu_{2-x}Se NPs biosynthesis and seawater desalination by integrating the functionality synergism between bacteria and nanoparticles, diversifying **solar energy conversion** products.
- (2) Photoelectrons generated by extracellular Se^0 NPs wirelessly activate Cu_{2-x}Se synthesis through a **dual catalytic network** located in the periplasm and extracellular space, respectively. This is a groundbreaking advancement for utilizing inward photoelectrons for **biomineralization**.
- (3) Under visible light, periplasmic Cu(II) reduction is initiated by **two electron fluxes**, either intracellular metabolic electron or extracellular photoelectron, clarifying the underlying **electron transfer network**.

- (4) More importantly, Cu_{2-x}Se NPs biosynthesis is just a model of photo-driven bio-hybrid systems. Such an *S. oneidensis*- Se^0 hybrid can be extended to nitrate reduction (Supplementary Fig. 8) and HgSe synthesis (Supplementary Fig. 9), demonstrating its **broad applicability**.
- (5) The unique photothermal feature of the photosynthetic Cu_{2-x}Se NPs, combined with the natural hydrophilic and linking properties of bacteria, provides a convenient and sustainable way to tailor photothermal membranes for **solar water production**. These membranes exhibit **good structural, chemical and performance stability**.

Figure R1. Summary of the broad impact of this work

Table R1. Summary of the differences between recent studies and this work

Topic	Previous studies	This work	Table Panels
Biogenic Se^0	Synthetic mechanisms and regulatory strategies	In vivo application	Supplementary Table 6
Biogenic Cu_{2-x}Se NPs	Metabolic electron-driven	Metabolic electron & photoelectron -driven	Supplementary Table 7
	Regulating culture conditions	Sunlight -boost (simple and sustainable)	
	Formed in periplasm Normal Mtr pathway	Extracellular assembly Reversed Mtr pathway	
Reversed EET	For nitrate, fumarate, trypan blue reduction	For nanoparticles assembly	Supplementary Table 8
Photo-induced NPs biosynthesis	Inherent function (specific organisms)	Creating an unnatural photoelectronic pathway	Supplementary Table 9
Biotic-abiotic hybrid	Photoelectron transport along single pathway	Dual catalytic network	Supplementary Table 10
	Single site for photoelectron utilization	Two site for photoelectron utilization	

	For organic substance & H ₂ synthesis	For nanoparticles assembly	
Solar water production	Chemical synthesis (Harsh condition)	Biosynthesis (simple and sustainable)	Supplementary Table 4 and Table 5
	Instability	Good structural, chemical and performance stabilities (Biomass acts as a natural linker)	

2. The discovery regarding how nanoparticles are biosynthesized by bacteria function and their electronic pathways is indeed new. However, these properties could be anticipated from existing literature, and asserting that the findings indicate innovative phenomena or functionalities is challenging.

Thanks for recognizing the novelty of Cu_{2-x}Se NPs biosynthesis. To the best of our knowledge, it is difficult to anticipate from existing literature how bacteria biosynthesize nanoparticles and their electronic pathways.

(1) **In terms of sunlight-initiated reversed EET.** Indeed, the biological synthesis of Se⁰ NPs has been realized by various microorganisms (**Supplementary Table 6**), which mainly focused on the synthetic mechanisms and regulatory strategies. However, the in vivo application of biogenic Se⁰ NPs, that couple the functions of bacteria and nanoparticles, has not been reported. Therefore, it's challenging to determine whether Se⁰ NPs can accelerate bacterial functions by reversing EET, which is initiated by light illumination. We uncover the underlying synergistic mechanisms through mutant analyses and elucidate the corresponding electronic circuits.

(2) **Regarding the biosynthesis of Cu_{2-x}Se NPs.** Previous studies have demonstrated that various microorganisms can successfully assemble Cu_{2-x}Se NPs using Na₂SeO₃ as Se source. It has been proposed that Cu_{2-x}Se NPs were primarily formed in the cell periplasm and then excreted out (**Supplementary Table 7**). Meanwhile, Cu(II) reduction mechanisms in *S. oneidensis* MR-1 are not well understood^{1,2}. Consequently, it is challenging to determine whether the extracellular Se⁰ NPs can be used for Cu_{2-x}Se NPs biosynthesis with Cu(II) as a precursor, as well as the role of *S. oneidensis* in this process. We find that photoelectrons generated by extracellular Se⁰ NPs wirelessly activate Cu_{2-x}Se NPs synthesis through a dual catalytic network located in the periplasm and extracellular space, respectively. Such a dual catalytic network is a novel paradigm for balancing the source and sink of photoelectrons.

3. Furthermore, the results of PVDF and the proposed mechanisms require more rigorous examination. The validity of the mechanisms suggested by the authors remains uncertain.

Thanks for the reviewer's constructive suggestion. We have detected the photoactivity, UV-vis DRS and UV-vis-NIR spectra of the biogenic Cu_{2-x}Se NPs to uncover the working mechanism of Bio-Cu_{2-x}Se@MR-1@PVDF membrane in the water desalination process. Our results show that biogenic Cu_{2-x}Se NPs have broad solar energy absorption and LSPR properties (**Supplementary Figure 18**), making Bio-Cu_{2-x}Se@MR-1@PVDF membrane well-suited for water desalination. Accepting the reviewer's suggestion, we have updated the mechanisms related to the photothermal distillation membrane in this work as follows:

“To uncover the working mechanism of Bio-Cu_{2-x}Se@MR-1@PVDF membrane in the water desalination process, we detected the UV-vis DRS, photoactivity, and UV-vis-NIR spectra of the

biogenic Cu_{2-x}Se NPs. UV-NIS-DRS spectrum displays that biogenic Cu_{2-x}Se NPs exhibit strong light absorption in the wavelength range of 250-2500 nm, which forms the basis for good performance in photothermal distillation (Supplementary Fig. 18A). The photoactivity of the purified Cu_{2-x}Se NPs was evaluated by measuring the photocurrent in the presence and absence of light. An observable photocurrent was produced when the nanoparticles were exposed to light, and it returned to baseline levels after the light was turned off (Supplementary Fig. 18B). No photocurrent was detected in the control group without nanoparticles. These results suggest biogenic Cu_{2-x}Se NPs act as semiconductors and can generate electron-hole pairs upon exposure to light. Recent studies have found that the free carrier (hole) generated in Cu_{2-x}Se semiconductor can drive thermalization process, resulting in the generation of photogenerated heat. Meanwhile, the free carrier is response for NIR plasmonic absorption³. Thus, we tested the LSPR property of biogenic Cu_{2-x}Se semiconductor by detecting UV-vis-NIR spectrum of the purified Cu_{2-x}Se NPs. As expected, we observed apparent absorption in the near infrared region of the purified Cu_{2-x}Se NPs (Supplementary Fig. 18C). Overall, biogenic Cu_{2-x}Se NPs exhibit broad solar energy absorption and LSPR properties, making Bio- $\text{Cu}_{2-x}\text{Se}@MR-1@PVDF$ membrane well-suited for water desalination.

Accordingly, a schematic diagram of the working mechanism of Bio- $\text{Cu}_{2-x}\text{Se}@MR-1@PVDF$ membrane in the water desalination process is plotted in Supplementary Fig. 19. Once the light is turned on, Cu_{2-x}Se NPs are excited and generate electron-hole pairs. Meanwhile, Cu_{2-x}Se NPs have local surface plasmon resonance (LSPR) properties, in which free carriers (hole) can collectively resonate with incident photons and produce hot electrons. By incorporating Cu_{2-x}Se NPs and biomass into the PVDF membrane, Bio- $\text{Cu}_{2-x}\text{Se}@MR-1@PVDF$ can achieve an apparent increase in surface temperature. Simultaneously, the hydrophilic nature of bacterial cells and the porous structure of PVDF allow efficient water transport to the interface, allowing water to be efficiently transported to the thermally located surface. The excellent evaporation performance should be assigned to the synergistic effect between the good photothermal conversion efficiency of Cu_{2-x}Se NPs, the high hydrophilicity of biomass, the microchannel PVDF membrane, and the superior stability properties of Bio- $\text{Cu}_{2-x}\text{Se}@MR-1@PVDF$.” (P20-21, L427-456)

“Purification of Biogenic Cu_{2-x}Se NPs

S. oneidensis- Cu_{2-x}Se samples were collected by centrifugation and washed three times with ultrapure water. After that, the washed samples were resuspended in 2% sodium dodecyl sulfate (SDS) and then soaked in a 55°C water bath for 1 h. The precipitate was collected by centrifugation and washed three times with ultrapure water. Subsequently, the precipitate was resuspended in water and treated with ultrasonic disruption (200 W, 4 s ultrasound with 6 s intervals, 200 repeats) to fragment the bacterium. The fragmented samples were centrifugated to obtain the supernatant solution. The acquired supernatant solution was further digested with proteinase K (100 $\mu\text{g}/\text{mL}$) at 55 °C for 0.5 h. The digested solution was centrifugated and washed three times to obtain purified Cu_{2-x}Se NPs. The purified Cu_{2-x}Se NPs were used for UV-vis-NIR measurement.” (SI, P8, L164-174)

Supplementary Figure 18. Properties of the biogenic Cu_{2-x}Se NPs. (A) UV-vis DRS spectra of the *S. oneidensis*- Cu_{2-x}Se and *S. oneidensis* control and the corresponding optical pictures (left inset: *S. oneidensis*- Cu_{2-x}Se , right inset: *S. oneidensis*). (B) I-t curves of the *S. oneidensis*- Cu_{2-x}Se and *S. oneidensis* control with a light on/off cycle (50/50 s). (C) UV-vis-NIR absorption spectra of purified Cu_{2-x}Se NPs (in CCl_4).

Supplementary Figure 19. Schematic illustration of the working mechanism of Bio- $\text{Cu}_{2-x}\text{Se}@MR-1@PVDF$ membrane in the water desalination process.

4. Also, the high efficiency should be verified by the use of mutant strains and explained quantitatively.

Thanks. To explore whether the high efficiency of Bio- $\text{Cu}_{2-x}\text{Se}@MR-1@PVDF$ membrane was attributed to the biogenic Cu_{2-x}Se NPs, we employed $\Delta cymA$ to synthesize Cu_{2-x}Se NPs, which were then encapsulated in PVDF substrate to fabricate a solar steam generation device ($\Delta cymA$ -NPs@MR-1@PVDF). As expected, destroying the CymA protein synthesis severely inhibited Cu_{2-x}Se NPs production under dark conditions (**Supplementary Figure 17A**), resulting in a substantial decrease in

photothermal performance, such as evaporation rate and photothermal conversion efficiency (Supplementary Figure 17B-17C). In comparison, the solar evaporation efficiency of Bio-Cu_{2-x}Se@MR-1@PVDF membrane (using WT as bio-nano-factory) was 1.73 times that of Δ*cymA*-NPs@MR-1@PVDF (using Δ*cymA* as bio-nano-factory). Hence, Cu_{2-x}Se NPs contents is closely linked to the photothermal evaporation performance of the membrane.

Accepting the reviewer’s suggestion, we have updated the manuscript as follows:

“To further explore whether the high efficiency of Bio-Cu_{2-x}Se@MR-1@PVDF membrane was attributed to the biogenic Cu_{2-x}Se NPs, we employed Δ*cymA* to synthesize Cu_{2-x}Se NPs, which were then encapsulated in PVDF substrate to fabricate a solar steam generation device (Δ*cymA*-NPs@MR-1@PVDF). As expected, destroying the CymA protein synthesis severely inhibited Cu_{2-x}Se NPs production under dark conditions (Supplementary Fig. 17A), resulting in a substantial decrease in photothermal performance (Supplementary Fig. 17B-17C). The solar evaporation rate of Δ*cymA*-NPs@MR-1@PVDF is only 0.89 kg m⁻² h⁻¹, which is much lower than that of Bio-Cu_{2-x}Se@MR-1@PVDF (using WT as bio-nano-factory) (Supplementary Fig. 17B). In comparison, the solar evaporation efficiency of Bio-Cu_{2-x}Se@MR-1@PVDF membrane was 1.73 times that of Δ*cymA*-NPs@MR-1@PVDF. Thus, the high efficiency of Bio-Cu_{2-x}Se@MR-1@PVDF membrane was tightly associated with the amount of Cu_{2-x}Se NPs.” (P20, L416-426)

Supplementary Figure 17. Solar vapor generation performance of Bio-Cu_{2-x}Se@MR-1@PVDF membrane (using WT as bio-nano-factory) and Δ*cymA*-NPs@MR-1@PVDF (using Δ*cymA* as bio-nano-factory). (A) Schematic diagram of the Bio-Cu_{2-x}Se of solid powder and corresponding Bio-Cu_{2-x}Se@MR-1@PVDF membrane. (B) Evaporation rate of membranes under 1 sun irradiation. (C) Photothermal conversion efficiency of membranes under 1 sun irradiation. The synthetic precursors of biogenic Cu_{2-x}Se were all lysed hybrid-Se⁰ NPs. The data points represent means ± SD (n = 3). Error bars correspond to standard deviations. “****” represents p < 0.0001.

“Fabrication of Δ*cymA*-NPs@MR-1@PVDF Membrane (using Δ*cymA* Strain as Bio-Nano-Factory)

Biogenic Cu_{2-x}Se NPs were synthesized by Δ*cymA* following the procedure in Supplementary Fig. 1B. The resulting Δ*cymA*-nanoparticles hybrid was collected, freeze-dried, ground, and then used for

membrane preparation. The casting solution was prepared by mixing polyvinyl pyrrolidone (PVP) and N,N-dimethylformamide (DMF) according to Supplementary Table 3.”(SI, P8, L175-180)

Reviewer 3's comments

Recommendation: Significant improvements have been made on the revised manuscript, and the majority of the reviewers' comments have been addressed. After careful consideration of the revised manuscript, I recommend its publication in Nature Communications.

Thanks.

REFERENCES

1. Fan, G., Dundas, C.M., Graham, A.J., Lynd, N.A. & Keitz, B.K. *Shewanella oneidensis* as a living electrode for controlled radical polymerization. *Proc. Natl. Acad. Sci. U. S. A.* **115**, 4559-4564 (2018).
2. Kimber, R.L. et al. Biosynthesis and characterization of copper nanoparticles using *Shewanella oneidensis*: application for click chemistry. *Small* **14**, 1703145 (2018).
3. Ghorai, N. & Ghosh, H.N. Ultrafast plasmon dynamics in near-infrared active non-stoichiometric Cu_{2-x}Se nanocrystals and effect of chemical interface damping. *J. Phys. Chem. C* **125**, 11468-11477 (2021).

Reviewers' Comments:

Reviewer #2:

Remarks to the Author:

The authors addressed all the questions well. Additional experiments support their idea that bacteria and biosynthesized particle synergetically enhance the membrane function for water evaporation.